# A KERNEL DISTRIBUTION CLOSENESS TESTING

## ABSTRACT

The *distribution closeness testing* (DCT) assesses whether the distance between an unknown distribution pair is at least $\epsilon$-far; in practice, the $\epsilon$ can be defined as the distance between a reference (known) distribution pair. However, existing DCT methods are mainly measure discrepancies between a distribution pair defined on discrete one-dimensional spaces (e.g., total variation on a discrete one-dimensional space), which limits the DCT to be used on complex data (e.g., images). To make DCT applicable on complex data, a natural idea is to introduce the *maximum mean discrepancy* (MMD), a powerful measurement to see the difference between a pair of two complex distributions, to DCT scenarios. Nonetheless, in this paper, we find that MMD value is less informative when assessing the closeness levels for multiple distribution pairs with the same kernel, i.e., MMD value can be the same for many pairs of distributions that have different norms in the same *reproducing kernel Hilbert space* (RKHS). To mitigate the issue, we propose a new kernel DCT with the *norm-adaptive MMD* (NAMMD) by scaling MMD with the norms of distributions, effective for kernels $\kappa(\boldsymbol{x}, \boldsymbol{x}') = \Psi(\boldsymbol{x} - \boldsymbol{x}') \leq K$ with a positive-definite $\Psi(\cdot)$ and $\Psi(\boldsymbol{0}) = K$. Theoretically, we prove that our NAMMD test achieves higher test power compared to the MMD test, along with asymptotic distribution analysis. We also present upper bounds on the sample complexity of our NAMMD test and prove that Type-I error is controlled. We finally conduct experiments to validate the effectiveness of our NAMMD test.

## 1 INTRODUCTION

Assessing difference between a distribution pair is important in the field of machine learning, because the test and training data are from different distributions in many real-world scenarios [1]. Thus, tons of research has been done on problem settings where distributional differences exist. Two phenomena can be observed in the literature. On the one hand, a large distributional discrepancy between training and test data might cause poor performance on test data for a model trained on the training data [2]. This phenomenon can be theoretically explained by the domain adaptation theory [3]. On the other hand, it is also empirically proved that models trained on a large dataset (e.g., ImageNet [4]) can have good performance on relevant/similar downstream test data (e.g., Pascal VOC [5]) that is different from training dataset [6]. This means that, even if training and test data are from different distributions, we can still expect relatively good performance because they might be close to each other.

Therefore, *seeing to what statistically significant extent* two distributions are close to each other is important and might have the potential to help us decide if we really need to adapt a model when we observe upcoming data that follow a different distribution from training data. Two-sample testing can naturally help see if training and test data are from the same distribution [7], but it is less useful in the second phenomenon above as we might also have good empirical performance when the training and test data are close to each other. Fortunately, in the field of theoretical computer science, researchers have proposed *distribution closeness testing* to see if the distance between a distribution pair is at least $\epsilon$-far, including two-sample testing as a specific case with $\epsilon = 0$ [8–11]. This kind of testing exactly fits the aim of *seeing to what statistically significant extent* two distributions are close to each other. Distribution closeness testing has been used to evaluate Markov chain mixing time [12], testing language membership [13], analyzing feature combinations [14].

However, existing distribution closeness testing methods mainly measure closeness using total variation [15–18], and primarily focus on the theoretical analysis of the sample complexity of sublinear algorithms applied to *discrete one-dimensional distributions* defined on a support set only containing finite elements (e.g., distribution defined on a positive-integer domain $\{1, 2, ..., n\}$). This

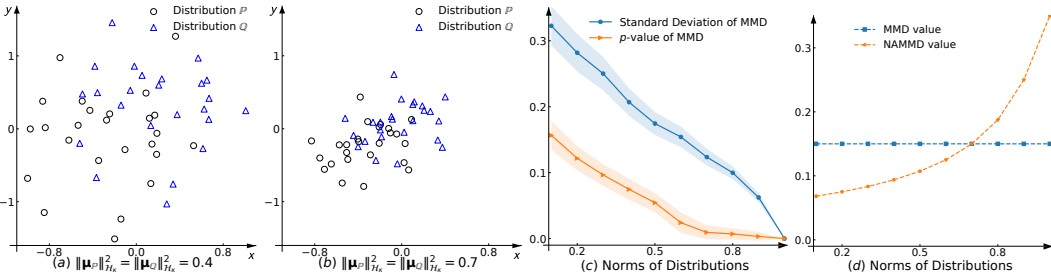

**Figure 1:** All visualizations are presented with a constant MMD value $\|\boldsymbol{\mu}_{\mathbb{P}} - \boldsymbol{\mu}_{\mathbb{Q}}\|^2_{\mathcal{H}_\kappa} = 0.15$ on a Gaussian kernel, extendable to other kernels of the form: $\kappa(\boldsymbol{x}, \boldsymbol{x}') = \Psi(\boldsymbol{x} - \boldsymbol{x}') \leq K$ for a positive-definite $\Psi(\cdot)$ and $\Psi(\mathbf{0}) = K$. (Relevant Limitation Statement regarding kernel forms can be found in C.4) Panels (a) and (b) depict distribution $\mathbb{P}$ and $\mathbb{Q}$ with varying norms, i.e. $\|\boldsymbol{\mu}_{\mathbb{P}}\|^2_{\mathcal{H}_\kappa}$ and $\|\boldsymbol{\mu}_{\mathbb{Q}}\|^2_{\mathcal{H}_\kappa}$. Panel (c) presents the standard deviations and $p$-values of the empirical MMD estimator in the two-sample testing. Panel (d) presents the values of original MMD and our NAMMD as the norms of distributions increase.

limits the distribution closeness testing to be used on complex data (e.g., images) often used in the machine learning tasks (e.g., image classification task).

Although it is possible to discretize complex data to a simple support set (then conducting distribution closeness testing using existing methods [19]), it is not easy to maintain intrinsic structures and patterns of complex data after the discretization [20, 21]. To handle complex data, in the literature, the kernel trick is also helpful to measure the closeness in higher-dimensional spaces [22]. Significant efforts have been made to apply the kernel trick in hypothesis testing statistics, including Hilbert-Schmidt Independence Criterion for independence testing [23], Kernel Stein Discrepancy for good-of-fitness testing [24, 25], Maximum Mean Discrepancy (MMD) for two-sample testing [7].

Since MMD is an effective measurement to see the distributional discrepancy [26] and is frequently used in two-sample testing tasks [27] (a special case of distribution closeness testing), there is a natural idea to introduce MMD to distribution closeness testing. MMD provides a versatile approach across both discrete and continuous domains, and many approaches have extended it to various scenarios, including mean embeddings with test locations [28, 29], local difference exploration [30], stochastic process [31], multiple kernel [32, 33], adversarial learning [34], and domain adaptation [35]. Yet, no one has explored how to extend distribution closeness testing to complex data with MMD.

In this paper, however, we find it is not ideal to directly use MMD in distribution closeness testing, because the MMD is less informative when comparing the closeness levels of different distribution pairs for a fixed kernel $\kappa$. Specifically, the MMD value can be the same for many pairs of distributions that have different norms in the RKHS $\mathcal{H}_\kappa$, which actually reflect different closeness levels for these distribution pairs. We present an example to analyze the above issue on a Gaussian kernel, extendable to other characteristic kernels of the form $\kappa(\boldsymbol{x}, \boldsymbol{x}') = \Psi(\boldsymbol{x} - \boldsymbol{x}') \leq K$ with a positive-definite $\Psi(\cdot)$ and $\Psi(\mathbf{0}) = K$, including Laplace [33], Mahalanobis [30] and Deep kernels [27] (frequently used in kernel-based hypothesis testing). Denote by $\|\boldsymbol{\mu}_{\mathbb{P}}\|^2_{\mathcal{H}_\kappa}$ and $\|\boldsymbol{\mu}_{\mathbb{Q}}\|^2_{\mathcal{H}_\kappa}$ the norms of distributions $\mathbb{P}$ and $\mathbb{Q}$, respectively. We can observe that larger norms imply smaller variances $\mathrm{Var}(\mathbb{P}, \kappa) = 1 - \|\boldsymbol{\mu}_{\mathbb{P}}\|^2_{\mathcal{H}_\kappa}$ and $\mathrm{Var}(\mathbb{Q}, \kappa) = 1 - \|\boldsymbol{\mu}_{\mathbb{Q}}\|^2_{\mathcal{H}_\kappa}$, indicating more tightly concentrated distributions as shown in Figure 1a and 1b. Nonetheless, the MMD values regarding Figure 1a and 1b are the same, showing a case where MMD is less informative in comparing multiple distribution pairs with different norms.

Furthermore, we adapt the standard deviation and $p$-value of the empirical MMD estimator in two-sample testing to see if pairs of distributions in Figures 1a and 1b have the same closeness. As illustrated in Figures 1c, the standard deviation decreases as the norms of distributions increase, which is a result of the more tightly concentrated distributions. As we know, a smaller standard deviation signifies a reduced probability of the empirical MMD estimator falling outside the expected range, resulting in a smaller $p$-value. Therefore, the standard deviation decreases as the norms of distributions increase, even when the MMD value is held at a constant $0.15$ as shown in Figures 1c. Notably, smaller $p$-values indicate more significant difference and less closeness between distributions. Hence, the pairs of distributions in Figures 1a and 1b actually have different levels of closeness.

We mitigate the above issue by scaling MMD value with the norms of distributions, and we propose a new kernel distribution closeness testing called the *norm-adaptive MMD* (NAMMD) test. Specifically, our NAMMD distance is scaled up as the norms of distributions increase, while the MMD value is held at constant. Figure 1c and 1d illustrate that our NAMMD exhibits a stronger correlation with the $p$-

value. This enhancement in correlation translates to improved test power, as supported by comparisons between NAMMD and the original MMD under the same kernel, as outlined in Theorem 10 and 12.

In the above analysis, we use a fixed global kernel for different distribution pairs, which is essential for effectively comparing their closeness levels under a unified distance measurement. Yet, existing kernel selection methods are primarily designed for two-sample testing [27, 36], focusing on selecting a kernel that maximizes the test power estimator to distinguish a fixed distribution pair $\mathbb{P}$ and $\mathbb{Q}$. Despite efforts to extend these kernel selections to distribution closeness testing, deriving a test power estimator with multiple distribution pairs remains an open question and poses a significant challenge.

When we want to test distribution closeness, we could use a reference (known) pair of distributions $\mathbb{P}_1$ and $\mathbb{Q}_1$, with their distance serving as threshold $\epsilon$. Here, the global kernel can be selected by maximizing the test power estimator of $\mathbb{P}_1$ and $\mathbb{Q}_1$ following two-sample testing methods. With the kernel, we then test whether the distance between an unknown distribution pair $\mathbb{P}_2$ and $\mathbb{Q}_2$ exceeds that between $\mathbb{P}_1$ and $\mathbb{Q}_1$. Given this, we conduct experiments to validate the effectiveness of our NAMMD test, including three case studies demonstrating its application in evaluating whether a model performs similarly across training and testing datasets without ground truth labels (Section 5.2).

## 2 PRELIMINARIES

**Distribution Closeness Testing.** Distribution closeness testing accesses whether two unknown discrete distributions are $\epsilon$-far from each other in the closeness measure. Let $\mathbb{P}_n = \{p_1, p_2, ..., p_n\}$ and $\mathbb{Q}_n = \{q_1, q_2, ..., q_n\}$ be two discrete distributions over domain $Z = \{\boldsymbol{z}_1, \boldsymbol{z}_2, ..., \boldsymbol{z}_n\} \subseteq \mathbb{R}^d$ such that $\sum_{i=1}^n p_i = 1$ and $\sum_{i=1}^n q_i = 1$. We define the total variation [37] of $\mathbb{P}_n$ and $\mathbb{Q}_n$ as

$$\mathrm{TV}(\mathbb{P}_n, \mathbb{Q}_n) = \sup_{S \subseteq Z} (\mathbb{P}_n(S) - \mathbb{Q}_n(S)) = \frac{1}{2} \sum_{i=1}^n |p_i - q_i| = \frac{1}{2} \|\mathbb{P}_n - \mathbb{Q}_n\|_1 \in [0, 1] .$$

Taking the total variation as the closeness measure for distribution closeness, the goal is to test between the null and alternative hypothesis as follows

$$\boldsymbol{H}_0' : \mathrm{TV}(\mathbb{P}_n, \mathbb{Q}_n) \leq \epsilon' \quad \text{and} \quad \boldsymbol{H}_1' : \mathrm{TV}(\mathbb{P}_n, \mathbb{Q}_n) > \epsilon',$$

where $\epsilon' \in [0, 1)$ denotes the predetermined closeness parameter.

**Maximum Mean Discrepancy.** The MMD [26] is a typical kernel-based distance between two distributions. Denote by $\mathbb{P}$ and $\mathbb{Q}$ two Borel probability measures over an instance space $\mathcal{X} \subseteq \mathbb{R}^d$. Let $\kappa : \mathcal{X} \times \mathcal{X} \to \mathbb{R}$ be the kernel of a reproducing kernel Hilbert space $\mathcal{H}_\kappa$, with feature map $\kappa(\cdot, \boldsymbol{x}) \in \mathcal{H}_\kappa$ and $0 \leq \kappa(\boldsymbol{x}, \boldsymbol{y}) \leq K$. The kernel mean embeddings [38, 39] of $\mathbb{P}$ and $\mathbb{Q}$ are given as

$$\boldsymbol{\mu}_\mathbb{P} = E_{\boldsymbol{x} \sim \mathbb{P}}[\kappa(\cdot, \boldsymbol{x})] \quad \text{and} \quad \boldsymbol{\mu}_\mathbb{Q} = E_{\boldsymbol{y} \sim \mathbb{Q}}[\kappa(\cdot, \boldsymbol{y})] .$$

We now define the MMD of $\mathbb{P}$ and $\mathbb{Q}$ as

$$\mathrm{MMD}^2(\mathbb{P}, \mathbb{Q}, \kappa) = \|\boldsymbol{\mu}_\mathbb{P} - \boldsymbol{\mu}_\mathbb{Q}\|_{\mathcal{H}_\kappa}^2 = E[\kappa(\boldsymbol{x}, \boldsymbol{x}') + \kappa(\boldsymbol{y}, \boldsymbol{y}') - 2\kappa(\boldsymbol{x}, \boldsymbol{y})] \in [0, 2K] ,$$

where the expectation are taken with respect to $\boldsymbol{x}, \boldsymbol{x}' \sim \mathbb{P}$ and $\boldsymbol{y}, \boldsymbol{y}' \sim \mathbb{Q}$.

For characteristic kernels, $\mathrm{MMD}(\mathbb{P}, \mathbb{Q}, \kappa) = 0$ if and only if $\mathbb{P} = \mathbb{Q}$. Hence, MMD can be readily applied to the two-sample testing with null and alternative hypotheses as follows

$$\boldsymbol{H}_0'' : \mathbb{P} = \mathbb{Q} \quad \text{and} \quad \boldsymbol{H}_1'' : \mathbb{P} \neq \mathbb{Q} ,$$

which can be viewed as a specific case of distribution closeness testing using MMD and setting $\epsilon = 0$.

## 3 THE PROPOSED NAMMD TEST

As discussed in introduction and shown in Figure 1, while MMD can detect whether two distributions are identical, it is less informative in measuring the closeness between distributions. Specifically, different pairs of distributions with varying norms in the RKHS can yield the same MMD value, despite having different levels of closeness, as revealed through the analysis of $p$-values.

**NAMMD Distance and Its Asymptotic Property.** We define our NAMMD distance as follows.

**Definition 1.** Let $\kappa$ be the kernel of $\mathcal{H}_\kappa$ and $0 \leq \kappa(\boldsymbol{x}, \boldsymbol{y}) \leq K$. Let $\boldsymbol{x}, \boldsymbol{x}' \sim \mathbb{P}$ and $\boldsymbol{y}, \boldsymbol{y}' \sim \mathbb{Q}$ with $\boldsymbol{\mu}_\mathbb{P}$ and $\boldsymbol{\mu}_\mathbb{Q}$. We define the *norm-adaptive maximum mean discrepancy* (NAMMD) as follows:

$$\mathrm{NAMMD}(\mathbb{P}, \mathbb{Q}, \kappa) = \frac{\|\boldsymbol{\mu}_\mathbb{P} - \boldsymbol{\mu}_\mathbb{Q}\|_{\mathcal{H}_\kappa}^2}{4K - \|\boldsymbol{\mu}_\mathbb{P}\|_{\mathcal{H}_\kappa}^2 - \|\boldsymbol{\mu}_\mathbb{Q}\|_{\mathcal{H}_\kappa}^2} = \frac{E[\kappa(\boldsymbol{x}, \boldsymbol{x}') + \kappa(\boldsymbol{y}, \boldsymbol{y}') - 2\kappa(\boldsymbol{x}, \boldsymbol{y})]}{4K - E[\kappa(\boldsymbol{x}, \boldsymbol{x}')] - E[\kappa(\boldsymbol{y}, \boldsymbol{y}')]} , \quad (1)$$

and it is clear that $\mathrm{NAMMD} \in [0, 1]$. Here, the value of NAMMD approaches 1 when the two distributions are well-separated and both highly concentrated.

**Remark.** In NAMMD, we essentially capture differences between two distributions using their characteristic kernel mean embeddings (i.e. $\boldsymbol{\mu}_{\mathbb{P}}$ and $\boldsymbol{\mu}_{\mathbb{Q}}$), which uniquely represent probability distributions and capture distinct characteristics for effective comparison [40]. A natural way to measure the difference is by the Euclidean-like distance $\|\boldsymbol{\mu}_{\mathbb{P}} - \boldsymbol{\mu}_{\mathbb{Q}}\|^2_{\mathcal{H}_\kappa}$ (i.e., MMD). However, as discussed in Section 1, MMD can yields same value for many pairs of distributions that have different norms with the same kernel (which results in different closeness levels). To mitigate the issue, we scale it using $4K - \|\boldsymbol{\mu}_{\mathbb{P}}\|^2_{\mathcal{H}_\kappa} - \|\boldsymbol{\mu}_{\mathbb{Q}}\|^2_{\mathcal{H}_\kappa}$, making NAMMD increase with the norms $\|\boldsymbol{\mu}_{\mathbb{P}}\|^2_{\mathcal{H}_\kappa}$ and $\|\boldsymbol{\mu}_{\mathbb{Q}}\|^2_{\mathcal{H}_\kappa}$. This leverages an insight that we separate two distributions more effectively at same MMD distance with larger norms. Figure 1c and 1d demonstrate that our NAMMD exhibits a stronger correlation with the $p$-value in testing, while MMD is held constant. We also prove that scaling improves NAMMD's effectiveness as a closeness measure in Theorems 10 and 12.

Probability measures $\mathbb{P}$ and $\mathbb{Q}$ are generally unknown, and we can only observe are two i.i.d. samples

$$X = \{\boldsymbol{x}_i\}_{i=1}^m \sim \mathbb{P}^m \text{ and } Y = \{\boldsymbol{y}_j\}_{j=1}^m \sim \mathbb{Q}^m \ .$$

Following Liu et al. [27], we assume equal size for two samples to simplify the notation, yet our results can be easily extended to unequal sample sizes by changing the empirical estimator.

Based on two samples $X$ and $Y$, we introduce the empirical estimator of NAMMD as follows

$$\widehat{\mathrm{NAMMD}}(X, Y, \kappa) = \sum_{i \neq j} H_{i,j} / \sum_{i \neq j} [4K - \kappa(\boldsymbol{x}_i, \boldsymbol{x}_j) - \kappa(\boldsymbol{y}_i, \boldsymbol{y}_j)] \ ,$$

where $H_{i,j} = \kappa(\boldsymbol{x}_i, \boldsymbol{x}_j) + \kappa(\boldsymbol{y}_i, \boldsymbol{y}_j) - \kappa(\boldsymbol{x}_i, \boldsymbol{y}_j) - \kappa(\boldsymbol{y}_i, \boldsymbol{x}_j)$.

We then present asymptotic behavior of our empirical estimator as follows.

**Theorem 2.** *If* $\mathrm{NAMMD}(\mathbb{P}, \mathbb{Q}, \kappa) = \epsilon$ *with* $\epsilon \in (0, 1]$*, we have*

$$\sqrt{m}(\widehat{\mathrm{NAMMD}}(X, Y, \kappa) - \epsilon) \xrightarrow{d} \mathcal{N}(0, \sigma^2_{\mathbb{P},\mathbb{Q}}) \ ,$$

*where* $\sigma_{\mathbb{P},\mathbb{Q}} = \sqrt{4E[H_{1,2}H_{1,3}] - 4(E[H_{1,2}])^2}/(4K - \|\boldsymbol{\mu}_{\mathbb{P}}\|^2_{\mathcal{H}_\kappa} - \|\boldsymbol{\mu}_{\mathbb{Q}}\|^2_{\mathcal{H}_\kappa})$*, and the expectation are taken over* $\boldsymbol{x}_1, \boldsymbol{x}_2, \boldsymbol{x}_3 \sim \mathbb{P}^3$ *and* $\boldsymbol{y}_1, \boldsymbol{y}_2, \boldsymbol{y}_3 \sim \mathbb{Q}^3$*; and if* $\mathrm{NAMMD}(\mathbb{P}, \mathbb{Q}, \kappa) = 0$*, we have*

$$m\widehat{\mathrm{NAMMD}}(X, Y, \kappa) \xrightarrow{d} \sum_i \lambda_i \left( Z_i^2 - 2 \right) / (4K - \|(\boldsymbol{\mu}_{\mathbb{P}} + \boldsymbol{\mu}_{\mathbb{Q}})/\sqrt{2}\|^2_{\mathcal{H}_\kappa}) \ ,$$

*where* $Z_i \sim \mathcal{N}(0, 2)$*, and the* $\lambda_i$ *are eigenvalues of the* $\mathbb{P}$*-covariance operator of the centered kernel.*

Building on this result, we now present the distribution closeness testing by taking our NAMMD as the measure of closeness, along with an appropriately estimated testing threshold.

**NAMMD Testing Procedure.** We now define the distribution closeness testing as follows.

**Definition 3.** Given the closeness parameter $\epsilon \in [0, 1)$, the goal is to test between hypotheses

$$\boldsymbol{H}_0 : \mathrm{NAMMD}(\mathbb{P}, \mathbb{Q}, \kappa) \leq \epsilon \text{ and } \boldsymbol{H}_1 : \mathrm{NAMMD}(\mathbb{P}, \mathbb{Q}, \kappa) > \epsilon$$

with the significance level $\alpha \in (0, 1)$.

To perform testing procedure for above definition, we need to determine the testing threshold $\tau_\alpha$ based on Theorem 2. This can be outlined under two asymptotic scenarios 1): when $\mathrm{NAMMD}(\mathbb{P}, \mathbb{Q}, \kappa) = \epsilon$ with $\epsilon \in (0, 1)$ and 2): when $\mathrm{NAMMD}(\mathbb{P}, \mathbb{Q}, \kappa) = 0$. In the first scenario, which corresponds to the null hypothesis $\boldsymbol{H}_0 : \mathrm{NAMMD}(\mathbb{P}, \mathbb{Q}, \kappa) \leq \epsilon$ with $\epsilon \in (0, 1)$, we set $\tau_\alpha$ as the $(1 - \alpha)$-quantile of the asymptotic Gaussian distribution in Theorem 2 (which can be easily calculated). Here, the term $\sigma^2_{\mathbb{P},\mathbb{Q}}$ is unknown in practice and we use the empirical estimator

$$\sigma_{X,Y} = \frac{\sqrt{((4m - 8)\zeta_1 + 2\zeta_2)/(m - 1)}}{(m^2 - m)^{-1} \sum_{i \neq j} 4K - \kappa(\boldsymbol{x}_i, \boldsymbol{x}_j) - \kappa(\boldsymbol{y}_i, \boldsymbol{y}_j)} \ ,$$

where $\zeta_1$ and $\zeta_2$ are standard variance components of the MMD [41, 42]. We present the details of the estimator in Appendix C.2 due to page limitations.

We have the testing threshold for the null hypothesis $\boldsymbol{H}_0 : \mathrm{NAMMD}(\mathbb{P}, \mathbb{Q}, \kappa) \leq \epsilon$ with $\epsilon \in (0, 1)$ as

$$\tau_\alpha = \epsilon + \sigma_{X,Y} \mathcal{N}_{1-\alpha}/\sqrt{m} \ , \tag{2}$$

where $\mathcal{N}_{1-\alpha}$ is the $(1 - \alpha)$-quantile of the standard normal distribution.

In the second scenario, which corresponds to the null hypothesis $\boldsymbol{H}_0 : \text{NAMMD}(\mathbb{P}, \mathbb{Q}, \kappa) = 0$ (i.e. $\epsilon = 0$ in Definition 3), the problem reduces to the standard two-sample testing based on Lemma 9. In this case, it is challenging to directly estimate the null distribution [43]. We instead use the simpler permutation test to obtain $\tau_\alpha$ [36], which estimate the null distribution by repeatedly re-computing estimator with the samples randomly re-assigned to $X$ or $Y$.

Specifically, denote by $B$ the iteration number of permutation test. Let $\boldsymbol{\Pi}_{2m}$ be the set of all possible permutations of $\{1, \ldots, 2m\}$ over the pooled sample $Z = \{\boldsymbol{x}_1, \ldots, \boldsymbol{x}_m, \boldsymbol{y}_1, \ldots, \boldsymbol{y}_m\} = \{\boldsymbol{z}_1, \ldots, \boldsymbol{z}_m, \boldsymbol{z}_{m+1}, \ldots, \boldsymbol{z}_{2m}\}$. In $b$-th iteration ($b \in [B]$), we generate a permutation $\boldsymbol{\pi} = (\pi_1, \ldots, \pi_{2m}) \in \boldsymbol{\Pi}_{2m}$ and then calculate the empirical estimator of NAMMD statistic as follows

$$T_b = \widehat{\text{NAMMD}}(X_{\boldsymbol{\pi}}, Y_{\boldsymbol{\pi}}, \kappa) \,,$$

where $X_{\boldsymbol{\pi}} = \{\boldsymbol{z}_{\pi_1}, \boldsymbol{z}_{\pi_2}, ..., \boldsymbol{z}_{\pi_m}\}$ and $Y_{\boldsymbol{\pi}} = \{\boldsymbol{z}_{\pi_{m+1}}, \boldsymbol{z}_{\pi_{m+2}}, ..., \boldsymbol{z}_{\pi_{2m}}\}$.

During such process, we obtain $B$ statistics $T_1, T_2, ..., T_B$ and introduce the testing threshold for the null hypothesis $\boldsymbol{H}_0 : \text{NAMMD}(\mathbb{P}, \mathbb{Q}, \kappa) = 0$ as follows

$$\tau_\alpha = \arg\min_\tau \left\{ \sum_{b=1}^B \frac{\mathbb{I}[T_b \leq \tau]}{B} \geq 1 - \alpha \right\} \,. \tag{3}$$

Finally, we have the following test with testing threshold $\tau_\alpha$ from either Eqn. 2 or 3

$$h(X, Y, \kappa) = \mathbb{I}[\widehat{\text{NAMMD}}(X, Y, \kappa) > \tau_\alpha] \,. \tag{4}$$

For the variance estimator, we present its asymptotic behavior as follows.

**Lemma 4.** *Given samples $X$ and $Y$ with size $m$, we have that $\left| E[\sigma_{X,Y}^2] - \sigma_{\mathbb{P},\mathbb{Q}}^2 \right| = O(1/\sqrt{m})$.*

We present theoretical analysis for Type-I error as follows.

**Theorem 5.** *Under null hypothesis $\boldsymbol{H}_0 : \text{NAMMD} \leq \epsilon$, Type-I error of NAMMD is bounded by $\alpha$.*

**Performing Distribution Closeness Testing in Practice.** We have demonstrated how to perform distribution closeness testing above, yet it is still not clear how the $\epsilon$ of Definition 3 should be set in practice. Normally, when we want to test the closeness, we often have a reference pair of distributions $\mathbb{P}_1$ and $\mathbb{Q}_1$ that we know its true/approximate distributional discrepancy, i.e. $\text{NAMMD}(\mathbb{P}_1, \mathbb{Q}_1, \kappa)$.

Then, given two samples $X$ and $Y$ drawn from unknown distributions $\mathbb{P}_2$ and $\mathbb{Q}_2$ respectively, we seek to determine whether the distance between $\mathbb{P}_2$ and $\mathbb{Q}_2$ is as close or closer to that between $\mathbb{P}_1$ and $\mathbb{Q}_1$, by applying distribution closeness testing. Here, we set $\epsilon = \text{NAMMD}(\mathbb{P}_1, \mathbb{Q}_1, \kappa)$, and this can be formalized by Definition 3 with null and alternative hypotheses as follows

$$\boldsymbol{H}_0 : \text{NAMMD}(\mathbb{P}_2, \mathbb{Q}_2, \kappa) \leq \epsilon \quad \text{and} \quad \boldsymbol{H}_1 : \text{NAMMD}(\mathbb{P}_2, \mathbb{Q}_2, \kappa) > \epsilon \,.$$

Finally, we can perform the NAMMD test procedure with samples $X$ and $Y$.

**Relevant work.** A well-known class of two-sample testing constructs kernel embeddings for each distribution and then test the differences between these embeddings [44–47]. Another relevant approach assesses the differences between distributions with classification performance [48–56]. Kernel-based MMD has been one of the most important statistic for two-sample testing, which includes popular classifier-based two-sample testing approaches as a special case [27].

Previous distribution closeness testing approaches primarily focus on theoretical analysis of the sample complexity of sub-linear algorithms, and these approaches often rely on total variation over discrete one-dimensional distributions [12, 15–18]. Other measures of closeness also include $\ell_2$ distance [57–59], entropy [60], probability difference [8, 61], etc. In comparison, we turn to kernel methods that have shown effectiveness in non-parametric testing.

Permutation tests are widely used in statistics for equality of distributions, providing a finite-sample guarantee on the Type-I error whenever the samples are exchangeable under null hypothesis [62–65]. As shown in Lemma 9, $\text{NAMMD}(\mathbb{P}, \mathbb{Q}, \kappa) = 0$ if and only if $\mathbb{P} = \mathbb{Q}$, indicating that our NAMMD satisfies the exchangeability under null hypothesis $\boldsymbol{H}_0 : \text{NAMMD}(\mathbb{P}, \mathbb{Q}, \kappa) = 0$. For null hypothesis $\boldsymbol{H}_0 : \text{NAMMD}(\mathbb{P}, \mathbb{Q}, \kappa) \leq \epsilon$ and $\epsilon \in (0, 1)$, the empirical estimator of our NAMMD distance, i.e., $\text{NAMMD}(\mathbb{P}, \mathbb{Q}, \kappa) = \epsilon$, has an asymptotic Gaussian distribution as shown in Theorem 2. Hence, we use the $(1 - \alpha)$-quantile of asymptotic distribution as the testing threshold following [30, 46, 47].

Some approaches select kernels in a supervised manner using held-out data [29, 36], while others rely on unsupervised methods, such as the median heuristic [26], or adaptively combine multiple kernels [32, 33]. Our NAMMD is compatible with these methods; for instance, the kernel can be selected by maximizing the test power estimator derived from Theorem 2 (details are provided in Appendix C.1). However, these approaches are primarily designed for distinguishing between a fixed distribution pair in two-sample testing. It remains an open question and an important future work to select an optimal global kernel for distribution closeness testing with multiple distribution pairs.

## 4 THEORETICAL ANALYSIS

In this section, we make further theoretical investigations on our NAMMD and the comparison between our NAMMD and the original MMD in two-sample testing and distribution closeness testing.

### 4.1 SAMPLE COMPLEXITY OF OUR NAMMD

We now present the large deviation bound for our NAMMD estimator.

**Lemma 6.** *The following holds over sample $X$ and $Y$ of size $m$,*

$$\Pr\left(|\widehat{\mathrm{NAMMD}}(X, Y, \kappa) - \mathrm{NAMMD}(\mathbb{P}, \mathbb{Q}, \kappa)| \geq t\right) \leq 4\exp(-mt^2/9) \ \textit{for } t > 0.$$

We present the concentration of our NAMMD estimator with permuted two samples as follows.

**Lemma 7.** *Let $\mathbf{\Pi}_{2m}$ be the set of permutations over sample $Z = \{\boldsymbol{x}_1, \ldots, \boldsymbol{x}_m, \boldsymbol{y}_1, \ldots, \boldsymbol{y}_m\}$ and $0 \leq \kappa(\boldsymbol{x}, \boldsymbol{y}) \leq K$. Given a permutation $\boldsymbol{\pi}$, we have permuted two samples $X_{\boldsymbol{\pi}}$ and $Y_{\boldsymbol{\pi}}$. Then,*

$$\Pr\left(\widehat{\mathrm{NAMMD}}(X_{\boldsymbol{\pi}}, Y_{\boldsymbol{\pi}}, \kappa) \geq t\right) \leq \exp\left\{-C\min\left(4K^2t^2/\Sigma_m^2, 2Kt/\Sigma_m\right)\right\},$$

*for every $t > 0$ and some constant $C > 0$, where*

$$\Sigma_m^2 := \frac{1}{m^2(m-1)^2} \sup_{\boldsymbol{\pi} \in \mathbf{\Pi}_{2m}} \left\{\sum_{i \neq j}^m \kappa^2\left(\boldsymbol{z}_{\pi_i}, \boldsymbol{z}_{\pi_j}\right)\right\}.$$

We now derive upper bounds on the sample complexity required for our NAMMD test to correctly reject the null hypothesis with high probability as follows.

**Theorem 8.** *For our NAMMD test, as formalized in Eqn. 4, we correctly reject null hypothesis with probability at least $1 - \upsilon$ given the sample size*

$$m \geq \frac{\left(\sqrt{9\log 2/\upsilon} + \sqrt{9\log 2/\upsilon + 2C_{\alpha}\mathrm{NAMMD}(\mathbb{P}, \mathbb{Q}, \kappa)}\right)^2}{4 \cdot \mathrm{NAMMD}^2(\mathbb{P}, \mathbb{Q}, \kappa)} + 1,$$

*if $\epsilon = 0$ and $\mathrm{NAMMD}(\mathbb{P}, \mathbb{Q}, \kappa) \in (0, 1]$; and this is also holds given*

$$m \geq \left(2 * \mathcal{N}_{1-\alpha} + \sqrt{9\log 2/\upsilon}\right)^2 / (\mathrm{NAMMD}(\mathbb{P}, \mathbb{Q}, \kappa) - \epsilon)^2,$$

*if $\epsilon \in (0, 1)$ and $\mathrm{NAMMD}(\mathbb{P}, \mathbb{Q}, \kappa) \in (\epsilon, 1]$.*

This theorem shows that, in both cases, the ratio $1/(\mathrm{NAMMD}(\mathbb{P}, \mathbb{Q}, \kappa) - \epsilon)^2$ is the main quantity dictating the upper bound of the sample complexity of our NAMMD test under alternative hypothesis $\boldsymbol{H}_1 : \mathrm{NAMMD}(\mathbb{P}, \mathbb{Q}, \kappa) > \epsilon$. This result is in accordance with the intuitive understanding.

### 4.2 COMPARISON WITH ORIGINAL MMD FOR TWO-SAMPLE TESTING

We recall that original MMD is applied to two-sample testing with null hypothesis $\boldsymbol{H}_0'' : \mathbb{P} = \mathbb{Q}$. By following Lemma, we present that our NAMMD can also be used to test whether $\mathbb{P} = \mathbb{Q}$.

**Lemma 9.** *We have $\mathrm{NAMMD}(\mathbb{P}, \mathbb{Q}, \kappa) = 0$ if and only if $\mathbb{P} = \mathbb{Q}$ for characteristic kernel $\kappa$.*

Hence, the two-sample testing can be formalized as distribution closeness testing in Definition 3 with null and alternative hypotheses: $\boldsymbol{H}_0 : \mathrm{NAMMD} = 0$ and $\boldsymbol{H}_1 : \mathrm{NAMMD} > 0$.

We present the empirical estimator of MMD as follows [26]

$$\widehat{\mathrm{MMD}}(X, Y, \kappa) = (m(m-1))^{-1}\sum_{i \neq j} \kappa(\boldsymbol{x}_i, \boldsymbol{x}_j) + \kappa(\boldsymbol{y}_i, \boldsymbol{y}_j) - \kappa(\boldsymbol{x}_i, \boldsymbol{y}_j) - \kappa(\boldsymbol{y}_i, \boldsymbol{x}_j).$$

Given this, we provide theoretical analysis of the advantages of our NAMMD for two-sample testing.

**Theorem 10.** *Under alternative hypothesis $\boldsymbol{H}_1 : \mathrm{NAMMD}(\mathbb{P}, \mathbb{Q}, \kappa) > 0$, i.e. $\mathbb{P} \neq \mathbb{Q}$, the following holds with probability at least $1 - \exp(-m\|\boldsymbol{\mu}_{\mathbb{P}} - \boldsymbol{\mu}_{\mathbb{Q}}\|_{\mathcal{H}_\kappa}^4/(16K^2))$ over sample $X$ and $Y$,*

$$m\widehat{\mathrm{MMD}}(X, Y, \kappa) > r_M \quad \Rightarrow \quad m\widehat{\mathrm{NAMMD}}(X, Y, \kappa) > r_N \ .$$

*Here, $r_M$ and $r_N$ are $(1 - \alpha)$-quantiles of asymptotic null distribution of $m\widehat{\mathrm{NAMMD}}$ and $m\widehat{\mathrm{MMD}}$. Furthermore, following holds with probability $\varsigma \geq 1/65$ over samples $X$ and $Y$,*

$$m\widehat{\mathrm{MMD}}(X, Y, \kappa) \leq r_M \quad yet \quad m\widehat{\mathrm{NAMMD}}(X, Y, \kappa) > r_N \ ,$$

*if $m \geq C'$, where $C'$ is dependent on distributions $\mathbb{P}$ and $\mathbb{Q}$, and probability $\varsigma$.*

This theorem shows that, under the same kernel, if MMD test rejects null hypothesis correctly, our NAMMD test also rejects null hypothesis with high probability. Furthermore, we present that our NAMMD test can correctly reject null hypothesis even in cases where the original MMD test fails to do so. For two-sample testing, NAMMD and MMD have the same test power estimator because, asymptotically, after we fixed two distributions $\mathbb{P}$ and $\mathbb{Q}$, NAMMD can be viewed as MMD scaled by a constant $4K - \|\boldsymbol{\mu}_{\mathbb{P}}\|_{\mathcal{H}_\kappa}^2 - \|\boldsymbol{\mu}_{\mathbb{Q}}\|_{\mathcal{H}_\kappa}^2$. Hence, NAMMD and MMD has the same optimal kernel based on the test power estimator (details are in Appendix B.10). Based on the optimal kernel, NAMMD also achieves better performance than MMD using the permutation test as stated above Theorem.

### 4.3 COMPARISON WITH ORIGINAL MMD FOR DISTRIBUTION CLOSENESS TESTING

Inspired by **Performing Distribution Closeness Testing in Practice** (Section 3), we provide a more structured definition to compare our NAMMD with original MMD in distribution closeness testing.

**Definition 11.** Given the known distributions $\mathbb{P}_1$ and $\mathbb{Q}_1$, and samples $X$ and $Y$ drawn from unknown distributions $\mathbb{P}_2$ and $\mathbb{Q}_2$, the goal of distribution closeness testing is to correctly determine whether the distance between $\mathbb{P}_2$ and $\mathbb{Q}_2$ is larger than that between $\mathbb{P}_1$ and $\mathbb{Q}_1$. To compare the test power, we perform our NAMMD test and original MMD test separately, under scenarios where the following null hypotheses are simultaneously false:

$$\boldsymbol{H}_0^N : \mathrm{NAMMD}(\mathbb{P}_2, \mathbb{Q}_2, \kappa) \leq \epsilon^N \quad \text{and} \quad \boldsymbol{H}_0^M : \mathrm{MMD}(\mathbb{P}_2, \mathbb{Q}_2, \kappa) \leq \epsilon^M \ ,$$

and following alternative hypotheses simultaneously hold true:

$$\boldsymbol{H}_1^N : \mathrm{NAMMD}(\mathbb{P}_2, \mathbb{Q}_2, \kappa) > \epsilon^N \quad \text{and} \quad \boldsymbol{H}_1^M : \mathrm{MMD}(\mathbb{P}_2, \mathbb{Q}_2, \kappa) > \epsilon^M \ ,$$

where $\epsilon^N = \mathrm{NAMMD}(\mathbb{P}_1, \mathbb{Q}_1, \kappa)$ and $\epsilon^M = \mathrm{MMD}(\mathbb{P}_1, \mathbb{Q}_1, \kappa)$.

Based on the definition, we present theoretical analysis of the advantages of our NAMMD test.

**Theorem 12.** *Under $\boldsymbol{H}_1^N : \mathrm{NAMMD}(\mathbb{Q}_2, \mathbb{P}_2, \kappa) > \epsilon^N$ and $\boldsymbol{H}_1^M : \mathrm{MMD}(\mathbb{Q}_2, \mathbb{P}_2, \kappa) > \epsilon^M$, and assuming $\|\boldsymbol{\mu}_{\mathbb{P}_1}\| + \|\boldsymbol{\mu}_{\mathbb{Q}_1}\| < \|\boldsymbol{\mu}_{\mathbb{P}_2}\| + \|\boldsymbol{\mu}_{\mathbb{Q}_2}\|$, then the following holds with probability at least $1 - \exp\left(-m\Delta^2(4K - \|\boldsymbol{\mu}_{\mathbb{P}_2}\|_{\mathcal{H}_\kappa}^2 - \|\boldsymbol{\mu}_{\mathbb{Q}_2}\|_{\mathcal{H}_\kappa}^2)^2/(4K^2(1 - \Delta)^2)\right)$,*

$$\sqrt{m}\widehat{\mathrm{MMD}}(X, Y, \kappa) > r'_M \quad \Rightarrow \quad \sqrt{m}\widehat{\mathrm{NAMMD}}(X, Y, \kappa) > r'_N \ ,$$

*where*

$$\Delta = \sqrt{m}\mathrm{NAMMD}(\mathbb{P}_1, \mathbb{Q}_1, \kappa)\frac{\|\boldsymbol{\mu}_{\mathbb{P}_2}\|_{\mathcal{H}_\kappa}^2 + \|\boldsymbol{\mu}_{\mathbb{Q}_2}\|_{\mathcal{H}_\kappa}^2 - \|\boldsymbol{\mu}_{\mathbb{P}_1}\|_{\mathcal{H}_\kappa}^2 - \|\boldsymbol{\mu}_{\mathbb{Q}_1}\|_{\mathcal{H}_\kappa}^2}{\sqrt{m}\mathrm{MMD}(\mathbb{P}_1, \mathbb{Q}_1, \kappa) + \sigma'_M \mathcal{N}_{1-\alpha}} \in (0, 1/2) \ .$$

*$r'_M$ and $r'_N$ are asymptotic $(1-\alpha)$-thresholds for $\sqrt{m}\widehat{\mathrm{MMD}}$ and $\sqrt{m}\widehat{\mathrm{NAMMD}}$ under null hypothesis. Furthermore, following holds with probability $\varsigma \geq 1/65$ over samples $X$ and $Y$,*

$$\sqrt{m}\widehat{\mathrm{MMD}}(X, Y, \kappa) \leq r'_M \quad yet \quad \sqrt{m}\widehat{\mathrm{NAMMD}}(X, Y, \kappa) > r'_N \ ,$$

*if $m \geq C''$, where $C''$ is dependent on distributions $\mathbb{P}$ and $\mathbb{Q}$, and probability $\varsigma$.*

Similarly, our NAMMD test is proven with higher test power than MMD test in distribution closeness testing, given that both alternative hypotheses, $\boldsymbol{H}_1^N$ and $\boldsymbol{H}_1^M$, hold true. Notably, the condition $\|\boldsymbol{\mu}_{\mathbb{P}_1}\| + \|\boldsymbol{\mu}_{\mathbb{Q}_1}\| < \|\boldsymbol{\mu}_{\mathbb{P}_2}\| + \|\boldsymbol{\mu}_{\mathbb{Q}_2}\|$ is often met in practice as norms of mean embeddings are typically positively correlated with MMD value. The improvement analysis is conducted using the same kernel for both NAMMD and MMD. Given this, we can derive an conjunction showing that NAMMD test with its (unknown) optimal global kernel $\kappa_*^N$ also achieves improvements over MMD test with its (unknown) optimal global kernel $\kappa_*^M$. The key insight is that, for the optimal kernel of MMD $\kappa_*^M$, NAMMD test with $\kappa_*^M$ performs better than MMD test with $\kappa_*^M$ (details are in Appendix B.10).

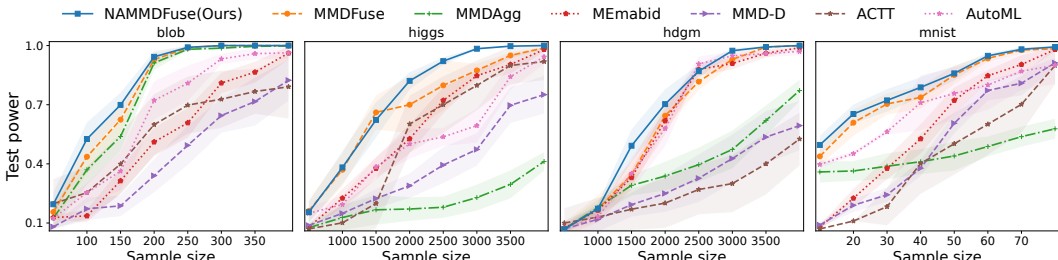

**Figure 2:** The comparisons of test power vs sample size for our NAMMDFuse and SOTA two-sample tests.

**Table 1:** Comparisons of test power (mean±std) on two-sample testing with the same kernel, and the bold denotes the highest mean between our NAMMD test and the original MMD test.

| Dataset | Gaus. Kernel | | Maha. Kernel | | Deep Kernel | | Lapl. Kernel | |
|---|---|---|---|---|---|---|---|---|
| | MMD | NAMMD | MMD | NAMMD | MMD | NAMMD | MMD | NAMMD |
| blob | .600±.090 | **.616**±**.090** | **1.00**±**.000** | **1.00**±**.000** | .859±.084 | **.863**±**.083** | .359±.088 | **.364**±**.088** |
| higgs | .563±.073 | **.566**±**.075** | .904±.087 | **.905**±**.086** | .796±.091 | **.797**±**.091** | .556±.062 | **.581**±**.062** |
| hdgm | .707±.042 | **.713**±**.041** | .801±.097 | **.805**±**.095** | .332±.087 | **.334**±**.086** | .090±.012 | **.100**±**.013** |
| mnist | .405±.019 | **.411**±**.020** | .970±.013 | **.975**±**.012** | .462±.100 | **.467**±**.098** | .873±.016 | **.881**±**.010** |
| cifar10 | .219±.017 | **.222**±**.020** | .984±.007 | **.987**±**.006** | .997±.003 | **1.00**±**.000** | .998±.002 | **1.00**±**.000** |
| Average | .499±.048 | **.506**±**.049** | .932±.041 | **.934**±**.040** | .689±.073 | **.692**±**.072** | .575±.036 | **.585**±**.035** |

## 5 EXPERIMENTS

We first conduct experiments on five benchmark datasets that have been studied in previous hypothesis testing approaches [27, 30]. Specifically, "blob" and "hdgm" are synthetic datasets based on Gaussain mixtures with dimensions 2 and 10, respectively. For "higgs", we compare the 4 dimension $\phi$-momenta distribution of Higgs-producing processes to background processes. "mnist" and "cifar" are image datasets consisting of original and generative images. Additionally, we perform distribution closeness testing on practical tasks related to domain adaptation using ImageNet and its variants. Notably, in all experiments, we use the selected characteristic kernels of the form $\kappa(\boldsymbol{x}, \boldsymbol{x}') = \Psi(\boldsymbol{x} - \boldsymbol{x}') \in (0, K]$ with $\Psi(\boldsymbol{0}) = K$, including Gaussian, Laplace, Mahalanobis and Deep kernels.

### 5.1 TWO-SAMPLE TESTING EXPERIMENTS

We begin by extending our NAMMD to the NAMMDFuse (Appendix C.3) by simply replacing original MMD distance with our NAMMD distance in the fusing statistics approach [33]. We compare our NAMMDFuse with state-of-the-art (SOTA) two-sample testings (Appendix D.3): 1). MMDFuse [33]; 2). MMD-D [27]; 3). MMDAgg [32]; 4). AutoTST [55]; 5). ME$_{\text{MaBiD}}$ [30]; 6). ACTT [66]. We follow parameter settings for these methods as their respective inferences. The ratio is set to $1:1$ for training and test sample sizes. We repeat such process 10 times for each dataset. Note that we set the test sample size for NAMMDFuse, MMDFuse, MMDAgg, and ACTT to be twice that of other methods, as these methods do not require training for kernel selection. For our NAMMDFuse, the null hypothesis is NAMMDFuse($\mathbb{P}, \mathbb{Q}, \kappa$) = 0, and we apply permutation test in two-sample testing.

From Figure 2, it is observed that our NAMMDFuse achieves test power that is either higher or comparable to other methods. In comparison with MMD-D, AutoTST and ME$_{\text{MaBiD}}$, our method utilizes all available samples for testing without training procedure. Compared to MMDAgg and ACTT, the fusion of our NAMMD distance use a log-sum-exp soft maximum, which incorporates information from multiple kernels simultaneously [33]. It is also evident that our method takes better performance than MMDFuse by scaling MMD distance with norms of mean embeddings.

For further comparison, we evaluate our NAMMD test (with $\epsilon = 0$) against the MMD test in terms of test power with the same kernel. We perform this experiments across four frequently used kernels (Appendix D.4): 1). Gaussian kernel [67]; 2). Laplace kernel [33]; 3). Deep kernel [27]; 4). Mahalanobis kernel [30]. Following [30, 27], we learn kernels on a subset of each available dataset for 2000 epochs, and then test on 100 random same size subsets from remaining dataset. The ratio is set to $1:1$ for training and test sample sizes. We repeat such process 10 times for each dataset. For our NAMMD test, the null hypothesis is NAMMD($\mathbb{P}, \mathbb{Q}, \kappa$) = 0, and we apply permutation test.

**Table 2:** Comparisons of test power (mean±std) on distribution closeness testing with respect to different total variation values, and the bold denotes the highest mean between our NAMMD test and Canonne's test.

| Dataset | $\epsilon' = 0.1$ | | $\epsilon' = 0.3$ | | $\epsilon' = 0.5$ | | $\epsilon' = 0.7$ | |
| --- | --- | --- | --- | --- | --- | --- | --- | --- |
| | Canonne's | NAMMD | Canonne's | NAMMD | Canonne's | NAMMD | Canonne's | NAMMD |
| blob | .856±.023 | **.968**±**.022** | .809±.014 | **.912**±**.053** | .944±.013 | **.960**±**.020** | **.998**±**.002** | .961±.029 |
| higgs | .883±.015 | **.908**±**.050** | .825±.010 | **.947**±**.027** | .960±.005 | **.962**±**.023** | .994±.003 | **.995**±**.005** |
| hdgm | .861±.011 | **.942**±**.023** | .888±.016 | **.946**±**.017** | .937±.014 | **.965**±**.014** | .987±.004 | **.989**±**.004** |
| mnist | .715±.021 | **.931**±**.024** | .786±.026 | **.965**±**.007** | .896±.013 | **.997**±**.001** | .971±.008 | **1.00**±**.000** |
| cifar10 | .686±.030 | **.919**±**.017** | .751±.021 | **.923**±**.021** | .917±.006 | **.997**±**.002** | .981±.004 | **.999**±**.001** |
| Average | .800±.020 | **.934**±**.027** | .812±.017 | **.939**±**.025** | .931±.010 | **.976**±**.012** | .986±.004 | **.989**±**.008** |

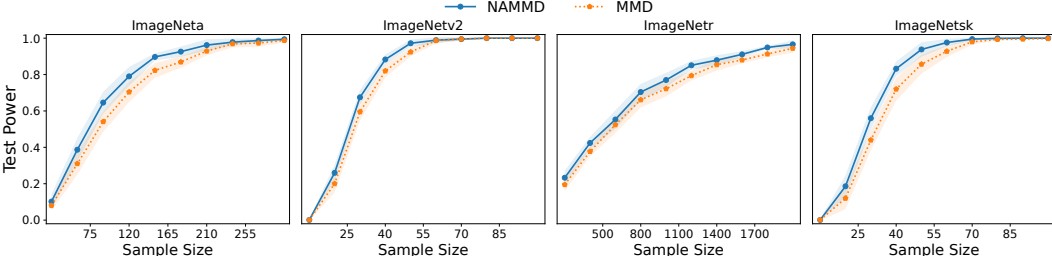

**Figure 3:** Comparisons in distinguishing the closeness levels between the original and variants of ImageNet.

Table 1 summarizes the average of test powers and standard deviations of our NAMMD test and the MMD test with the same kernel. It is evident that our NAMMD test achieves better performance than original MMD test as for Gaussian, Laplace, Mahalanobis and Deep kernels. It is because scaling maximum mean discrepancy with the norms of mean embeddings improves the effectiveness of our NAMMD test in two-sample testing, and this is nicely in accordance with Theorem 10.

## 5.2 DISTRIBUTION CLOSENESS TESTING EXPERIMENTS

Here, we first compare the test power of distribution closeness tests using our NAMMD and the statistic based on total variation introduced by Canonne et al. [37], and the experiments are performed on discrete distributions with a support set containing only finite elements. For each datasets, we draw 50 elements $Z = \{z_1, z_2, ..., z_{50}\}$, and denote by $\mathbb{P}_{50}$ the uniform distribution over domain $Z$. Starting with the uniform distribution, we increase the probability of randomly selected 25 elements and decrease the probabilities of remaining 25 elements uniformly to construct distribution $\mathbb{Q}_{50}$, which satisfies $\text{TV}(\mathbb{P}_{50}, \mathbb{Q}_{50}) = \epsilon'$ and is used for null hypothesis. We similarly construct distribution $\mathbb{Q}_{50}^A$ with $\text{TV}(\mathbb{P}_{50}, \mathbb{Q}_{50}^A) = \epsilon' + 0.2$ for alternative hypothesis.

Then, the corresponding null hypothesis for our NAMMD test is $\boldsymbol{H}_0 : \text{NAMMD}(\mathbb{P}, \mathbb{Q}, \kappa) \leq \epsilon$ with the selected Mahalanobis kernel. In experiments, we randomly draw two samples from $\mathbb{P}_{50}$ and $\mathbb{Q}_{50}^A$ to evaluate the test power. Table 2 summarizes the average test powers and standard deviations of our NAMMD test and Canonne's test, which measures the difference in the occurrences of each element between the two samples (i.e., the estimated distance for total variation). For comparison, we set $\epsilon' \in \{0.1, 0.3, 0.5, 0.7\}$. The threshold for Canonne's test is determined by resampling the estimated distance from distributions $\mathbb{P}_{50}$ and $\mathbb{Q}_{50}$. Further details are provided in the Appendix D.1.

From Table 2, it is evident that our NAMMD for distribution closeness testing achieves better performances than Canonne's test, due to the inherent difficulty in making accurate estimates based on occurrences, particularly when data is limited. On the other hand, kernel trick in our NAMMD can effectively capture intrinsic structures and complex patterns in real-word datasets. For 2-dimensional dataset blob, the statistic of Canonne's test exhibits smaller variance at $\epsilon' = 0.7$ and preserves much of the structural information from data, thus leading to higher test power.

**Performing Distribution Closeness Testing in Practice.** We present three case studies demonstrating the application of our NAMMD distribution closeness testing to evaluate whether a model performs similarly across training and testing datasets. First, given the pre-trained ResNet50, which performs well on the original ImageNet dataset, we wish to evaluate its performance on variants of ImageNet. A natural metric is the accuracy margin, defined as the difference in accuracy between the ImageNet and its variant, where a smaller margin indicates more comparable performance. For variants

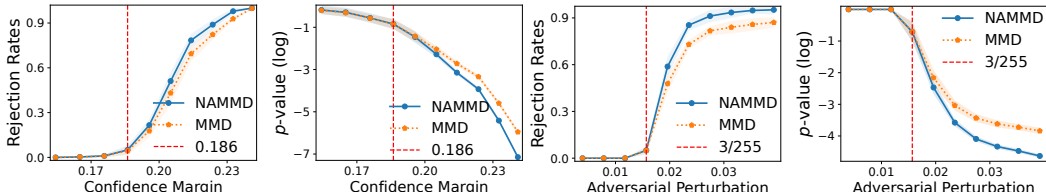

**Figure 4:** Performing distribution closeness testing to detect the confidence margin for domain adaptation between ImageNet and ImageNetv2 datasets.

**Figure 5:** Comparisons between our NAMMD and the original MMD for distribution closeness testing in adversarial perturbation detection (cifar10 dataset).

{ImageNetsk, ImageNetr, ImageNetv2, ImageNeta}, we can compute their accuracy margins as {0.529, 0.564, 0.751, 0.827} with ground truth labels, reflecting their relative similarity to ImageNet.

However, obtaining ground truth labels for variant ImageNet datasets is often challenging or expensive. In such cases, we demonstrate that model performance can be assessed using our NAMMD closeness testing without labels. The key is to validate that our NAMMD distance reflects the same closeness relationships as the accuracy margin, and it performs effectively in distribution closeness testing. Following Definition 11, we set ImageNet as $\mathbb{P}_1$ and $\mathbb{P}_2$, and sequentially set each of its variants (ImageNeta, ImageNetv2, ImageNetr, and ImageNetsk) as $\mathbb{Q}_2$. We further sequentially set each of the variants (ImageNetv2, ImageNetr, ImageNetsk, slightly perturbed ImageNet) as $\mathbb{Q}_1$, and performs testing to assess if the distance between $\mathbb{P}_2$ and $\mathbb{Q}_2$ is larger than that between $\mathbb{P}_1$ and $\mathbb{Q}_1$. Figure 3 shows that our NAMMD achieves higher test power than MMD by incorporating norms of distributions, and effectively reflects the closeness relationships indicated by accuracy margin. Moreover, even with a limited sample size (significantly smaller than that of ImageNet or its variants), our NAMMD distance can successfully identify the closeness relationships.

For datasets with limited samples, accuracy margin may be dispersed and fail to reliably capture differences in model performance. We introduce the confidence margin (Eqn. 12 in Appendix D.5) between two datasets, where a smaller margin also indicate similar model performance. We also validate that our NAMMD reflects the same closeness relationships as confidence margin. We use pre-trained ResNet50 model to compute confidence margin for each class individually between ImageNet and ImageNetv2. Following Definition 11, we define the classes with average margin 0.186 in ImageNet and ImageNetv2 as $\mathbb{P}_1$ and $\mathbb{Q}_1$. We further set $\mathbb{P}_2$ and $\mathbb{Q}_2$ as the classes in ImageNet and ImageNetv2 with margins in {0.154, 0.165, 0.176, 0.186, 0.196, 0.205, 0.214, 0.224, 0.233, 0.241}. We perform testing with sample size 150 and present the rejection rates and $p$-values of NAMMD and MMD are presented in Figure 4. For margins up to 0.186 (left side of red line), rejection rates (type-I errors) are limited given $\alpha = 0.05$. Conversely, for margins exceed 0.186 (right side of red line), our NAMMD achieves higher rejection rates (test powers) and lower $p$-values by incorporating norms.

Similarly, we validate that our NAMMD can be used to assess the level of adversarial perturbation over the cifar10 dataset. Using ResNet18 as the base model, we apply the PGD attack with perturbations $\{i/255\}_{i=1}^{[10]}$. As expected, a larger perturbation generally result in poor model performance on the perturbed cifar10 dataset, indicating that the perturbed cifar10 is farther from the original cifar10. Following Definition 11, we define the original cifar10 as $\mathbb{P}_1 = \mathbb{P}_2$ and the cifar10 dataset with $4/255$ perturbation as $\mathbb{Q}_1$. We further set $\mathbb{Q}_2$ as the cifar10 dataset after applying perturbations $\{i/255\}_{i=1}^{[10]}$, and perform testing with sample size 1500. It is evident that our NAMMD performs better than MMD and effectively assesses the levels of adversarial perturbations, as shown in Figure 5.

For each experiment using a deep neural network, we use the corresponding deep kernel with selected bandwidth following two-sample testing approach [27]. More experiments, including **Type-I Error Experiments** for both distribution closeness and two-sample testings, can be found in Appendix D.6.

## 6 CONCLUSION

This work introduces a new kernel distribution closeness testing by proposing the *norm-adaptive MMD* (NAMMD) distance, which leverages the insight that we separate two distributions more effectively at the same MMD distance with larger norms of distributions. An intriguing future research direction is to selecting an optimal global kernel for distribution closeness testing. We provide the **Ethics Statement** in Appendix A and the **Limitation Statement** in Appendix C.4.

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

## A  ETHICS STATEMENT

We confirm that this study adheres to the ICLR Code of Ethics. This research does not involve human subjects, and all datasets used are publicly available, ensuring compliance with privacy and security regulations. We have taken necessary precautions to avoid any potentially harmful insights or applications that may arise from our methodologies. Additionally, there are no potential conflicts of interest or sponsorships that could bias the outcomes of this work. This research complies with all relevant legal, ethical, and research integrity guidelines.

## B  DETAILED PROOFS AND ADDITIONAL DISCUSSIONS OF OUR THEORETICAL RESULTS

To begin, we define the concept of the U-statistic, which is a key statistical tool.

**Definition 13.** [41] Let $h(\boldsymbol{x}_1, \boldsymbol{x}_2, \ldots, \boldsymbol{x}_r)$ be a symmetric function of $r$ arguments. Suppose we have a random sample $\boldsymbol{x}_1, \boldsymbol{x}_2, \ldots, \boldsymbol{x}_m$ from some distribution. The U-statistic is given by:

$$U_m = \binom{m}{r}^{-1} \sum_{1 \le i_1 < i_2 < \cdots < i_r \le m} h(\boldsymbol{x}_{i_1}, \boldsymbol{x}_{i_2}, ..., \boldsymbol{x}_{i_r}) \, .$$

Here, $\binom{m}{r}$ is the number of ways to choose $r$ distinct indices from $m$, i.e., the binomial coefficient, and the summation is taken over all possible $r$-tuples from the sample.

We further present the large deviation for U-statistic as follows.

**Theorem 14.** [68] If the function $h$ is bounded, $a \le h(\boldsymbol{x}_{i_1}, \boldsymbol{x}_{i_2}, ..., \boldsymbol{x}_{i_r}) \le b$, we have

$$\Pr(|U_m - \theta| \ge t) \le 2 \exp\left(-2\lfloor m/r \rfloor t^2/(b-a)^2\right) ,$$

where $\theta = E[h(\boldsymbol{x}_{i_1}, \boldsymbol{x}_{i_2}, ..., \boldsymbol{x}_{i_r})]$.

### B.1  DETAILED PROOFS OF LEMMA 9

We begin with a useful theorem as follows.

**Theorem 15.** [26] Denote by $\mathbb{P}$ and $\mathbb{Q}$ two Borel probability measures over space $\mathcal{X} \subseteq \mathbb{R}^d$. Let $\kappa : \mathcal{X} \times \mathcal{X} \to \mathbb{R}$ be a characteristic kernel. Then $\mathrm{MMD}^2(\mathbb{P}, \mathbb{Q}, \kappa) = 0$ if and only if $\mathbb{P} = \mathbb{Q}$.

We now present the proofs of Lemma 9 as follows.

*Proof.* Recall that $0 \le \kappa(\boldsymbol{x}, \boldsymbol{y}) \le K$ and

$$\mathrm{NAMMD}(\mathbb{P}, \mathbb{Q}, \kappa) = \frac{\|\boldsymbol{\mu}_{\mathbb{P}} - \boldsymbol{\mu}_{\mathbb{Q}}\|_{\mathcal{H}_\kappa}^2}{4K - \|\boldsymbol{\mu}_{\mathbb{P}}\|_{\mathcal{H}_\kappa}^2 - \|\boldsymbol{\mu}_{\mathbb{Q}}\|_{\mathcal{H}_\kappa}^2} = \frac{\mathrm{MMD}^2(\mathbb{P}, \mathbb{Q}, \kappa)}{4K - \|\boldsymbol{\mu}_{\mathbb{P}}\|_{\mathcal{H}_\kappa}^2 - \|\boldsymbol{\mu}_{\mathbb{Q}}\|_{\mathcal{H}_\kappa}^2} \, .$$

It is evident that $4K - \|\boldsymbol{\mu}_{\mathbb{P}}\|_{\mathcal{H}_\kappa}^2 - \|\boldsymbol{\mu}_{\mathbb{Q}}\|_{\mathcal{H}_\kappa}^2 > 0$. Consequently, $\mathrm{NAMMD}(\mathbb{P}, \mathbb{Q}, \kappa) = 0$ if and only if $\mathbb{P} = \mathbb{Q}$ for characteristic kernels. This completes the proof. $\qquad\square$

### B.2  DETAILED PROOFS OF THEOREM 2

We begin with the empirical estimator of MMD as

$$\widehat{\mathrm{MMD}}^2(X, Y, \kappa) = 1/(m(m-1)) \sum_{i \ne j} \kappa(\boldsymbol{x}_i, \boldsymbol{x}_j) + \kappa(\boldsymbol{y}_i, \boldsymbol{y}_j) - \kappa(\boldsymbol{x}_i, \boldsymbol{y}_j) - \kappa(\boldsymbol{y}_i, \boldsymbol{x}_j) \, .$$

Given this, we introduce a useful theorem as follows.

**Theorem 16.** Under the null hypothesis $\boldsymbol{H}_0'' : \mathbb{P} = \mathbb{Q}$, let $Z_i \sim \mathcal{N}(0, 2)$ and we have

$$m\widehat{\mathrm{MMD}}^2(X, Y, \kappa) \xrightarrow{d} \sum_i \lambda_i \left(Z_i^2 - 2\right) ;$$

*here $\lambda_i$ are the eigenvalues of the $\mathbb{P}$-covariance operator of the centered kernel [26, Theorem 12], and $\xrightarrow{d}$ denotes convergence in distribution. On the other hand, under the alternative $\boldsymbol{H}_1'' : \mathbb{P} \neq \mathbb{Q}$, a standard central limit theorem holds [41, Section 5.5.1]*

$$\sqrt{m}\left(\widehat{\mathrm{MMD}}^2(X,Y,\kappa) - \mathrm{MMD}^2(\mathbb{P},\mathbb{Q},\kappa)\right) \xrightarrow{d} \mathcal{N}\left(0, \sigma_M^2\right) ,$$

$$\sigma_M^2 := 4E[H_{1,2}H_{1,3}] - 4(E[H_{1,2}])^2 ,$$

*where $H_{i,j} = \kappa(\boldsymbol{x}_i, \boldsymbol{x}_j) + \kappa(\boldsymbol{y}_i, \boldsymbol{y}_j) - \kappa(\boldsymbol{x}_i, \boldsymbol{y}_j) - \kappa(\boldsymbol{y}_i, \boldsymbol{x}_j)$ and the expectation are taken with respect to $\boldsymbol{x}_1, \boldsymbol{x}_2, \boldsymbol{x}_3 \sim \mathbb{P}^3$ and $\boldsymbol{y}_1, \boldsymbol{y}_2, \boldsymbol{y}_3 \sim \mathbb{Q}^3$.*

We now present the proofs of Theorem 2 as follows.

*Proof.* Recall the empirical estimator of our NAMMD distance

$$
\begin{aligned}
m\widehat{\mathrm{NAMMD}}(X,Y,\kappa) &= \frac{\sum_{i\neq j} \kappa(\boldsymbol{x}_i,\boldsymbol{x}_j) + \kappa(\boldsymbol{y}_i,\boldsymbol{y}_j) - \kappa(\boldsymbol{x}_i,\boldsymbol{y}_j) - \kappa(\boldsymbol{y}_i,\boldsymbol{x}_j)}{\sum_{i\neq j} 4K - \kappa(\boldsymbol{x}_i,\boldsymbol{x}_j) - \kappa(\boldsymbol{y}_i,\boldsymbol{y}_j)} \\
&= \frac{m\widehat{\mathrm{MMD}}^2(X,Y,\kappa)}{1/(m^2-m)\sum_{i\neq j} 4K - \kappa(\boldsymbol{x}_i,\boldsymbol{x}_j) - \kappa(\boldsymbol{y}_i,\boldsymbol{y}_j)} .
\end{aligned}
$$

As a U-statistic, it is easy to see that

$$1/(m(m-1))\sum_{i\neq j} 4K - \kappa(\boldsymbol{x}_i,\boldsymbol{x}_j) - \kappa(\boldsymbol{y}_i,\boldsymbol{y}_j) \xrightarrow{p} 4K - \|\boldsymbol{\mu}_{\mathbb{P}}\|_{\mathcal{H}_\kappa}^2 - \|\boldsymbol{\mu}_{\mathbb{Q}}\|_{\mathcal{H}_\kappa}^2 ,$$

where $\xrightarrow{p}$ denotes convergence in probability.

If $\mathrm{NAMMD}(\mathbb{P},\mathbb{Q},\kappa) = 0$, we have $\mathbb{P} = \mathbb{Q}$ from Lemma 9, and

$$m\widehat{\mathrm{MMD}}^2(X,Y,\kappa) \xrightarrow{d} \sum_i \lambda_i\left(Z_i^2 - 2\right) ,$$

from Theorem 16. Then, by slutsky's theorem [69], we have

$$
\begin{aligned}
m\widehat{\mathrm{NAMMD}}(X,Y,\kappa) &\xrightarrow{d} \frac{\sum_i \lambda_i\left(Z_i^2 - 2\right)}{4K - \|\boldsymbol{\mu}_{\mathbb{P}}\|_{\mathcal{H}_\kappa}^2 - \|\boldsymbol{\mu}_{\mathbb{Q}}\|_{\mathcal{H}_\kappa}^2} \\
&\xrightarrow{d} \frac{\sum_i \lambda_i\left(Z_i^2 - 2\right)}{4K - \|(\boldsymbol{\mu}_{\mathbb{P}} + \boldsymbol{\mu}_{\mathbb{Q}})/\sqrt{2}\|_{\mathcal{H}_\kappa}^2} ,
\end{aligned}
$$

where $\boldsymbol{\mu}_{\mathbb{P}} = \boldsymbol{\mu}_{\mathbb{Q}} = (\boldsymbol{\mu}_{\mathbb{P}} + \boldsymbol{\mu}_{\mathbb{Q}})/2$.

If $\mathrm{NAMMD}(\mathbb{P},\mathbb{Q},\kappa) = \epsilon$ with $\epsilon \in (0,1)$, we present the asymptotic distribution of the empirical estimator in a similar manner, which can be formalized as

$$\sqrt{m}(\widehat{\mathrm{NAMMD}}(X,Y,\kappa) - \epsilon) \xrightarrow{d} \mathcal{N}\left(0, \frac{4E[H_{1,2}H_{1,3}] - 4(E[H_{1,2}])^2}{(4K - \|\boldsymbol{\mu}_{\mathbb{P}}\|_{\mathcal{H}_\kappa}^2 - \|\boldsymbol{\mu}_{\mathbb{Q}}\|_{\mathcal{H}_\kappa}^2)^2}\right) ,$$

which completes the proof.

$\square$

## B.3 DETAILED PROOFS OF LEMMA 4

We present the proofs of Lemma 4 as follows.

*Proof.* For simplicity, we let

$$\hat{A} = \sqrt{((4m-8)\zeta_1 + 2\zeta_2)/(m-1)} \quad \text{and} \quad A = \sqrt{4E[H_{1,2}H_{1,3}] - 4(E[H_{1,2}])^2} \,,$$

and

$$\hat{B} = (m^2 - m)^{-1} \sum_{i \neq j} 4K - \kappa(\mathbf{x}_i, \mathbf{x}_j) - \kappa(\mathbf{y}_i, \mathbf{y}_j) \quad \text{and} \quad B = 4K - \|\boldsymbol{\mu}_{\mathbb{P}}\|_{\mathcal{H}_\kappa}^2 - \|\boldsymbol{\mu}_{\mathbb{Q}}\|_{\mathcal{H}_\kappa}^2 \,.$$

Build on these results, we can bound the bias as follows:

$$
\begin{aligned}
\left| E[\sigma_{X,Y}^2] - \sigma_{\mathbb{P},\mathbb{Q}}^2 \right| = \left| E\left[\frac{\hat{A}^2}{\hat{B}^2}\right] - \frac{A^2}{B^2} \right| &= \left| E\left[\frac{\hat{A}^2}{\hat{B}^2}\right] - E\left[\frac{\hat{A}^2}{B^2}\right] + E\left[\frac{\hat{A}^2}{B^2}\right] - \frac{A^2}{B^2} \right| \\
&= \left| E\left[\frac{\hat{A}^2}{\hat{B}^2}\right] - E\left[\frac{\hat{A}^2}{B^2}\right] \right| \\
&\leq E\left[ \left| \frac{\hat{A}^2}{\hat{B}^2} - \frac{\hat{A}^2}{B^2} \right| \right] \\
&= E\left[ \left| \frac{\hat{A}^2(B - \hat{B})(B + \hat{B})}{\hat{B}^2 B^2} \right| \right] \\
&\leq C * E\left[ \left| B - \hat{B} \right| \right]
\end{aligned}
$$

where $C > 0$ is a constant that ensures $\frac{\hat{A}^2(B+\hat{B})}{\hat{B}^2 B^2} \leq C$, and it exists since the kernel is bounded. The second equation is based on the unbiased variance estimator of the U-statistic, i.e. $\hat{A}$. Based on the large deviation bound for $B$, we have

$$\Pr\left( \left| B - \hat{B} \right| \geq t \right) \leq 2\exp\left(-mt^2/4K^2\right)$$

and

$$
\begin{aligned}
C * E\left[ \left| B - \hat{B} \right| \right] = C * \int_0^\infty \Pr\left( \left| B - \hat{B} \right| \geq t \right) dt &\leq C * \int_0^\infty 2\exp\left(-mt^2/4K^2\right) dt \\
&= C * \int_0^\infty 2\exp\left(-u\right) \frac{K}{\sqrt{m}\sqrt{u}} du \\
&= C * \frac{2K\sqrt{\pi}}{\sqrt{m}} = O\left(\frac{1}{\sqrt{m}}\right) \,.
\end{aligned}
$$

This completes the proof. $\square$

### B.4 DETAILED PROOFS OF THEOREM 5

We begin with a useful definition as follows.

**Definition 17.** [63] Let $Z$ be the sample taking values in the instance space $\mathcal{X}$. Let $\mathcal{G}$ be a finite set of transformations $g : \mathcal{X} \to \mathcal{X}$, such that $\mathcal{G}$ is a group with respect to the operation of composition of transformations. Let $\mathcal{H}_0$ be any null hypothesis which implies that the joint distribution of the test statistics $T(gZ)$, $g \in \mathcal{G}$, is invariant under all transformations in $\mathcal{G}$ of $Z$. Denote by $B$ the cardinality of the set $\mathcal{G}$ and write $\mathcal{G} = \{g_1, ..., g_B\}$. We have, under $\mathcal{H}_0$,

$$(T(g_1 Z), ..., T(g_B Z)) \overset{d}{=} (T(g \cdot g_1 Z), ..., T(g \cdot g_B Z)) \quad \text{for all } g \in \mathcal{G} \,,$$

where $\overset{d}{=}$ denotes equality in distribution.

We now present the proofs of Theorem 5 as follows.

*Proof.* Under null hypothesis $\boldsymbol{H}_0 : \mathrm{NAMMD}(\mathbb{P}, \mathbb{Q}, \kappa) = 0$, we have $\mathbb{P} = \mathbb{Q}$ from Lemma 9. Let $Z = \{X, Y\}$ and $\boldsymbol{\Pi}_{2m}$ be the set of all possible permutations of $\{1, \ldots, 2m\}$ over the pooled sample $Z = \{\boldsymbol{x}_1, \ldots, \boldsymbol{x}_m, \boldsymbol{y}_1, \ldots, \boldsymbol{y}_m\} = \{\boldsymbol{z}_1, \ldots, \boldsymbol{z}_m, \boldsymbol{z}_{m+1}, \ldots, \boldsymbol{z}_{2m}\}$. Recall that we set the testing threshold as the $(1 - \alpha)$-quantile of estimated null distribution by permutation test as

$$\tau_\alpha(Z) = \arg\min_\tau \left\{ \sum_{b=1}^{B} \frac{\mathbb{I}[T_b(\boldsymbol{\pi} Z) \le \tau]}{B} \ge 1 - \alpha \right\},$$

with empirical estimator of permutation $\boldsymbol{\pi} \in \boldsymbol{\Pi}_{2m}$ of $b$-th iteration

$$T_b(\boldsymbol{\pi} Z) = \widehat{\mathrm{NAMMD}}(X_{\boldsymbol{\pi}}, Y_{\boldsymbol{\pi}}, \kappa) = \sum_{i \ne j} H_{\pi_i, \pi_j} / \sum_{i \ne j} (4K - \kappa(\boldsymbol{x}_{\pi_i}, \boldsymbol{x}_{\pi_j}) - \kappa(\boldsymbol{y}_{\pi_i}, \boldsymbol{y}_{\pi_j})),$$

where $X_{\boldsymbol{\pi}} = \{\boldsymbol{z}_{\pi_1}, \boldsymbol{z}_{\pi_2}, ..., \boldsymbol{z}_{\pi_m}\}$ and $Y_{\boldsymbol{\pi}} = \{\boldsymbol{z}_{\pi_{m+1}}, \boldsymbol{z}_{\pi_{m+2}}, ..., \boldsymbol{z}_{\pi_{2m}}\}$.

It is easy to see that $\boldsymbol{\Pi}_{2m}$ is a group with respect to operation of composition of transformations, and the null hypothesis $\boldsymbol{H}_0 : \mathrm{NAMMD}(\mathbb{P}, \mathbb{Q}, \kappa) = 0$, i.e., $\mathbb{P} = \mathbb{Q}$ implies the joint distribution of $T(\boldsymbol{\pi} Z)$ for $\boldsymbol{\pi} \in \boldsymbol{\Pi}_{2m}$, is invariant under all transformation.

By group structure, we have $\boldsymbol{\pi} \boldsymbol{\Pi}_{2m} = \boldsymbol{\Pi}_{2m}$ for all $\boldsymbol{\pi} \in \boldsymbol{\Pi}_{2m}$. Hence, we have $\boldsymbol{\pi} Z \stackrel{d}{=} Z$ and

$$\tau_\alpha(\boldsymbol{\pi} Z) = \tau_\alpha(Z).$$

Denote by $T(Z) = \widehat{\mathrm{NAMMD}}(X, Y, \kappa)$. Then, under null hypothesis $\boldsymbol{H}_0 : \mathrm{NAMMD}(\mathbb{P}, \mathbb{Q}, \kappa) = 0$, the reject probability is given by

$$
\begin{aligned}
\Pr(\, T(Z) > \tau_\alpha(Z)\,) &= E_{\boldsymbol{\pi} \sim \boldsymbol{\Pi}_{2m}}[\Pr(\, T(\boldsymbol{\pi} Z) > \tau_\alpha(\boldsymbol{\pi} Z)\,)] \\
&= E_{\boldsymbol{\pi} \sim \boldsymbol{\Pi}_{2m}}[\Pr(\, T(\boldsymbol{\pi} Z) > \tau_\alpha(Z)\,)] \\
&\le \alpha.
\end{aligned}
$$

The first equality holds since the null hypothesis implies the invariant joint distribution, and the second equality follows $\tau_\alpha(\boldsymbol{\pi} Z) = \tau_\alpha(Z)$. The final inequality follows from the definition of $\tau_\alpha(Z)$.

Under null hypothesis $\boldsymbol{H}_0 : \mathrm{NAMMD}(\mathbb{P}, \mathbb{Q}, \kappa) \le \epsilon$ with $\epsilon \in (0, 1)$, we set the testing threshold as the $(1 - \alpha)$-quantile of the asymptotic null distribution of $\mathrm{NAMMD}(\mathbb{P}, \mathbb{Q}, \kappa) = \epsilon$ from Theorem 2. Hence, the reject probability is also bounded by $\alpha$. This completes the proof.

$\square$

### B.5 DETAILED PROOFS OF LEMMA 6

*Proof.* Recall our NAMMD distance as follows:

$$\mathrm{NAMMD}(\mathbb{P}, \mathbb{Q}, \kappa) = \frac{\|\boldsymbol{\mu}_{\mathbb{P}} - \boldsymbol{\mu}_{\mathbb{Q}}\|_{\mathcal{H}_\kappa}^2}{4K - \|\boldsymbol{\mu}_{\mathbb{P}}\|_{\mathcal{H}_\kappa}^2 - \|\boldsymbol{\mu}_{\mathbb{Q}}\|_{\mathcal{H}_\kappa}^2} = \frac{\mathrm{MMD}^2(\mathbb{P}, \mathbb{Q}, \kappa)}{4K - \|\boldsymbol{\mu}_{\mathbb{P}}\|_{\mathcal{H}_\kappa}^2 - \|\boldsymbol{\mu}_{\mathbb{Q}}\|_{\mathcal{H}_\kappa}^2}.$$

Given two i.i.d. samples $X = \{\boldsymbol{x}_1, \boldsymbol{x}_2, ..., \boldsymbol{x}_m\} \sim \mathbb{P}^m$ and $Y = \{\boldsymbol{y}_1, \boldsymbol{y}_2, ..., \boldsymbol{y}_m\} \sim \mathbb{Q}^m$, we have the empirical estimator as follows

$$
\begin{aligned}
\widehat{\mathrm{NAMMD}}(X, Y, \kappa) &= \frac{\sum_{i \ne j} \kappa(\boldsymbol{x}_i, \boldsymbol{x}_j) + \kappa(\boldsymbol{y}_i, \boldsymbol{y}_j) - \kappa(\boldsymbol{x}_i, \boldsymbol{y}_j) - \kappa(\boldsymbol{y}_i, \boldsymbol{x}_j)}{\sum_{i \ne j} 4K - \kappa(\boldsymbol{x}_i, \boldsymbol{x}_j) - \kappa(\boldsymbol{y}_i, \boldsymbol{y}_j)} \\
&= \frac{\widehat{\mathrm{MMD}}^2(X, Y, \kappa)}{1/(m^2 - m) \sum_{i \ne j} 4K - \kappa(\boldsymbol{x}_i, \boldsymbol{x}_j) - \kappa(\boldsymbol{y}_i, \boldsymbol{y}_j)}.
\end{aligned}
$$

We denote by

$$
\begin{aligned}
A &= |\widehat{\mathrm{NAMMD}}(X, Y, \kappa) - \mathrm{NAMMD}(\mathbb{P}, \mathbb{Q}, \kappa)| \\
&= \left| \frac{\widehat{\mathrm{MMD}}^2(X, Y, \kappa) - \mathrm{MMD}^2(\mathbb{P}, \mathbb{Q}, \kappa) + \mathrm{MMD}^2(\mathbb{P}, \mathbb{Q}, \kappa)}{1/(m^2 - m) \sum_{i \ne j} 4K - \kappa(\boldsymbol{x}_i, \boldsymbol{x}_j) - \kappa(\boldsymbol{y}_i, \boldsymbol{y}_j)} - \frac{\mathrm{MMD}^2(\mathbb{P}, \mathbb{Q}, \kappa)}{4K - \|\boldsymbol{\mu}_{\mathbb{P}}\|_{\mathcal{H}_\kappa}^2 - \|\boldsymbol{\mu}_{\mathbb{Q}}\|_{\mathcal{H}_\kappa}^2} \right|.
\end{aligned}
$$

Given this, we let

$$B = \left| \frac{\widehat{\mathrm{MMD}}^2(X, Y, \kappa) - \mathrm{MMD}^2(\mathbb{P}, \mathbb{Q}, \kappa)}{1/(m^2 - m) \sum_{i \neq j} 4K - \kappa(\boldsymbol{x}_i, \boldsymbol{x}_j) - \kappa(\boldsymbol{y}_i, \boldsymbol{y}_j)} \right|,$$

and

$$C = \left| \frac{\mathrm{MMD}^2(\mathbb{P}, \mathbb{Q}, \kappa)}{1/(m^2 - m) \sum_{i \neq j} 4K - \kappa(\boldsymbol{x}_i, \boldsymbol{x}_j) - \kappa(\boldsymbol{y}_i, \boldsymbol{y}_j)} - \frac{\mathrm{MMD}^2(\mathbb{P}, \mathbb{Q}, \kappa)}{4K - \|\boldsymbol{\mu}_{\mathbb{P}}\|_{\mathcal{H}_\kappa}^2 - \|\boldsymbol{\mu}_{\mathbb{Q}}\|_{\mathcal{H}_\kappa}^2} \right|.$$

It is easy to see that $A \leq B + C$ and we have

$$\Pr(A \geq t) \leq \Pr(B + C \geq t) \leq \Pr(B \geq b) + \Pr(C \geq c),$$

for $b + c = t$ with $t > 0$ and $b, c \geq 0$.

Based on the large deviation bound for U-statistic (Theorem 14), we have

$$\Pr(B \geq b) \leq \Pr\left( \left| \widehat{\mathrm{MMD}}^2(X, Y, \kappa) - \mathrm{MMD}^2(\mathbb{P}, \mathbb{Q}, \kappa) \right| / 2K \geq b \right) \leq 2\exp\left( -mb^2/4 \right),$$

In a similar manner, we have

$\Pr(C \geq c)$

$$\leq \Pr\left( \frac{\mathrm{MMD}^2(\mathbb{P}, \mathbb{Q}, \kappa) | \sum_{i \neq j} (\kappa(\boldsymbol{x}_i, \boldsymbol{x}_j) + \kappa(\boldsymbol{y}_i, \boldsymbol{y}_j))/(m^2 - m)) - \|\boldsymbol{\mu}_{\mathbb{P}}\|_{\mathcal{H}_\kappa}^2 - \|\boldsymbol{\mu}_{\mathbb{Q}}\|_{\mathcal{H}_\kappa}^2 |}{(1/(m^2 - m) \sum_{i \neq j} 4K - \kappa(\boldsymbol{x}_i, \boldsymbol{x}_j) - \kappa(\boldsymbol{y}_i, \boldsymbol{y}_j)) \cdot (4K - \|\boldsymbol{\mu}_{\mathbb{P}}\|_{\mathcal{H}_\kappa}^2 - \|\boldsymbol{\mu}_{\mathbb{Q}}\|_{\mathcal{H}_\kappa}^2)} \geq c \right)$$

$$\leq \Pr\left( \left| \sum_{i \neq j} \frac{\kappa(\boldsymbol{x}_i, \boldsymbol{x}_j)}{m(m-1)} + \frac{\kappa(\boldsymbol{y}_i, \boldsymbol{y}_j)}{m(m-1)} - \|\boldsymbol{\mu}_{\mathbb{P}}\|_{\mathcal{H}_\kappa}^2 - \|\boldsymbol{\mu}_{\mathbb{Q}}\|_{\mathcal{H}_\kappa}^2 \right| \frac{\mathrm{MMD}^2(\mathbb{P}, \mathbb{Q}, \kappa)}{4K^2} \geq c \right)$$

$$\leq \Pr\left( \left| \sum_{i \neq j} \frac{\kappa(\boldsymbol{x}_i, \boldsymbol{x}_j)}{m(m-1)} + \frac{\kappa(\boldsymbol{y}_i, \boldsymbol{y}_j)}{m(m-1)} - \|\boldsymbol{\mu}_{\mathbb{P}}\|_{\mathcal{H}_\kappa}^2 - \|\boldsymbol{\mu}_{\mathbb{Q}}\|_{\mathcal{H}_\kappa}^2 \right| / 2K \geq c \right)$$

$$\leq 2\exp\left( -mc^2 \right)$$

For simplicity, let $b = 2t/3$ and $c = t/3$, we have

$$\begin{aligned} \Pr(A \geq t) &\leq \Pr(B \geq 2t/3) + \Pr(C \geq t/3) \\ &= 4\exp\left( -mt^2/9 \right). \end{aligned}$$

This completes the proof. $\qquad \square$

### B.6 DETAILED PROOFS OF LEMMA 7

Let $\boldsymbol{\Pi}_{2m}$ be the set of all possible permutations of $\{1, \ldots, 2m\}$ over the pooled sample $Z = \{\boldsymbol{x}_1, \ldots, \boldsymbol{x}_m, \boldsymbol{y}_1, \ldots, \boldsymbol{y}_m\} = \{\boldsymbol{z}_1, \ldots, \boldsymbol{z}_m, \boldsymbol{z}_{m+1}, \ldots, \boldsymbol{z}_{2m}\}$. Given a permutation $\boldsymbol{\pi} = (\pi_1, \ldots, \pi_{2m}) \in \boldsymbol{\Pi}_{2m}$, we have $X_{\boldsymbol{\pi}} = \{\boldsymbol{z}_{\pi_i}\}_{i=1}^m$ and $Y_{\boldsymbol{\pi}} = \{\boldsymbol{z}_{\pi_i}\}_{i=m+1}^{2m}$.

We begin with a useful Theorem as follows.

**Theorem 18.** *[65, Theorem 6.1] Consider the permuted two-sample U-statistic $U(X_{\boldsymbol{\pi}}, Y_{\boldsymbol{\pi}}, \kappa)$ with size $m$ for each sample and define*

$$\Sigma_m^2 := \frac{1}{m^2(m-1)^2} \sup_{\boldsymbol{\pi} \in \boldsymbol{\Pi}_{2m}} \left\{ \sum_{i \neq j}^m \kappa^2\left( \boldsymbol{z}_{\pi_i}, \boldsymbol{z}_{\pi_j} \right) \right\}.$$

*Then, for every $t > 0$ and some constant $C > 0$, we have*

$$\Pr\left( U(X_{\boldsymbol{\pi}}, Y_{\boldsymbol{\pi}}, \kappa) \geq t \right) \leq \exp\left( -C \min\left( \frac{t^2}{\Sigma_m^2}, \frac{t}{\Sigma_m} \right) \right).$$

We now present the proofs of Lemma 7 as follows.

*Proof.* Recall that

$$\widehat{\text{NAMMD}}(X_{\boldsymbol{\pi}}, Y_{\boldsymbol{\pi}}, \kappa) \;=\; \frac{\widehat{\text{MMD}}(X_{\boldsymbol{\pi}}, Y_{\boldsymbol{\pi}}, \kappa)}{1/(m^2 - m)\sum_{i \neq j} 4K - \kappa(\boldsymbol{x}_i, \boldsymbol{x}_j) - \kappa(\boldsymbol{y}_i, \boldsymbol{y}_j)} \;,$$

As we can see, $\widehat{\text{MMD}}(X_{\boldsymbol{\pi}}, Y_{\boldsymbol{\pi}}, \kappa)$ is a U-statistic. Hence, we have

$$\Pr\left(\widehat{\text{NAMMD}}(X_{\boldsymbol{\pi}}, Y_{\boldsymbol{\pi}}, \kappa) \geq t\right) \;\leq\; \Pr\left(\frac{\widehat{\text{MMD}}(X_{\boldsymbol{\pi}}, Y_{\boldsymbol{\pi}}, \kappa)}{2K} \geq t\right)$$

$$\leq\; \exp\left(-C\min\left(\frac{4K^2 t^2}{\Sigma_m^2}, \frac{2Kt}{\Sigma_m}\right)\right) \;.$$

This completes the proof. $\qquad\square$

## B.7 DETAILED PROOFS OF THEOREM 8

*Proof.* Under the alternative hypothesis $\boldsymbol{H}_1 : \text{NAMMD}(\mathbb{P}, \mathbb{Q}, \kappa) > 0$, we need to correctly reject the null hypothesis $\boldsymbol{H}_0 : \text{NAMMD}(\mathbb{P}, \mathbb{Q}, \kappa) = 0$. According to Eqn. 3, we set $\tau_\alpha$ as the $(1-\alpha)$-quantile of the estimated null distribution by permutation test.

By Lemma 7, it is easy to see that

$$\Pr\left(\widehat{\text{NAMMD}}(X_{\boldsymbol{\pi}}, Y_{\boldsymbol{\pi}}, \kappa) \geq t\right) \;\leq\; \exp\left(-C\min\left(\frac{4K^2 t^2}{\Sigma_m^2}, \frac{2Kt}{\Sigma_m}\right)\right)$$

$$\leq\; \exp\left(-C\min\left(\frac{4(m-1)^2 K^2 t^2}{K^2}, \frac{2(m-1)Kt}{K}\right)\right)$$

$$\leq\; \exp\left(-C\min\left(4(m-1)^2 t^2, 2(m-1)t\right)\right) \;,$$

the second inequality holds with $\Sigma_m^2 \leq (m(m-1))^{-1}K^2 \leq (m-1)^{-2}K^2$.

Let

$$\exp\left(-C\min\left(4(m-1)^2 t^2, 2(m-1)t\right)\right) \;=\; \alpha$$

$$\min\left(4(m-1)^2 t^2, 2(m-1)t\right) \;=\; \frac{\log \alpha^{-1}}{C} \;.$$

If $4(m-1)^2 t^2 \leq 2(m-1)t$, i.e., $t \leq (2(m-1))^{-1}$, and we have

$$t = \frac{1}{2(m-1)}\frac{\log \alpha^{-1}}{C},$$

which implies $\log \alpha^{-1}/C \leq 1$ and $\log \alpha^{-1}/C \leq \sqrt{\log \alpha^{-1}/C}$.

If $4(m-1)^2 t^2 > 2(m-1)t$, i.e., $t > (2(m-1))^{-1}$, and we have

$$t = \frac{1}{2(m-1)}\sqrt{\frac{\log \alpha^{-1}}{C}},$$

which implies $\log \alpha^{-1}/C > 1$ and $\log \alpha^{-1}/C > \sqrt{\log \alpha^{-1}/C}$.

In summary, we have

$$t = \frac{1}{2(m-1)}\min\left(\frac{\log \alpha^{-1}}{C}, \sqrt{\frac{\log \alpha^{-1}}{C}}\right)$$

For simplicity, let $C_\alpha = \min\left(\log \alpha^{-1}/C, \sqrt{\log \alpha^{-1}/C}\right)$, we have

$$t = \frac{1}{2(m-1)} * C_\alpha \;.$$

It is easy to see that the testing threshold $\tau_\alpha \leq t$.

By the large deviation bound for our NAMMD as shown in Lemma 6, we have

$$\Pr\left(\widehat{\text{NAMMD}}(X, Y, \kappa) - \text{NAMMD}(\mathbb{P}, \mathbb{Q}, \kappa) \geq -t\right) \leq 2\exp(-mt^2/9),$$

for $t > 0$.

To derive the upper bound, it follows with at least $1 - \upsilon$ probability, according to the large deviation bound discussed above,

$$\widehat{\text{NAMMD}}(X, Y, \kappa) \geq \text{NAMMD}(\mathbb{P}, \mathbb{Q}, \kappa) - \sqrt{\frac{9\log 2/\upsilon}{m}}.$$

To ensure the correct rejection of the null hypothesis , we have

$$\widehat{\text{NAMMD}}(X, Y, \kappa) > \frac{1}{2(m-1)} * C_\alpha$$

$$\text{NAMMD}(\mathbb{P}, \mathbb{Q}, \kappa) - \sqrt{\frac{9\log 2/\upsilon}{m}} > \frac{1}{2(m-1)} * C_\alpha,$$

which is equivalent to

$$(m-1)\text{NAMMD}(\mathbb{P}, \mathbb{Q}, \kappa) - (m-1)\sqrt{\frac{9\log 2/\upsilon}{m}} > \frac{m-1}{2(m-1)} * C_\alpha.$$

For the upper bound, we further scale as follows

$$(m-1)\text{NAMMD}(\mathbb{P}, \mathbb{Q}, \kappa) - \sqrt{9(m-1)\log 2/\upsilon} \geq \frac{C_\alpha}{2}.$$

We finally present the upper bound for sample complexity of our NAMMD test under the alternative hypothesis $\boldsymbol{H}_1 : \text{NAMMD}(\mathbb{P}, \mathbb{Q}, \kappa) > 0$ as follows

$$m \geq \frac{\left(\sqrt{9\log 2/\upsilon} + \sqrt{9\log 2/\upsilon + 2C_\alpha\text{NAMMD}(\mathbb{P}, \mathbb{Q}, \kappa)}\right)^2}{4 \cdot \text{NAMMD}^2(\mathbb{P}, \mathbb{Q}, \kappa)} + 1.$$

Under the alternative hypothesis $\boldsymbol{H}_1 : \text{NAMMD}(\mathbb{P}, \mathbb{Q}, \kappa) > 0$, we need to correctly reject the null hypothesis $\boldsymbol{H}_0 : \text{NAMMD}(\mathbb{P}, \mathbb{Q}, \kappa) = 0$. According to Eqn. 3, we set $\tau_\alpha$ as the $(1-\alpha)$-quantile of the estimated null distribution by permutation test.

Under the alternative hypothesis $\boldsymbol{H}_1 : \text{NAMMD}(\mathbb{P}, \mathbb{Q}, \kappa) > \epsilon$ with $\epsilon \in (0, 1)$, , we need to correctly reject the null hypothesis $\boldsymbol{H}_0 : \text{NAMMD}(\mathbb{P}, \mathbb{Q}, \kappa) \leq \epsilon$. According to Eqn. 3, we set $\tau_\alpha$ as the $(1-\alpha)$-quantile of the asymptotic null distribution of $\text{NAMMD}(\mathbb{P}, \mathbb{Q}, \kappa) = \epsilon$ from Theorem 2 as,

$$\tau_\alpha = \epsilon + \frac{\sigma_{X,Y}\mathcal{N}_{1-\alpha}}{\sqrt{m}},$$

where the empirical estimator of variance is given by

$$\sigma_{X,Y} = \frac{\sqrt{((4m-8)\zeta_1 + 2\zeta_2)/(m-1)}}{(m^2-m)^{-1}\sum_{i\neq j} 4K - \kappa(\boldsymbol{x}_i, \boldsymbol{x}_j) - \kappa(\boldsymbol{y}_i, \boldsymbol{y}_j)},$$

where $\zeta_1$ and $\zeta_2$ are standard variance components of the MMD [41, 42]. We present the details of the estimator in Appendix C.2.

It is easy to see that

$$(m^2-m)^{-1}\sum_{i\neq j} 4K - \kappa(\boldsymbol{x}_i, \boldsymbol{x}_j) - \kappa(\boldsymbol{y}_i, \boldsymbol{y}_j) \geq 2K \quad \text{and} \quad \zeta_1 \leq 4K^2 \quad \text{and} \quad \zeta_2 \leq 4K^2,$$

Hence, as we can see,

$$
\begin{aligned}
\sigma_{X,Y} &\leq \frac{\sqrt{(4m-6)/(m-1)4K^2}}{2K} \\
&\leq 4K/2K \\
&\leq 2 \, ,
\end{aligned}
$$

and we have

$$
\tau_\alpha \leq \epsilon + \frac{2\mathcal{N}_{1-\alpha}}{\sqrt{m}} \, .
$$

In a similar manner, to ensure the rejection, we have

$$
\widehat{\mathrm{NAMMD}}(X, Y, \kappa) > \epsilon + \frac{2\mathcal{N}_{1-\alpha}}{\sqrt{m}}.
$$

To derive the upper bound, the following holds with at least probability $1 - \upsilon$,

$$
\widehat{\mathrm{NAMMD}}(X, Y, \kappa) \geq \mathrm{NAMMD}(\mathbb{P}, \mathbb{Q}, \kappa) - \sqrt{\frac{9\log 2/\upsilon}{m}} \, ,
$$

then, we have

$$
\mathrm{NAMMD}(\mathbb{P}, \mathbb{Q}, \kappa) - \sqrt{\frac{9\log 2/\upsilon}{m}} > \epsilon + \frac{2\mathcal{N}_{1-\alpha}}{\sqrt{m}} \, ,
$$

which leads to

$$
m \geq \frac{\left(2 * \mathcal{N}_{1-\alpha} + \sqrt{9\log 2/\upsilon}\right)^2}{(\mathrm{NAMMD}(\mathbb{P}, \mathbb{Q}, \kappa) - \epsilon)^2} \, .
$$

This completes the proof. $\qquad\square$

## B.8 DETAILED PROOFS OF THEOREM 10

Let $\mathbf{\Pi}_{2m}$ be the set of all possible permutations of $\{1, \ldots, 2m\}$ over the pooled sample $Z = \{\boldsymbol{x}_1, \ldots, \boldsymbol{x}_m, \boldsymbol{y}_1, \ldots, \boldsymbol{y}_m\} = \{\boldsymbol{z}_1, \ldots, \boldsymbol{z}_m, \boldsymbol{z}_{m+1}, \ldots, \boldsymbol{z}_{2m}\}$. Given a permutation $\boldsymbol{\pi} = (\pi_1, \ldots, \pi_{2m}) \in \mathbf{\Pi}_{2m}$, we have $X_{\boldsymbol{\pi}} = \{\boldsymbol{z}_{\pi_i}\}_{i=1}^m$ and $Y_{\boldsymbol{\pi}} = \{\boldsymbol{z}_{\pi_i}\}_{i=m+1}^{2m}$.

We now present the proofs of Theorem 10 as follows.

*Proof.* Let $r_M$ be the $(1 - \alpha)$-quantile of the asymptotic null distribution of $m\widehat{\mathrm{MMD}}(X_{\boldsymbol{\pi}}, Y_{\boldsymbol{\pi}}, \kappa)$ from Theorem 16, where $X_{\boldsymbol{\pi}}$ and $Y_{\boldsymbol{\pi}}$ can be viewed as two i.i.d. samples drawn from $(\mathbb{P} + \mathbb{Q})/2$.

We also denote by $r_N$ be the $(1 - \alpha)$-quantile of the asymptotic null distribution of $m\widehat{\mathrm{NAMMD}}(X_{\boldsymbol{\pi}}, Y_{\boldsymbol{\pi}}, \kappa)$ from Theorem 2, and it is easy to see that

$$
r_N = \frac{r_M}{4K - \|(\boldsymbol{\mu}_{\mathbb{P}} + \boldsymbol{\mu}_{\mathbb{Q}})/\sqrt{2}\|_{\mathcal{H}_\kappa}^2} \, .
$$

It is easy to see that the inequality $m\widehat{\mathrm{MMD}}(X, Y, \kappa) > r_M$ can be rewritten as

$$
\frac{m\widehat{\mathrm{MMD}}(X, Y, \kappa)}{4K - \|(\boldsymbol{\mu}_{\mathbb{P}} + \boldsymbol{\mu}_{\mathbb{Q}})/\sqrt{2}\|_{\mathcal{H}_\kappa}^2} > r_N \, .
$$

Recall that

$$
\widehat{\mathrm{NAMMD}}(X_{\boldsymbol{\pi}}, Y_{\boldsymbol{\pi}}, \kappa) = \frac{\widehat{\mathrm{MMD}}(X_{\boldsymbol{\pi}}, Y_{\boldsymbol{\pi}}, \kappa)}{1/(m^2 - m)\sum_{i \neq j} 4K - \kappa(\boldsymbol{z}_{\boldsymbol{\pi}_i}, \boldsymbol{z}_{\boldsymbol{\pi}_j}) - \kappa(\boldsymbol{z}_{\boldsymbol{\pi}_{i+m}}, \boldsymbol{z}_{\boldsymbol{\pi}_{j+m}})} \, .
$$

We rewrite the inequality $m\widehat{\mathrm{NAMMD}}(X, Y, \kappa) > r_N$ as

$$
\frac{m\widehat{\mathrm{MMD}}(X, Y, \kappa)}{1/(m^2 - m)\sum_{i \neq j} 4K - \kappa(\boldsymbol{x}_i, \boldsymbol{x}_j) - \kappa(\boldsymbol{y}_i, \boldsymbol{y}_j)} > r_N \, .
$$

Then, the following relationship

$$m\widehat{\text{MMD}}(X, Y, \kappa) > r_M \quad \Rightarrow \quad m\widehat{\text{NAMMD}}(X, Y, \kappa) > r_N \ , \tag{5}$$

holds with

$$1/(m^2 - m) \sum_{i \neq j} 4K - \kappa(\boldsymbol{x}_i, \boldsymbol{x}_j) - \kappa(\boldsymbol{y}_i, \boldsymbol{y}_j) \leq 4K - \|(\boldsymbol{\mu}_{\mathbb{P}} + \boldsymbol{\mu}_{\mathbb{Q}})/\sqrt{2}\|_{\mathcal{H}_\kappa}^2 \ ,$$

which can be transformed to

$$\frac{\sum_{i \neq j} \kappa(\boldsymbol{x}_i, \boldsymbol{x}_j) + \kappa(\boldsymbol{y}_i, \boldsymbol{y}_j)}{m^2 - m} - \|\boldsymbol{\mu}_{\mathbb{P}}\|_{\mathcal{H}_\kappa}^2 - \|\boldsymbol{\mu}_{\mathbb{Q}}\|_{\mathcal{H}_\kappa}^2 \quad \geq \quad \left\|\frac{\boldsymbol{\mu}_{\mathbb{P}} + \boldsymbol{\mu}_{\mathbb{Q}}}{\sqrt{2}}\right\|_{\mathcal{H}_\kappa}^2 - \|\boldsymbol{\mu}_{\mathbb{P}}\|_{\mathcal{H}_\kappa}^2 - \|\boldsymbol{\mu}_{\mathbb{Q}}\|_{\mathcal{H}_\kappa}^2$$

$$\geq \quad -\frac{1}{2}\|\boldsymbol{\mu}_{\mathbb{P}} - \boldsymbol{\mu}_{\mathbb{Q}}\|_{\mathcal{H}_\kappa}^2 \ .$$

Using the large deviation bound as follows

$$P\left(\frac{\sum_{i \neq j} \kappa(\boldsymbol{x}_i, \boldsymbol{x}_j) + \kappa(\boldsymbol{y}_i, \boldsymbol{y}_j)}{m^2 - m} - (\|\boldsymbol{\mu}_{\mathbb{P}}\|_{\mathcal{H}_\kappa}^2 + \|\boldsymbol{\mu}_{\mathbb{Q}}\|_{\mathcal{H}_\kappa}^2) \leq -t\right) \leq \exp(-mt^2/4K^2) \ ,$$

with $t > 0$, the Eqn. 5 holds with probability at least

$$1 - \exp(-m\|\boldsymbol{\mu}_{\mathbb{P}} - \boldsymbol{\mu}_{\mathbb{Q}}\|_{\mathcal{H}_\kappa}^4/16K^2) \ .$$

This completes the proof of first part.

From Theorem 16, we have the test power of MMD test as follows

$$p_M = \Pr\left(m\widehat{\text{MMD}}^2(X, Y, \kappa) \geq r_M\right) \to \Phi\left(\frac{m\text{MMD}^2(\mathbb{P}, \mathbb{Q}, \kappa) - r_M}{\sqrt{m}\sigma_M}\right) \ .$$

The test power of NAMMD test is given by, according to Theorem 2,

$$p_N = \Pr\left(m\widehat{\text{NAMMD}}(X, Y, \kappa) \geq r_N\right) \to \Phi\left(\frac{m\text{NAMMD}(\mathbb{P}, \mathbb{Q}, \kappa) - r_N}{\sqrt{m}\sigma_{\mathbb{P},\mathbb{Q}}}\right) \ .$$

It is easy to see that

$$\Phi\left(\frac{m\text{NAMMD}(\mathbb{P}, \mathbb{Q}, \kappa) - r_N}{\sqrt{m}\sigma_{\mathbb{P},\mathbb{Q}}}\right) = \Phi\left(\frac{\frac{m\text{MMD}^2(\mathbb{P}, \mathbb{Q}, \kappa)}{4K - \|\boldsymbol{\mu}_{\mathbb{P}}\|_{\mathcal{H}_\kappa}^2 - \|\boldsymbol{\mu}_{\mathbb{Q}}\|_{\mathcal{H}_\kappa}^2} - \frac{r_M}{4K - \|(\boldsymbol{\mu}_{\mathbb{P}} + \boldsymbol{\mu}_{\mathbb{Q}})/\sqrt{2}\|_{\mathcal{H}_\kappa}^2}}{\frac{\sqrt{m}\sigma_M}{4K - \|\boldsymbol{\mu}_{\mathbb{P}}\|_{\mathcal{H}_\kappa}^2 - \|\boldsymbol{\mu}_{\mathbb{Q}}\|_{\mathcal{H}_\kappa}^2}}\right) \ ,$$

which yields that

$$p_N \to \Phi\left(\frac{m\text{MMD}^2(\mathbb{P}, \mathbb{Q}, \kappa) - r_M}{\sqrt{m}\sigma_M} + \left(1 - \frac{4K - \|\boldsymbol{\mu}_{\mathbb{P}}\|_{\mathcal{H}_\kappa}^2 - \|\boldsymbol{\mu}_{\mathbb{Q}}\|_{\mathcal{H}_\kappa}^2}{4K - \|(\boldsymbol{\mu}_{\mathbb{P}} + \boldsymbol{\mu}_{\mathbb{Q}})/\sqrt{2}\|_{\mathcal{H}_\kappa}^2}\right) r_M/(\sqrt{m}\sigma_M)\right) \ .$$

Let

$$A = \frac{m\text{MMD}^2(\mathbb{P}, \mathbb{Q}, \kappa) - r_M}{\sqrt{m}\sigma_M} \quad \text{and} \quad B = \left(1 - \frac{4K - \|\boldsymbol{\mu}_{\mathbb{P}}\|_{\mathcal{H}_\kappa}^2 - \|\boldsymbol{\mu}_{\mathbb{Q}}\|_{\mathcal{H}_\kappa}^2}{4K - \|(\boldsymbol{\mu}_{\mathbb{P}} + \boldsymbol{\mu}_{\mathbb{Q}})/\sqrt{2}\|_{\mathcal{H}_\kappa}^2}\right) r_M/(\sqrt{m}\sigma_M) \ ,$$

we have

$$\varsigma = p_N - p_M = \frac{1}{\sqrt{2\pi}} \int_A^{A+B} e^{-t^2/2} dt \ .$$

Let $A \geq -0.5$, we have

$$m_A \geq \left(\frac{-\sigma_M + \sqrt{\sigma_M^2 + 16\text{MMD}^2(\mathbb{P}, \mathbb{Q}, \kappa)r_M}}{4\text{MMD}^2(\mathbb{P}, \mathbb{Q}, \kappa)}\right)^2 \ .$$

In a similar manner, let $B \geq 0.05$, we have

$$m_B \geq \left( 20 r_M \left( 1 - \frac{4K - \|\boldsymbol{\mu}_{\mathbb{P}}\|^2_{\mathcal{H}_\kappa} - \|\boldsymbol{\mu}_{\mathbb{Q}}\|^2_{\mathcal{H}_\kappa}}{4K - \|(\boldsymbol{\mu}_{\mathbb{P}} + \boldsymbol{\mu}_{\mathbb{Q}})/\sqrt{2}\|^2_{\mathcal{H}_\kappa}} \right) / \sigma_M \right)^{-2}$$

By introducing

$$m \geq C' \quad \text{with} \quad C' = \max\{m_A, m_B\} ,$$

we have $B \geq 0.05$ and $A \geq -0.5$, and the lower bound of the power improvement is given by

$$\varsigma = p_N - p_M \geq \frac{1}{\sqrt{2\pi}} \int_{-0.5}^{-0.45} e^{-t^2/2} dt \geq 1/65 .$$

This completes the proof. $\qquad\square$

## B.9 DETAILED PROOFS OF THEOREM 12

Given Definition 11, we assume $\mathbb{P}_1$ and $\mathbb{Q}_1$ are known, and $X$ and $Y$ are two i.i.d. samples drawn from $\mathbb{P}_2$ and $\mathbb{Q}_2$. The goals of distribution closeness testing are to correctly reject null hypotheses with calculated statistics $\widehat{\mathrm{NAMMD}}(X, Y, \kappa)$ and $\widehat{\mathrm{MMD}}(X, Y, \kappa)$.

For simplicity, we let

$$\mathrm{NORM}(\mathbb{P}_1, \mathbb{Q}_1, \kappa) = 4K - \|\boldsymbol{\mu}_{\mathbb{P}_1}\|^2_{\mathcal{H}_\kappa} - \|\boldsymbol{\mu}_{\mathbb{Q}_1}\|^2_{\mathcal{H}_\kappa}$$
$$\mathrm{NORM}(\mathbb{P}_2, \mathbb{Q}_2, \kappa) = 4K - \|\boldsymbol{\mu}_{\mathbb{P}_2}\|^2_{\mathcal{H}_\kappa} - \|\boldsymbol{\mu}_{\mathbb{Q}_2}\|^2_{\mathcal{H}_\kappa} ,$$

and rewrite the empirical estimator with $X$ and $Y$ as follows

$$\widehat{\mathrm{NORM}}(X, Y, \kappa) = 1/(m^2 - m) \sum_{i \neq j} 4K - \kappa(\boldsymbol{x}_i, \boldsymbol{x}_j) - \kappa(\boldsymbol{y}_i, \boldsymbol{y}_j) .$$

*Proof.* Recall that $r'_M$ and $r'_N$ are the asymptotic thresholds of estimators $\sqrt{m}\widehat{\mathrm{MMD}}(X, Y, \kappa)$ and $\sqrt{m}\widehat{\mathrm{NAMMD}}(X, Y, \kappa)$, respectively.

Specifically, from Theorem 16, we have

$$r'_M = \sqrt{m}\mathrm{MMD}(\mathbb{P}_1, \mathbb{Q}_1, \kappa) + \sigma'_M \mathcal{N}_{1-\alpha} ,$$

where $\sigma_M^2 := 4E[H_{1,2}H_{1,3}] - 4(E[H_{1,2}])^2$ and $H_{i,j} = \kappa(\boldsymbol{x}_i, \boldsymbol{x}_j) + \kappa(\boldsymbol{y}_i, \boldsymbol{y}_j) - \kappa(\boldsymbol{x}_i, \boldsymbol{y}_j) - \kappa(\boldsymbol{y}_i, \boldsymbol{x}_j)$, and the expectation are taken with respect to $\boldsymbol{x}_1, \boldsymbol{x}_2, \boldsymbol{x}_3 \overset{\text{i.i.d.}}{\sim} \mathbb{P}_2$ and $\boldsymbol{y}_1, \boldsymbol{y}_2, \boldsymbol{y}_3 \overset{\text{i.i.d.}}{\sim} \mathbb{Q}_2$.

In a similar manner, from Theorem 2, we have

$$\begin{aligned} r'_N &= \sqrt{m}\mathrm{NAMMD}(\mathbb{P}_1, \mathbb{Q}_1, \kappa) + \sigma_{\mathbb{P}_2, \mathbb{Q}_2} \mathcal{N}_{1-\alpha} \\ &= \frac{\sqrt{m}\mathrm{MMD}(\mathbb{P}_1, \mathbb{Q}_1, \kappa)}{4K - \|\boldsymbol{\mu}_{\mathbb{P}_1}\|^2_{\mathcal{H}_\kappa} - \|\boldsymbol{\mu}_{\mathbb{Q}_1}\|^2_{\mathcal{H}_\kappa}} + \frac{\sigma'_M \mathcal{N}_{1-\alpha}}{(4K - \|\boldsymbol{\mu}_{\mathbb{P}_2}\|^2_{\mathcal{H}_\kappa} - \|\boldsymbol{\mu}_{\mathbb{Q}_2}\|^2_{\mathcal{H}_\kappa})} \\ &= \frac{\sqrt{m}\mathrm{MMD}(\mathbb{P}_1, \mathbb{Q}_1, \kappa)}{\mathrm{NORM}(\mathbb{P}_1, \mathbb{Q}_1, \kappa)} + \frac{\sigma'_M \mathcal{N}_{1-\alpha}}{\mathrm{NORM}(\mathbb{P}_2, \mathbb{Q}_2, \kappa)} , \end{aligned}$$

It is easy to see that $\sqrt{m}\widehat{\mathrm{MMD}}(X, Y, \kappa) > r'_M$ is equivalent to

$$\sqrt{m}\widehat{\mathrm{MMD}}(X, Y, \kappa) - \sqrt{m}\mathrm{MMD}(\mathbb{P}_1, \mathbb{Q}_1, \kappa) > \sigma'_M \mathcal{N}_{1-\alpha} , \qquad (6)$$

and in a similar manner, $\sqrt{m}\widehat{\mathrm{NAMMD}}(X, Y, \kappa) > r'_N$ is equivalent to

$$\frac{\mathrm{NORM}(\mathbb{P}_2, \mathbb{Q}_2, \kappa)}{\widehat{\mathrm{NORM}}(X, Y, \kappa)} \sqrt{m}\widehat{\mathrm{MMD}}(X, Y, \kappa) - \frac{\mathrm{NORM}(\mathbb{P}_2, \mathbb{Q}_2, \kappa)}{\mathrm{NORM}(\mathbb{P}_1, \mathbb{Q}_1, \kappa)} \sqrt{m}\mathrm{MMD}(\mathbb{P}_1, \mathbb{Q}_1, \kappa) > \sigma'_M \mathcal{N}_{1-\alpha} , \qquad (7)$$

Hence, to ensure

$$\sqrt{m}\widehat{\mathrm{MMD}}(X, Y, \kappa) > r'_M \implies \sqrt{m}\widehat{\mathrm{NAMMD}}(X, Y, \kappa) > r'_N , \qquad (8)$$

we must verify that, according to Eqn. 6 and 7,

$$\left(\frac{\text{NORM}(\mathbb{P}_2, \mathbb{Q}_2, \kappa)}{\widehat{\text{NORM}}(X, Y, \kappa)} - 1\right) \sqrt{m}\widehat{\text{MMD}}(X, Y, \kappa) \geq \left(\frac{\text{NORM}(\mathbb{P}_2, \mathbb{Q}_2, \kappa)}{\text{NORM}(\mathbb{P}_1, \mathbb{Q}_1, \kappa)} - 1\right) \sqrt{m}\text{MMD}(\mathbb{P}_1, \mathbb{Q}_1, \kappa).$$

(9)

Based on Eqn. 6, the inequality in Eqn. 9 can be adjusted to

$$\frac{\text{NORM}(\mathbb{P}_2, \mathbb{Q}_2, \kappa) - \widehat{\text{NORM}}(X, Y, \kappa)}{\widehat{\text{NORM}}(X, Y, \kappa)}$$

$$\geq \frac{\text{NORM}(\mathbb{P}_2, \mathbb{Q}_2, \kappa) - \text{NORM}(\mathbb{P}_1, \mathbb{Q}_1, \kappa)}{\text{NORM}(\mathbb{P}_1, \mathbb{Q}_1, \kappa)} \frac{\sqrt{m}\text{MMD}(\mathbb{P}_1, \mathbb{Q}_1, \kappa)}{\sqrt{m}\text{MMD}(\mathbb{P}_1, \mathbb{Q}_1, \kappa) + \sigma'_M \mathcal{N}_{1-\alpha}}$$

$$\geq \sqrt{m}\text{NAMMD}(\mathbb{P}_1, \mathbb{Q}_1, \kappa) \frac{\text{NORM}(\mathbb{P}_2, \mathbb{Q}_2, \kappa) - \text{NORM}(\mathbb{P}_1, \mathbb{Q}_1, \kappa)}{\sqrt{m}\text{MMD}(\mathbb{P}_1, \mathbb{Q}_1, \kappa) + \sigma'_M \mathcal{N}_{1-\alpha}}.$$

Given this, we have

$$\text{NORM}(\mathbb{P}_2, \mathbb{Q}_2, \kappa)$$

$$\geq \left(1 + \sqrt{m}\text{NAMMD}(\mathbb{P}_1, \mathbb{Q}_1, \kappa) \frac{\text{NORM}(\mathbb{P}_2, \mathbb{Q}_2, \kappa) - \text{NORM}(\mathbb{P}_1, \mathbb{Q}_1, \kappa)}{\sqrt{m}\text{MMD}(\mathbb{P}_1, \mathbb{Q}_1, \kappa) + \sigma'_M \mathcal{N}_{1-\alpha}}\right) \widehat{\text{NORM}}(X, Y, \kappa)$$

$$\geq (1 - \Delta)\widehat{\text{NORM}}(X, Y, \kappa),$$

where we let, for simplicity

$$\Delta = \sqrt{m}\text{NAMMD}(\mathbb{P}_1, \mathbb{Q}_1, \kappa) \frac{\|\boldsymbol{\mu}_{\mathbb{P}_2}\|^2_{\mathcal{H}_\kappa} + \|\boldsymbol{\mu}_{\mathbb{Q}_2}\|^2_{\mathcal{H}_\kappa} - \|\boldsymbol{\mu}_{\mathbb{P}_1}\|^2_{\mathcal{H}_\kappa} - \|\boldsymbol{\mu}_{\mathbb{Q}_1}\|^2_{\mathcal{H}_\kappa}}{\sqrt{m}\text{MMD}(\mathbb{P}_1, \mathbb{Q}_1, \kappa) + \sigma'_M \mathcal{N}_{1-\alpha}}.$$

Here, by assuming $\|\boldsymbol{\mu}_{\mathbb{P}_1}\|^2_{\mathcal{H}_\kappa} + \|\boldsymbol{\mu}_{\mathbb{Q}_1}\|^2_{\mathcal{H}_\kappa} < \|\boldsymbol{\mu}_{\mathbb{P}_2}\|^2_{\mathcal{H}_\kappa} + \|\boldsymbol{\mu}_{\mathbb{Q}_2}\|^2_{\mathcal{H}_\kappa}$, we have $\Delta \in (0, 1/2)$.

As we can see, $\text{NORM}(\mathbb{P}_2, \mathbb{Q}_2, \kappa) \geq (1 - \Delta)\widehat{\text{NORM}}(X, Y, \kappa)$ is equivalent to

$$(1 - \Delta)\widehat{\text{NORM}}(X, Y, \kappa) - (1 - \Delta)\text{NORM}(\mathbb{P}_2, \mathbb{Q}_2, \kappa) \leq \Delta \cdot \text{NORM}(\mathbb{P}_2, \mathbb{Q}_2, \kappa),$$

which is

$$\widehat{\text{NORM}}(X, Y, \kappa) - \text{NORM}(\mathbb{P}_2, \mathbb{Q}_2, \kappa) \leq \frac{\Delta}{1 - \Delta}\text{NORM}(\mathbb{P}_2, \mathbb{Q}_2, \kappa).$$

Using the large deviation bound as follows

$$P\left(\widehat{\text{NORM}}(X, Y, \kappa) - \text{NORM}(\mathbb{P}_2, \mathbb{Q}_2, \kappa) \geq t\right) \leq \exp(-mt^2/4K^2),$$

with $t > 0$, the Eqn. 8 holds with probability at least

$$1 - \exp\left(-m\left(\frac{\Delta}{1 - \Delta}\text{NORM}(\mathbb{P}_2, \mathbb{Q}_2, \kappa)\right)^2 /4K^2\right).$$

This completes the proof of first part.

The proof of second part closely mirrors the proof of second part in Theorem 10 given in Appendix B.8.

From Theorem 16, we have the test power of MMD test as follows

$$p_M = \text{Pr}\left(\sqrt{m}\widehat{\text{MMD}}^2(X, Y, \kappa) \geq r'_M\right) \to \Phi\left(\frac{\sqrt{m}\text{MMD}^2(\mathbb{P}_2, \mathbb{Q}_2, \kappa) - r'_M}{\sigma'_M}\right),$$

which is equivalent to

$$\Phi\left(\frac{\sqrt{m}(\text{MMD}^2(\mathbb{P}_2, \mathbb{Q}_2, \kappa) - \text{MMD}^2(\mathbb{P}_1, \mathbb{Q}_1, \kappa)) - \sigma'_M \mathcal{N}_{1-\alpha}}{\sigma'_M}\right)$$

The test power of NAMMD test is given by, according to Theorem 2,

$$p_N = \Pr\left(\sqrt{m}\widehat{\text{NAMMD}}(X, Y, \kappa) \geq r'_N\right) \rightarrow \Phi\left(\frac{\sqrt{m}\text{NAMMD}(\mathbb{P}_2, \mathbb{Q}_2, \kappa) - r'_N}{\sigma_{\mathbb{P}_2, \mathbb{Q}_2}}\right),$$

which is equivalent to

$$\Phi\left(\frac{\sqrt{m}\left(\text{MMD}^2(\mathbb{P}_2, \mathbb{Q}_2, \kappa) - \frac{\text{NORM}(\mathbb{P}_2, \mathbb{Q}_2, \kappa)}{\text{NORM}(\mathbb{P}_1, \mathbb{Q}_1, \kappa)}\text{MMD}^2(\mathbb{P}_1, \mathbb{Q}_1, \kappa)\right) - \sigma'_M \mathcal{N}_{1-\alpha}}{\sigma'_M}\right).$$

For simplicity, we let

$$A = \frac{\sqrt{m}(\text{MMD}^2(\mathbb{P}_2, \mathbb{Q}_2, \kappa) - \text{MMD}^2(\mathbb{P}_1, \mathbb{Q}_1, \kappa)) - \sigma'_M \mathcal{N}_{1-\alpha}}{\sigma'_M},$$

and

$$B = \sqrt{m}\left(1 - \frac{\text{NORM}(\mathbb{P}_2, \mathbb{Q}_2, \kappa)}{\text{NORM}(\mathbb{P}_1, \mathbb{Q}_1, \kappa)}\right)\frac{\text{MMD}^2(\mathbb{P}_1, \mathbb{Q}_1, \kappa)}{\sigma'_M}.$$

Similarly, by assuming $\|\boldsymbol{\mu}_{\mathbb{P}_1}\|^2_{\mathcal{H}_\kappa} + \|\boldsymbol{\mu}_{\mathbb{Q}_1}\|^2_{\mathcal{H}_\kappa} < \|\boldsymbol{\mu}_{\mathbb{P}_2}\|^2_{\mathcal{H}_\kappa} + \|\boldsymbol{\mu}_{\mathbb{Q}_2}\|^2_{\mathcal{H}_\kappa}$, we have $B > 0$ with $\text{NORM}(\mathbb{P}_1, \mathbb{Q}_1, \kappa) > \text{NORM}(\mathbb{P}_2, \mathbb{Q}_2, \kappa)$.

As we can see,

$$\varsigma = p_N - p_M = \frac{1}{\sqrt{2\pi}}\int_A^{A+B} e^{-t^2/2} dt.$$

Let $A \geq -0.5$, we have

$$m_A \geq \left(\frac{(\mathcal{N}_{1-\alpha} - 0.5)\sigma'_M}{\text{MMD}^2(\mathbb{P}_2, \mathbb{Q}_2, \kappa) - \text{MMD}^2(\mathbb{P}_1, \mathbb{Q}_1, \kappa)}\right)^2.$$

In a similar manner, let $B \geq 0.05$, we have

$$m_B \geq \left(20\left(1 - \frac{\text{NORM}(\mathbb{P}_2, \mathbb{Q}_2, \kappa)}{\text{NORM}(\mathbb{P}_1, \mathbb{Q}_1, \kappa)}\right)\frac{\text{MMD}^2(\mathbb{P}_1, \mathbb{Q}_1, \kappa)}{\sigma'_M}\right)^{-2}.$$

By introducing

$$m \geq C'' \text{ with } C'' = \max\{m_A, m_B\},$$

we have $B \geq 0.05$ and $A \geq -0.5$, and the lower bound of the power improvement is given by

$$\varsigma = p_N - p_M \geq \frac{1}{\sqrt{2\pi}}\int_{-0.5}^{-0.45} e^{-t^2/2} dt \geq 1/65.$$

This completes the proof. $\qquad\square$

## B.10 DISCUSSIONS ON THE IMPROVEMENT OF NAMMD WITH KERNEL SELECTION

Existing kernel selection methods for MMD are primarily designed for *two-sample testing (TST)*, focusing on selecting the optimal kernel that maximizes the *test power estimator of TST* to distinguish two fixed distributions $\mathbb{P}$ and $\mathbb{Q}$ [27, 36]. For TST, NAMMD and MMD actually share the *same test power estimator* because, asymptotically, after we fixed two distributions $\mathbb{P}$ and $\mathbb{Q}$, NAMMD can be viewed as MMD scaled by a constant $4K - \|\boldsymbol{\mu}_{\mathbb{P}}\|^2_{\mathcal{H}_\kappa} - \|\boldsymbol{\mu}_{\mathbb{Q}}\|^2_{\mathcal{H}_\kappa}$, as detailed in Appendix C.1. Hence, the NAMMD and MMD has the *same optimal kernel for TST*. For the same kernel, when we use permutation test to do perform two-sample tests, our NAMMD achieves higher test power than MMD due to its scaling as stated in Theorem 10.

---

**Algorithm 1** Kernel Selection

---

**Input**: Two samples $X$ and $Y$, a kernel $\kappa$, step size $\eta$, iteration number $N$
**Output**: Two samples $X$ and $Y$

1: **for** $\ell = 1, 2, \cdots, N$ **do**
2:     Calculate the estimator $\widehat{\mathrm{NAMMD}}(X, Y, \kappa)/\sigma_{X,Y}$ according to Eqn. 10
3:     Calculate gradient $\nabla \cdot \left( \widehat{\mathrm{NAMMD}}(X, Y, \kappa)/\sigma_{X,Y} \right)$
4:     Gradient ascend with step size $\eta$ by the Adam method
5: **end for**

---

**One conjunction for distribution closeness testing (DCT).** Further, based on Theorem 12, we might have an interesting conjunction. We can assume a scenario where we can obtain the best kernel $\kappa_*^{\mathrm{M}}$ for MMD DCT (instead of MMD TST) and the best kernel $\kappa_*^{\mathrm{N}}$ for NAMMD DCT. Based on Theorem 12, if we use the kernel $\kappa_*^{\mathrm{M}}$ (MMD's best kernel) for NAMMD, then NAMMD DCT will perform better than MMD DCT already. Because $\kappa_*^{\mathrm{N}}$ is the kernel to make NAMMD DCT have the highest test power (in DCT, instead of TST), NAMMD DCT with $\kappa_*^{\mathrm{N}}$ should have a higher or equal test power compared to NAMMD DCT with $\kappa_*^{\mathrm{M}}$. Thus, NAMMD DCT with $k_*^{\mathrm{D}}$ has a higher test power than NAMMD DCT with $\kappa_*^{\mathrm{M}}$ (because NAMMD DCT with $\kappa_*^{\mathrm{M}}$ has a higher power than MMD DCT with $\kappa_*^{\mathrm{M}}$ based on Theorem 12).

## C    DETAILS AND ADDITIONAL DISCUSSIONS OF OUR NAMMD TEST

### C.1    DETAILS OF OPTIMIZATION FOR KERNEL SELECTING

Recall Theorem 2, if $\mathrm{NAMMD}(\mathbb{P}, \mathbb{Q}, \kappa) = \epsilon$ with $\epsilon \in (0, 1)$, we have

$$\sqrt{m}(\widehat{\mathrm{NAMMD}}(X, Y, \kappa) - \epsilon) \xrightarrow{d} \mathcal{N}(0, \sigma_{\mathbb{P},\mathbb{Q}}^2) \,,$$

where $\sigma_{\mathbb{P},\mathbb{Q}} = \sqrt{4E[H_{1,2}H_{1,3}] - 4(E[H_{1,2}])^2}/(4K - \|\boldsymbol{\mu}_{\mathbb{P}}\|_{\mathcal{H}_\kappa}^2 - \|\boldsymbol{\mu}_{\mathbb{Q}}\|_{\mathcal{H}_\kappa}^2)$, and the expectation are taken over $\boldsymbol{x}_1, \boldsymbol{x}_2, \boldsymbol{x}_3 \sim \mathbb{P}^3$ and $\boldsymbol{y}_1, \boldsymbol{y}_2, \boldsymbol{y}_3 \sim \mathbb{Q}^3$.

We can find the approximate test power by using the asymptotic testing threshold $r_N$ as follows:

$$\Pr\left( m\widehat{\mathrm{NAMMD}}(X, Y, \kappa) \geq r_N \right) \to \Phi\left( \frac{m\mathrm{NAMMD}(\mathbb{P}, \mathbb{Q}, \kappa) - r_N}{\sqrt{m}\sigma_{\mathbb{P},\mathbb{Q}}} \right) \,.$$

It is evident that maximizing the test power is equivalent to optimizing the following term

$$\frac{\mathrm{NAMMD}(\mathbb{P}, \mathbb{Q}, \kappa)}{\sigma_{\mathbb{P},\mathbb{Q}}} = \frac{\mathrm{MMD}(\mathbb{P}, \mathbb{Q}, \kappa)}{\sqrt{4E[H_{1,2}H_{1,3}] - 4(E[H_{1,2}])^2}} \,.$$

Recall that

$$\widehat{\mathrm{NAMMD}}(X, Y, \kappa) = \sum_{i \neq j} H_{i,j} / \sum_{i \neq j} (4K - \kappa(\boldsymbol{x}_i, \boldsymbol{x}_j) - \kappa(\boldsymbol{y}_i, \boldsymbol{y}_j)) \,,$$

with $H_{i,j} = \kappa(\boldsymbol{x}_i, \boldsymbol{x}_j) + \kappa(\boldsymbol{y}_i, \boldsymbol{y}_j) - \kappa(\boldsymbol{x}_i, \boldsymbol{y}_j) - \kappa(\boldsymbol{y}_i, \boldsymbol{x}_j)$ and

$$\sigma_{X,Y} = \frac{\sqrt{((4m - 8)\zeta_1 + 2\zeta_2)/(m - 1)}}{(m^2 - m)^{-1} \sum_{i \neq j} 4K - \kappa(\boldsymbol{x}_i, \boldsymbol{x}_j) - \kappa(\boldsymbol{y}_i, \boldsymbol{y}_j)} \,,$$

where $\zeta_1$ and $\zeta_2$ are standard variance components of the MMD [41, 42]. The details of the $\zeta_1$ and $\zeta_2$ are provided in Appendix C.2.

We have the empirical test power estimator as follows

$$\frac{\widehat{\mathrm{NAMMD}}(X, Y, \kappa)}{\sigma_{X,Y}} = \frac{\widehat{\mathrm{MMD}}(X, Y, \kappa)}{\sqrt{((4m - 8)\zeta_1 + 2\zeta_2)/(m - 1)}} \,, \tag{10}$$

It is evident that the empirical test power estimator for NAMMD is equal to the test power estimator of MMD [36]. We take gradient method [70] for the optimization of Eqn. 10. Algorithm 1 presents the detailed description on optimization.

## C.2 DETAILS OF VARIANCE ESTIMATOR

We adhere to the results of empirical variance estimators provided by Sutherland [42]. For simplicity, we first introduce the uncentred covariance operator as follows:

$$C_X = E_{\boldsymbol{x} \sim \mathbb{P}}[\varphi(\boldsymbol{x}) \otimes \varphi(\boldsymbol{x})],$$

where $\varphi(\cdot)$ is the feature map of the corresponding RKHS $\mathcal{H}_\kappa$.

For simplicity, we define the $m \times m$ matrix $\mathbf{K_{XY}}$ with $(\mathbf{K_{XY}})_{ij} = \kappa(\boldsymbol{x}_i, \boldsymbol{y}_j)$. Let $\tilde{\mathbf{K}}_{\mathbf{XY}}$ be $\mathbf{K_{XY}}$ with diagonals set to zero. In a similar manner, we have $\mathbf{K_{XX}}$ and $\mathbf{K_{YY}}$, and $\tilde{\mathbf{K}}_{\mathbf{XX}}$ and $\tilde{\mathbf{K}}_{\mathbf{YY}}$. Let $\mathbf{1}$ be the $m$-vector of all ones. Denote by $(m)_k := m(m-1)\cdots(m-k+1)$.

We have that

$$
\begin{aligned}
\zeta_1 =\ & \langle \boldsymbol{\mu}_X, C_X \boldsymbol{\mu}_X \rangle - \langle \boldsymbol{\mu}_X, \boldsymbol{\mu}_X \rangle^2 + \langle \boldsymbol{\mu}_Y, C_Y \boldsymbol{\mu}_Y \rangle - \langle \boldsymbol{\mu}_Y, \boldsymbol{\mu}_Y \rangle^2 \\
& + \langle \boldsymbol{\mu}_Y, C_X \boldsymbol{\mu}_Y \rangle + \langle \boldsymbol{\mu}_X, C_Y \boldsymbol{\mu}_X \rangle - \langle \boldsymbol{\mu}_X, \boldsymbol{\mu}_Y \rangle^2 - \langle \boldsymbol{\mu}_Y, \boldsymbol{\mu}_X \rangle^2 \\
& - 2\langle \boldsymbol{\mu}_X, C_X \boldsymbol{\mu}_Y \rangle + 2\langle \boldsymbol{\mu}_X, \boldsymbol{\mu}_X \rangle \langle \boldsymbol{\mu}_X, \boldsymbol{\mu}_Y \rangle - 2\langle \boldsymbol{\mu}_Y, C_Y \boldsymbol{\mu}_X \rangle + 2\langle \boldsymbol{\mu}_Y, \boldsymbol{\mu}_Y \rangle \langle \boldsymbol{\mu}_X, \boldsymbol{\mu}_Y \rangle \\
=\ & \frac{1}{(m)_3}\left[\left\|\tilde{\mathbf{K}}_{\mathbf{XX}}\mathbf{1}\right\|^2 - \left\|\tilde{\mathbf{K}}_{\mathbf{XX}}\right\|_F^2\right] - \frac{1}{(m)_4}\left[\left(\mathbf{1}^\top\tilde{\mathbf{K}}_{\mathbf{XX}}\mathbf{1}\right)^2 - 4\left\|\tilde{\mathbf{K}}_{\mathbf{XX}}\mathbf{1}\right\|^2 + 2\left\|\tilde{\mathbf{K}}_{\mathbf{XX}}\right\|_F^2\right] \\
& + \frac{1}{(m)_3}\left[\left\|\tilde{\mathbf{K}}_{\mathbf{YY}}\mathbf{1}\right\|^2 - \left\|\tilde{\mathbf{K}}_{\mathbf{YY}}\right\|_F^2\right] - \frac{1}{(m)_4}\left[\left(\mathbf{1}^\top\tilde{\mathbf{K}}_{\mathbf{YY}}\mathbf{1}\right)^2 - 4\left\|\tilde{\mathbf{K}}_{\mathbf{YY}}\mathbf{1}\right\|^2 + 2\left\|\tilde{\mathbf{K}}_{\mathbf{YY}}\right\|_F^2\right] \\
& + \frac{1}{m^2(m-1)}\left[\|\mathbf{K_{XY}}\mathbf{1}\|^2 - \|\mathbf{K_{XY}}\|_F^2\right] + \frac{1}{m^2(m-1)}\left[\|\mathbf{K_{XY}^\top}\mathbf{1}\|^2 - \|\mathbf{K_{XY}}\|_F^2\right] \\
& - \frac{2}{m^2(m-1)^2}\left[\left(\mathbf{1}^\top\mathbf{K_{XY}}\mathbf{1}\right)^2 - \|\mathbf{K_{XY}^\top}\mathbf{1}\|^2 - \|\mathbf{K_{XY}}\mathbf{1}\|^2 + \|\mathbf{K_{XY}}\|_F^2\right] \\
& - \frac{2}{m^2(m-1)}\mathbf{1}^\top\tilde{\mathbf{K}}_{\mathbf{XX}}\mathbf{K_{XY}}\mathbf{1} + \frac{2}{m(m)_3}\left[\mathbf{1}^\top\tilde{\mathbf{K}}_{\mathbf{XX}}\mathbf{1}\mathbf{1}^\top\mathbf{K_{XY}}\mathbf{1} - 2\mathbf{1}^\top\tilde{\mathbf{K}}_{\mathbf{XX}}\mathbf{K_{XY}}\mathbf{1}\right] \\
& - \frac{2}{m^2(m-1)}\mathbf{1}^\top\tilde{\mathbf{K}}_{\mathbf{YY}}\mathbf{K_{XY}^\top}\mathbf{1} + \frac{2}{m(m)_3}\left[\mathbf{1}^\top\tilde{\mathbf{K}}_{\mathbf{YY}}\mathbf{1}\mathbf{1}^\top\mathbf{K_{XY}^\top}\mathbf{1} - 2\mathbf{1}^\top\tilde{\mathbf{K}}_{\mathbf{YY}}\mathbf{K_{XY}^\top}\mathbf{1}\right]
\end{aligned}
$$

and

$$
\begin{aligned}
\zeta_2 =\ & \mathbb{E}\left[\kappa(\boldsymbol{x}_1, \boldsymbol{x}_2)^2\right] - \langle \boldsymbol{\mu}_X, \boldsymbol{\mu}_X \rangle^2 + \mathbb{E}\left[\kappa(\boldsymbol{y}_1, \boldsymbol{y}_2)^2\right] \\
& - \langle \boldsymbol{\mu}_Y, \boldsymbol{\mu}_Y \rangle^2 + 2\mathbb{E}\left[\kappa(\boldsymbol{x}, \boldsymbol{y})^2\right] - 2\langle \boldsymbol{\mu}_X, \boldsymbol{\mu}_Y \rangle^2 \\
& - 4\langle \boldsymbol{\mu}_X, C_X \boldsymbol{\mu}_Y \rangle + 4\langle \boldsymbol{\mu}_X, \boldsymbol{\mu}_X \rangle \langle \boldsymbol{\mu}_X, \boldsymbol{\mu}_Y \rangle - 4\langle \boldsymbol{\mu}_Y, C_Y \boldsymbol{\mu}_X \rangle + 4\langle \boldsymbol{\mu}_Y, \boldsymbol{\mu}_Y \rangle \langle \boldsymbol{\mu}_X, \boldsymbol{\mu}_Y \rangle \\
=\ & \frac{1}{m(m-1)}\left\|\tilde{\mathbf{K}}_{\mathbf{XX}}\right\|_F^2 - \frac{1}{(m)_4}\left[\left(\mathbf{1}^\top\tilde{\mathbf{K}}_{\mathbf{XX}}\mathbf{1}\right)^2 - 4\left\|\tilde{\mathbf{K}}_{\mathbf{XX}}\mathbf{1}\right\|^2 + 2\left\|\tilde{\mathbf{K}}_{\mathbf{XX}}\right\|_F^2\right] \\
& + \frac{1}{m(m-1)}\left\|\tilde{\mathbf{K}}_{\mathbf{YY}}\right\|_F^2 - \frac{1}{(m)_4}\left[\left(\mathbf{1}^\top\tilde{\mathbf{K}}_{\mathbf{YY}}\mathbf{1}\right)^2 - 4\left\|\tilde{\mathbf{K}}_{\mathbf{YY}}\mathbf{1}\right\|^2 + 2\left\|\tilde{\mathbf{K}}_{\mathbf{YY}}\right\|_F^2\right] \\
& + \frac{2}{m^2}\|\mathbf{K_{XY}}\|_F^2 - \frac{2}{m^2(m-1)^2}\left[\left(\mathbf{1}^\top\mathbf{K_{XY}}\mathbf{1}\right)^2 - \|\mathbf{K_{XY}^\top}\mathbf{1}\|^2 - \|\mathbf{K_{XY}}\mathbf{1}\|^2 + \|\mathbf{K_{XY}}\|_F^2\right] \\
& - \frac{4}{m^2(m-1)}\mathbf{1}^\top\tilde{\mathbf{K}}_{\mathbf{XX}}\mathbf{K_{XY}}\mathbf{1} + \frac{4}{m(m)_3}\left[\mathbf{1}^\top\tilde{\mathbf{K}}_{\mathbf{XX}}\mathbf{1}\mathbf{1}^\top\mathbf{K_{XY}}\mathbf{1} - 2\mathbf{1}^\top\tilde{\mathbf{K}}_{\mathbf{XX}}\mathbf{K_{XY}}\mathbf{1}\right] \\
& - \frac{4}{m^2(m-1)}\mathbf{1}^\top\tilde{\mathbf{K}}_{\mathbf{YY}}\mathbf{K_{XY}^\top}\mathbf{1} + \frac{4}{m(m)_3}\left[\mathbf{1}^\top\tilde{\mathbf{K}}_{\mathbf{YY}}\mathbf{1}\mathbf{1}^\top\mathbf{K_{XY}^\top}\mathbf{1} - 2\mathbf{1}^\top\tilde{\mathbf{K}}_{\mathbf{YY}}\mathbf{K_{XY}^\top}\mathbf{1}\right].
\end{aligned}
$$

where $\langle \cdot, \cdot \rangle$ denotes the inner product in RKHS $\mathcal{H}_\kappa$.

## C.3 DETAILS OF OUR NAMMDFUSE

Following the fusing statistics approach [33], we introduce the NAMMDFuse statistic through exponentiation of NAMMD with samples $X$ and $Y$ as follows

$$\widehat{\text{FUSE}}(X, Y) = \frac{1}{\lambda}\log\left(E_{\kappa \sim \pi(\langle X, Y \rangle)}\left[\exp\left(\lambda\frac{\widehat{\text{NAMMD}}(X, Y, \kappa)}{\sqrt{\widehat{N}(X, Y)}}\right)\right]\right)$$

where $\lambda > 0$ and $\widehat{N}(X, Y) = \frac{1}{m(m-1)} \sum_{i \neq j}^m \kappa(\mathbf{x}_i, \mathbf{x}_j)^2 + \kappa(\mathbf{y}_i, \mathbf{y}_j)^2$ is permutation invariant. $\pi(\langle X, Y \rangle)$ is the prior distribution on the kernel space $\mathcal{K}$. In experiments, we set the prior distribution $\pi(\langle X, Y \rangle)$ and the kernel space $\mathcal{K}$ to be the same for MMDFuse.

## C.4 LIMITATION STATEMENT

Our analysis in this paper focuses on kernels of the form $\kappa(\boldsymbol{x}, \boldsymbol{x}') = \Psi(\boldsymbol{x} - \boldsymbol{x}') \leq K$ with a positive-definite $\Psi(\cdot)$ and $\Psi(\mathbf{0}) = K$, including Laplace [33], Mahalanobis [30] and Deep kernels [27] (frequently used in kernel-based hypothesis testing). For these kernels, *a lager norm of mean embedding* $\|\boldsymbol{\mu}_\mathbb{P}\|^2_{\mathcal{H}_\kappa}$ *indicates a smaller variance* $Var(\mathbb{P}, \kappa) = K - \|\boldsymbol{\mu}_\mathbb{P}\|^2_{\mathcal{H}_\kappa}$, *which corresponds to a more tightly concentrated distribution* $\mathbb{P}$. Leveraging this property, we gain the insight that two distributions can be separated more effectively at the same MMD distance with larger norms. Hence, we scale MMD using $4K - \|\boldsymbol{\mu}_\mathbb{P}\|^2_{\mathcal{H}_\kappa} - \|\boldsymbol{\mu}_\mathbb{Q}\|^2_{\mathcal{H}_\kappa}$, making the new NAMMD increase with the norms $\|\boldsymbol{\mu}_\mathbb{P}\|^2_{\mathcal{H}_\kappa}$ and $\|\boldsymbol{\mu}_\mathbb{Q}\|^2_{\mathcal{H}_\kappa}$. Figure 1c and 1d demonstrate that our NAMMD exhibits a stronger correlation with the $p$-value in testing, while MMD is held constant. We also prove that scaling improves NAMMD's effectiveness as a closeness measure in Theorems 10 and 12.

However, all these improvements rely on the property that *"A lager norm of mean embedding* $\|\boldsymbol{\mu}_\mathbb{P}\|^2_{\mathcal{H}_\kappa}$ *indicates a smaller variance* $Var(\mathbb{P}, \kappa) = K - \|\boldsymbol{\mu}_\mathbb{P}\|^2_{\mathcal{H}_\kappa}$, *which corresponds to a more tightly concentrated distribution* $\mathbb{P}$*"*. The proposed method may not work well for kernels where the embedding norm of distribution may increases as the data variance increases. For these kernels, the "less informative" of MMD still arises when assessing the closeness levels for multiple distribution pairs with the same kernel, i.e., MMD value can be the same for many pairs of distributions that have different norms in the same RKHS. We will demonstrate this by further considering two other types of kernels as follows.

**Unbounded kernels for bounded data**: For polynomial kernels of the form

$$\kappa(\mathbf{x}, \mathbf{x}') = (\mathbf{x}^T \mathbf{x}' + c)^d ,$$

We define $\mathbb{P}_1 = \{\frac{1}{4}, \frac{3}{4}\}$ and $\mathbb{Q}_1 = \{\frac{1}{2}, \frac{1}{2}\}$ be discrete distributions over vector domains $\{(\sqrt{c}, ..., 0), (-\sqrt{c}, ..., 0)\}$, respectively. Furthermore, we define $\mathbb{P}_2 = \{\frac{3}{4}, \frac{1}{4}\}$ and $\mathbb{Q}_2 = \{1, 0\}$ be discrete distributions over domains $\{(\sqrt{c}, ..., 0), (-\sqrt{c}, ..., 0)\}$. It is evident that

$$\text{MMD}(\mathbb{P}_1, \mathbb{Q}_1, \kappa) = \text{MMD}(\mathbb{P}_2, \mathbb{Q}_2, \kappa) = \frac{1}{8}(2c)^d ,$$

with different norms for distributions pairs $\|\boldsymbol{\mu}_{\mathbb{P}_1}\|^2_{\mathcal{H}_\kappa} + \|\boldsymbol{\mu}_{\mathbb{Q}_1}\|^2_{\mathcal{H}_\kappa} = \frac{9}{8}(2c)^d$, and $\|\boldsymbol{\mu}_{\mathbb{P}_2}\|^2_{\mathcal{H}_\kappa} + \|\boldsymbol{\mu}_{\mathbb{Q}_2}\|^2_{\mathcal{H}_\kappa} = \frac{13}{8}(2c)^d$. Specifically, we have $\|\boldsymbol{\mu}_{\mathbb{P}_1}\|^2_{\mathcal{H}_\kappa} = \frac{5}{8}(2c)^d$, $\|\boldsymbol{\mu}_{\mathbb{Q}_1}\|^2_{\mathcal{H}_\kappa} = \frac{1}{2}(2c)^d$, $\|\boldsymbol{\mu}_{\mathbb{P}_2}\|^2_{\mathcal{H}_\kappa} = \frac{5}{8}(2c)^d$ and $\|\boldsymbol{\mu}_{\mathbb{Q}_2}\|^2_{\mathcal{H}_\kappa} = (2c)^d$.

In a similar manner, for matrix products kernels of the form

$$\kappa(\mathbf{x}, \mathbf{x}') = (\mathbf{x}^T M \mathbf{x}' + c)^d ,$$

and denote by $M_{11}$ the element in the first row and first column of the matrix $M$. We define $\mathbb{P}_1 = \{\frac{1}{4}, \frac{3}{4}\}$ and $\mathbb{Q}_1 = \{\frac{1}{2}, \frac{1}{2}\}$ over vector domains $\{(\sqrt{c/M_{11}}, ..., 0), (-\sqrt{c/M_{11}}, ..., 0)\}$, respectively. Furthermore, we define $\mathbb{P}_2 = \{\frac{3}{4}, \frac{1}{4}\}$ and $\mathbb{Q}_2 = \{1, 0\}$ over domains $\{(\sqrt{c/M_{11}}, ..., 0), (-\sqrt{c/M_{11}}, ..., 0)\}$. We obtain the same results as for polynomial kernels.

**Kernels with a positive limit at infinity**: Using the kernel as $\kappa(\mathbf{x}, \mathbf{x}') = \exp(-\frac{\|\mathbf{x} - \mathbf{x}'\|^2}{2\gamma})$ when $\|\mathbf{x} - \mathbf{x}'\|_\infty < K$, and otherwise $\kappa(\mathbf{x}, \mathbf{x}')$ with positive constants $K$ and $c$. We define $\mathbb{P}_1 = \{\frac{1}{4}, \frac{3}{4}\}$ and $\mathbb{Q}_1 = \{\frac{3}{4}, \frac{1}{4}\}$ over vector domains $\{(K, ..., 0), (4K, ..., 0)\}$, respectively. Furthermore, we define $\mathbb{P}_2 = \{\frac{1}{2}, \frac{1}{2}\}$ and $\mathbb{Q}_2 = \{1, 0\}$ over domains $\{(K, ..., 0), (4K, ..., 0)\}$. It is evident that

$$\text{MMD}(\mathbb{P}_1, \mathbb{Q}_1, \kappa) = \text{MMD}(\mathbb{P}_2, \mathbb{Q}_2, \kappa) = \frac{1}{2}(1 - c) ,$$

with different norms for pairs $\|\boldsymbol{\mu}_{\mathbb{P}_1}\|_{\mathcal{H}_\kappa} + \|\boldsymbol{\mu}_{\mathbb{Q}_1}\|^2_{\mathcal{H}_\kappa} = \frac{5+3c}{4}$, and $\|\boldsymbol{\mu}_{\mathbb{P}_2}\|^2_{\mathcal{H}_\kappa} + \|\boldsymbol{\mu}_{\mathbb{Q}_2}\|^2_{\mathcal{H}_\kappa} = \frac{3+c}{2}$. Specifically, we have $\|\boldsymbol{\mu}_{\mathbb{P}_1}\|^2_{\mathcal{H}_\kappa} = \frac{5+3c}{8}$, $\|\boldsymbol{\mu}_{\mathbb{Q}_1}\|^2_{\mathcal{H}_\kappa} = \frac{5+3c}{8}$, $\|\boldsymbol{\mu}_{\mathbb{P}_2}\|^2_{\mathcal{H}_\kappa} = \frac{1+c}{2}$ and $\|\boldsymbol{\mu}_{\mathbb{Q}_2}\|^2_{\mathcal{H}_\kappa} = 1$.

For these kernels, the relationship between the norm of mean embedding and the variance of distribution is not monotonic, where a smaller norm of mean embedding may indicate a smaller variance or a larger variance, depending on the properties of the data distributions. Hence, when using these kernels for distribution closeness testing, mitigating the issue (i.e., MMD being the same for multiple pairs of distributions with different norms in the same RKHS) by incorporating norms of distributions becomes more challenging, potentially leading to a more complex distance design.

## D DETAILS OF OUR EXPERIMENTS

### D.1 DETAILS OF EXPERIMENTS WITH DISTRIBUTIONS OVER IDENTICAL DOMAIN

Let $\mathbb{P}_n = \{p_1, p_2, ..., p_n\}$ and $\mathbb{Q}_n = \{q_1, q_2, ..., q_n\}$ be two discrete distributions over the same domain $Z = \{\boldsymbol{z}_1, \boldsymbol{z}_2, ..., \boldsymbol{z}_n\} \subseteq \mathbb{R}^d$ such that $\sum_{i=1}^n p_i = 1$ and $\sum_{i=1}^n q_i = 1$. We define the total variation [37] of $\mathbb{P}_n$ and $\mathbb{Q}_n$ as

$$\mathrm{TV}(\mathbb{P}_n, \mathbb{Q}_n) = \sup_{S \subseteq Z} (\mathbb{P}_n(S) - \mathbb{Q}_n(S)) = \frac{1}{2} \sum_{i=1}^n |p_i - q_i| = \frac{1}{2} \|\mathbb{P}_n - \mathbb{Q}_n\|_1 \in [0, 1] .$$

As we can see, the corresponding NAMMD distance can be calculated as

$$
\begin{aligned}
\mathrm{NAMMD}(\mathbb{P}_n, \mathbb{Q}_n, \kappa) &= \frac{\|\boldsymbol{\mu}_{\mathbb{P}_n} - \boldsymbol{\mu}_{\mathbb{Q}_n}\|_{\mathcal{H}_\kappa}^2}{4K - \|\boldsymbol{\mu}_{\mathbb{P}_n}\|_{\mathcal{H}_\kappa}^2 - \|\boldsymbol{\mu}_{\mathbb{Q}_n}\|_{\mathcal{H}_\kappa}^2} \\
&= \frac{\sum_{i,j} p_i p_j \kappa(\boldsymbol{z}_i, \boldsymbol{z}_j) + q_i q_j \kappa(\boldsymbol{z}_i, \boldsymbol{z}_j) - 2 p_i q_j \kappa(\boldsymbol{z}_i, \boldsymbol{z}_j)}{4K - \sum_{i,j} (p_i p_j \kappa(\boldsymbol{z}_i, \boldsymbol{z}_j) + q_i q_j \kappa(\boldsymbol{z}_i, \boldsymbol{z}_j))} .
\end{aligned}
$$

Here, we take the uniform distribution $\mathbb{P}_n = \{1/n, 1/n, ..., 1/n\}$ over sample $Z$, where $p_i = 1/n$ for $i \in \{1, 2, ..., n\}$. We construct discrete distribution $\mathbb{Q}_n$, which is $\epsilon' \in [0, 1]$ total variation away from the uniform distribution $\mathbb{P}_n$, as follows: We initiate the $\mathbb{Q}_n = \mathbb{P}_n$ and randomly split the sample $Z$ into two parts. In the first part, we increase the sample probability of each element by $\epsilon'/n$; and in the second part, we decrease the sample probability of each element by $\epsilon'/n$.

Under null hypothesis $H_0' : \mathrm{TV}(\mathbb{P}_n, \mathbb{Q}_n) = \epsilon'$, we set testing threshold $\tau_\alpha'$ as the $(1 - \alpha)$-quantile of the estimated null distribution of our NAMMD distance by resampling method, which repeatedly re-computing the empirical estimator of distance with the samples randomly drawn from $\mathbb{P}_n$ and $\mathbb{Q}_n$.

Specifically, denote by $B$ the iteration number of resampling method. In $b$-th iteration ($b \in [B]$), we randomly draw two samples $X$ and $X'$ from $\mathbb{P}_n$, and two samples $Y$ and $Y'$ from $\mathbb{Q}_n$. The sample sizes are set to be the same as the size of testing samples. Denote by $X_i$ and $X_i'$ the occurrences of $\boldsymbol{z}_i$ in samples $X$ and $X'$ respectively, and let $Y_i$ and $Y_i'$ be the occurrences of $\boldsymbol{z}_i$ in samples $Y$ and $Y'$ respectively. We then calculate the test statistic based on total variation given in Canonne's test as

$$T_b' = \sum_{i=1}^n \frac{(X_i - Y_i)^2 - X_i - Y_i}{\widehat{f}_i},$$

with the term

$$\widehat{f}_i := \max \left\{ |X_i' - Y_i'|, X_i' + Y_i', 1 \right\} .$$

During such process, we obtain $B$ statistics $T_1', T_2', ..., T_B'$ and set testing threshold as

$$\tau_\alpha' = \arg\min_\tau \left\{ \sum_{b=1}^B \frac{\mathbb{I}[T_b' \le \tau]}{B} \ge 1 - \alpha \right\} .$$

### D.2 DETAILS OF EXPERIMENTS WITH DISTRIBUTIONS OVER DIFFERENT DOMAINS

---

**Algorithm 2** Construction of distribution

---

**Input**: Two samples $Z$ and $Z'$, a kernel $\kappa$, step size $\eta$
**Output**: Two samples $Z$ and $Z'$
1: **for** NAMMD$(\mathbb{P}, \mathbb{Q}, \kappa) \neq \epsilon$ **do**
2:     Calculate the objective value $\mathcal{L}(Z, Z' \mid \kappa)$ according to Eqn. 11
3:     Calculate gradient $\nabla \mathcal{L}(Z, Z' \mid \kappa)$
4:     Gradient descend with step size $\eta$ by the Adam method
5: **end for**

---

Let $\mathbb{P}$ and $\mathbb{Q}$ be discrete uniform distributions over $Z = \{\boldsymbol{z}_i\}_{i=1}^m$ and $Z' = \{\boldsymbol{z}_i'\}_{i=1}^m$, respectively. As we can see, our NAMMD distance can be calculated as

$$
\begin{aligned}
\text{NAMMD}(\mathbb{P}, \mathbb{Q}, \kappa) &= \frac{\|\boldsymbol{\mu}_{\mathbb{P}} - \boldsymbol{\mu}_{\mathbb{Q}}\|_{\mathcal{H}_\kappa}^2}{4K - \|\boldsymbol{\mu}_{\mathbb{P}}\|_{\mathcal{H}_\kappa}^2 - \|\boldsymbol{\mu}_{\mathbb{Q}}\|_{\mathcal{H}_\kappa}^2} \\
&= \frac{1/m^2 \sum_{i,j} \kappa(\boldsymbol{z}_i, \boldsymbol{z}_j) + \kappa(\boldsymbol{z}_i', \boldsymbol{z}_j') - 2\kappa(\boldsymbol{z}_i, \boldsymbol{z}_j')}{4K - 1/m^2 \sum_{i,j} \left( \kappa(\boldsymbol{z}_i, \boldsymbol{z}_j) + \kappa(\boldsymbol{z}_i', \boldsymbol{z}_j') \right)} .
\end{aligned}
$$

Notably, NAMMD$(\mathbb{P}, \mathbb{Q}, \kappa) = 0$ can be effortlessly achieved by setting $Z = Z'$.

Here, we learn samples $Z$ and $Z'$ given NAMMD$(\mathbb{P}, \mathbb{Q}, \kappa) = \epsilon$ as follows

$$
\mathcal{L}(Z, Z' \mid \kappa) = (\text{NAMMD}(\mathbb{P}, \mathbb{Q}, \kappa) - \epsilon)^2 \tag{11}
$$

We take gradient method [70] for the optimization of Eqn. 11. Algorithm 2 presents the detailed description on optimization. The corresponding calculation of MMD$(\mathbb{P}, \mathbb{Q}, \kappa)$ is given as follows

$$
\begin{aligned}
\text{MMD}(\mathbb{P}, \mathbb{Q}, \kappa) &= \|\boldsymbol{\mu}_{\mathbb{P}} - \boldsymbol{\mu}_{\mathbb{Q}}\|_{\mathcal{H}_\kappa}^2 \\
&= 1/m^2 \sum_{i,j} \kappa(\boldsymbol{z}_i, \boldsymbol{z}_j) + \kappa(\boldsymbol{z}_i', \boldsymbol{z}_j') - 2\kappa(\boldsymbol{z}_i, \boldsymbol{z}_j') .
\end{aligned}
$$

### D.3 DETAILS OF STATE-OF-THE-ART TWO-SAMPLE TESTING METHODS

The details of six state-of-the-art two-sample testing methods used in the experiments (which are summarized in Figure 2) for test power comparison.

- MMDFuse: A fusion of MMD with multiple Gaussian kernels via a soft maximum [33];
- MMD-D: MMD with a learnable Deep kernel [27];
- MMDAgg: MMD with aggregation of multiple Gaussian kernels and multiple testing [32];
- AutoTST: Train a binary classifier of AutoML with a statistic about class probabilities [55];
- ME$_{\text{MaBiD}}$: Embeddings over multiple test locations and multiple Mahalanobis kernels [30];
- ACTT: MMDAgg with an accelerated optimization via compression [66].

### D.4 DETAILS OF DIFFERENT KERNELS

The details of the various kernels used in the experiments (which are summarized in Table 1) for test power comparison in two-sample testing, employing the same kernel for NAMMD and MMD.

- Gaussian: $\text{G}(\boldsymbol{x}, \boldsymbol{y}) = \exp(-\|\boldsymbol{x} - \boldsymbol{y}\|^2 / 2\gamma^2)$ for $\gamma > 0$ [67];
- Laplace: $\text{L}(\boldsymbol{x}, \boldsymbol{y}) = \exp(-\|\boldsymbol{x} - \boldsymbol{y}\|_1 / \gamma)$ for $\gamma > 0$ [33];
- Deep: $\text{D}(\boldsymbol{x}, \boldsymbol{y}) = [(1 - \lambda)\text{G}(\phi_\omega(\boldsymbol{x}), \phi_\omega(\boldsymbol{y})) + \lambda\text{G}(\boldsymbol{x}, \boldsymbol{y})]$ for $\lambda > 0$ and network $\phi_\omega$ [27];
- Mahalanobis: $\text{M}(\boldsymbol{x}, \boldsymbol{y}) = \exp\left(-(\boldsymbol{x} - \boldsymbol{y})^T M (\boldsymbol{x} - \boldsymbol{y}) / 2\gamma^2\right)$ for $\gamma > 0$ and $M \succ 0$ [30].

**Table 3:** Confidence and accuracy margins between the original ImageNet and its variants.

|  | ImageNetsk | ImageNetr | ImageNetv2 | ImageNeta |
|---|---|---|---|---|
| Accuracy Margin | 0.529 | 0.564 | 0.751 | 0.827 |
| Confidence Margin | 0.504 | 0.549 | 0.684 | 0.764 |

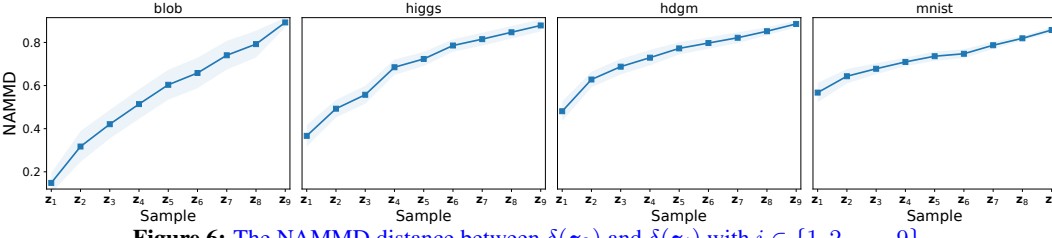

**Figure 6:** The NAMMD distance between $\delta(\boldsymbol{z}_0)$ and $\delta(\boldsymbol{z}_i)$ with $i \in \{1, 2, \ldots, 9\}$.

### D.5    DETAILS OF CONFIDENCE AND ACCURACY MARGINS

We can test the confidence margin between source dataset $S$ and target dataset $T$ for a model $f$. Let $f(x)$ represent the probability assigned by the model $f$ to the true label. We define the confidence margin as

$$|E_{\boldsymbol{x} \in S}[1 - f(\boldsymbol{x})] - E_{\boldsymbol{x} \in T}[1 - f(\boldsymbol{x})]| \,. \tag{12}$$

A smaller margin indicates similar model performance in the source and target dataset.

In a similar manner, we can also define the accuracy margin as follows

$$|E_{\boldsymbol{x} \in S}[f(\boldsymbol{x}; y_{\boldsymbol{x}})] - E_{\boldsymbol{x} \in T}[f(\boldsymbol{x}; y_{\boldsymbol{x}})]| \,,$$

where $f(\boldsymbol{x}; y_{\boldsymbol{x}}) = 1$ if the model $f$ correctly predicts the true label $y_{\boldsymbol{x}}$, and $f(\boldsymbol{x}; y_{\boldsymbol{x}}) = 0$ otherwise.

We present the confidence and accuracy margins between the original ImageNet and its variants in Table 3, with the values computed using the pre-trained ResNet50 model.

### D.6    MORE EXPERIMENTS

We demonstrate that our NAMMD better captures the differences between distributions by exploiting intrinsic structures. For each dataset, we sample ten elements and randomly selecting one element to serve as the base $\boldsymbol{z}_0$. The remaining elements are sorted as $\boldsymbol{z}_1, \boldsymbol{z}_2, ..., \boldsymbol{z}_9$ with $\|\boldsymbol{z}_0 - \boldsymbol{z}_1\|^2 \geq \|\boldsymbol{z}_0 - \boldsymbol{z}_2\|^2 \geq \cdots \geq \|\boldsymbol{z}_0 - \boldsymbol{z}_9\|^2$. For each element $\boldsymbol{z}_i$, we construct the Dirac distribution $\delta_{\boldsymbol{z}_i}$ with support only at element $\boldsymbol{z}_i$, and we calculate the distance NAMMD$(\delta_{\boldsymbol{z}_0}, \delta_{\boldsymbol{z}_i}, \kappa)$. We repeat this 10 times, using a Gaussian kernel with $\gamma = 1$ for blob, higgs, and hdgm, and $\gamma = 10$ for mnist.

From Figure 6, it is evident that our NAMMD$(\delta_{\boldsymbol{z}_0}, \delta_{\boldsymbol{z}_i}, \kappa)$ distance increases as $\|\boldsymbol{z}_0 - \boldsymbol{z}_i\|^2$ decrease for all datasets. This is different from previous total variation TV$(\delta_{\boldsymbol{z}_0}, \delta_{\boldsymbol{z}_i}) = 1$ for $i \in \{1, 2, ..., 9\}$, which merely measures the difference between probability mass functions of two distributions. In comparison, our NAMMD distance can effectively capture intrinsic structures and complex patterns in real-word datasets by leveraging kernel trick.

To compare our NAMMD test and original MMD test in distribution closeness testing, we first *select the kernel $\kappa$* based on the original distribution pair $(\mathbb{P}, \mathbb{Q})$ of the dataset, following the two-sample testing approach [27]. Notably, as analyzed in Appendix B.10, NAMMD and MMD share the same optimal kernel under two-sample testing for a fixed distribution pair. Following the setup in Definition 11, we construct two pairs of distributions: $\mathbb{P}_1$ and $\mathbb{Q}_1$, and $\mathbb{P}_2$ and $\mathbb{Q}_2$, where NAMMD$(\mathbb{P}_1, \mathbb{Q}_1, \kappa) = \epsilon$ and NAMMD$(\mathbb{Q}_2, \mathbb{P}_2, \kappa) = \epsilon + 0.01$, and MMD$(\mathbb{P}_1, \mathbb{Q}_1, \kappa) < $ MMD$(\mathbb{Q}_2, \mathbb{P}_2, \kappa)$. Specifically, we draw two sets of 500 elements from dataset, denoted as $Z = \{\boldsymbol{z}_i\}_{i=1}^{500}$ and $Z' = \{\boldsymbol{z}_i'\}_{i=1}^{500}$. Let $\mathbb{P}_1$ and $\mathbb{Q}_1$ be uniform distributions over $Z$ and $Z'$. We then optimize $Z$ and $Z'$ by gradient method [70] to ensure that NAMMD$(\mathbb{P}_1, \mathbb{Q}_1, \kappa) = \epsilon$. It is straightforward to calculate MMD$(\mathbb{P}_1, \mathbb{Q}_1, \kappa)$ and to construct $\mathbb{P}_2$ and $\mathbb{Q}_2$ in a similar manner. The details of construction are provided in Appendix D.2.

For comparison, we set $\epsilon \in \{0.1, 0.3, 0.5, 0.7\}$. We randomly draw two samples from $\mathbb{Q}_2$ and $\mathbb{P}_2$ to evaluate test power of tests. Table 4 summarizes the average test powers and standard deviations of

**Table 4:** Comparisons of test power (mean±std) on distribution closeness testing with respect to different NAMMD values, and the bold denotes the highest mean between tests with our NAMMD and original MMD. Notably, the same selected kernel is applied for both NAMMD and MMD in this table.

| Dataset | $\epsilon = 0.1$ MMD | $\epsilon = 0.1$ NAMMD | $\epsilon = 0.3$ MMD | $\epsilon = 0.3$ NAMMD | $\epsilon = 0.5$ MMD | $\epsilon = 0.5$ NAMMD | $\epsilon = 0.7$ MMD | $\epsilon = 0.7$ NAMMD |
|---|---|---|---|---|---|---|---|---|
| blob | .974±.009 | **.978**±**.008** | .890±.030 | **.923**±**.025** | .902±.032 | **.924**±**.021** | .909±.024 | **.933**±**.011** |
| higgs | .998±.002 | **.999**±**.001** | .938±.020 | **.965**±**.013** | .975±.012 | **.993**±**.003** | .978±.010 | **.996**±**.002** |
| hdgm | .980±.007 | **.984**±**.007** | .883±.027 | **.921**±**.021** | .901±.025 | **.941**±**.013** | **1.00**±**.000** | **1.00**±**.000** |
| mnist | **.982**±**.004** | **.982**±**.004** | .961±.006 | **.974**±**.004** | .946±.014 | **.983**±**.005** | .962±.010 | **.991**±**.003** |
| cifar10 | .932±.007 | **.938**±**.007** | .968±.019 | **.994**±**.003** | .898±.054 | **.912**±**.041** | **1.00**±**.000** | **1.00**±**.000** |
| Average | .973±.006 | **.976**±**.005** | .928±.020 | **.955**±**.013** | .924±.027 | **.951**±**.017** | .970±.009 | **.984**±**.003** |

**Table 5:** Comparisons of test power (mean±std) on distribution closeness testing with respect to different NAMMD values, and the bold denotes the highest mean between tests with our NAMMD and original MMD. Notably, different selected kernel are applied for NAMMD and MMD respectively in this table.

| Dataset | $\epsilon = 0.1$ MMD | $\epsilon = 0.1$ NAMMD | $\epsilon = 0.3$ MMD | $\epsilon = 0.3$ NAMMD | $\epsilon = 0.5$ MMD | $\epsilon = 0.5$ NAMMD | $\epsilon = 0.7$ MMD | $\epsilon = 0.7$ NAMMD |
|---|---|---|---|---|---|---|---|---|
| blob | .939±.009 | **.983**±**.004** | .968±.007 | **.991**±**.002** | .952±.010 | **.999**±**.001** | .934±.010 | **1.00**±**.000** |
| higgs | .914±.051 | **.972**±**.009** | .934±.056 | **.976**±**.007** | .967±.021 | **.994**±**.002** | .949±.036 | **.1.00**±**.000** |
| hdgm | .925±.071 | **.976**±**.005** | .915±.069 | **.978**±**.004** | .913±.058 | **.984**±**.004** | .938±.052 | **1.00**±**.000** |
| mnist | .951±.006 | **.962**±**.005** | .955±.032 | **.961**±**.021** | .935±.049 | **.967**±**.036** | .977±.011 | **.992**±**.002** |
| cifar10 | .976±.012 | **.987**±**.006** | .971±.007 | **.988**±**.003** | .991±.004 | **1.00**±**.000** | **1.00**±**.000** | **1.00**±**.000** |
| Average | .941±.030 | **.976**±**.006** | .949±.034 | **.979**±**.007** | .952±.028 | **.989**±**.009** | .960±.022 | **.998**±**.000** |

our NAMMD distance and original MMD distance in distribution closeness testing for distributions over different domains. It is evident that our NAMMD test achieves better performances than the original MMD test with respect to different datasets, and this improvement is achieved through scaling with the norms of mean embeddings of distributions according to Theorem 12.

In a similar manner, we conduct the experiments in Table 4, but with different selected kernels for NAMMD and MMD. For MMD, the kernel selection remains the same as in the experiments in Table 4, and we denote the kernel for MMD as $\kappa^{\mathrm{M}}$. However, for NAMMD, we select the kernel $\kappa^{\mathrm{N}}$ similar to the experiments in Table 4, but with an additional regularization term related to the norms of the original distributions in the dataset (i.e., $4K - \|\boldsymbol{\mu}_{\mathbb{P}}\|^2_{\mathcal{H}_\kappa} - \|\boldsymbol{\mu}_{\mathbb{Q}}\|^2_{\mathcal{H}_\kappa}$) during the optimization. Notably, these kernel selection methods are heuristic for distribution closeness testing, as obtaining a test power estimator for DCT with multiple distribution pairs and selecting an optimal global kernel for DCT based on the estimator remain open questions and poses a significant challenge. We use $\kappa^{\mathrm{N}}$ for the construction distribution pairs $(\mathbb{P}_1, \mathbb{Q}_1)$ and $(\mathbb{P}_2, \mathbb{Q}_2)$. Following Definition 11, we perform NAMMD DCT with $\kappa^{\mathrm{N}}$ and MMD DCT with $\kappa^{\mathrm{M}}$ respectively. Table 5 summarizes the average test powers and standard deviations of NAMMD DCT and MMD DCT. It is evident that our NAMMD test achieves better performance than the MMD test, and this improvement when using different selected kernels for NAMMD and MMD can be explained by the conjunction analysis for DCT in Appendix B.10 based on Theorem 12.

**Type-I Error Experiments** From Figure 7, it is evident that the Type-I error of our NAMMD test is limited about $\alpha = 0.05$ with respect to different kernels and datasets in two-sample testing (i.e. distribution closeness testing with $\epsilon = 0$) by using permutation tests. In a similar manner, Figure 8 shows that the Type-I error of our NAMMD test is limited about $\alpha = 0.05$ with respect to different $\epsilon \in (0, 1)$ and datasets in distribution closeness testing, where we derive the testing threshold based on asymptotic distribution. These results are nicely in accordance with Theorem 5.

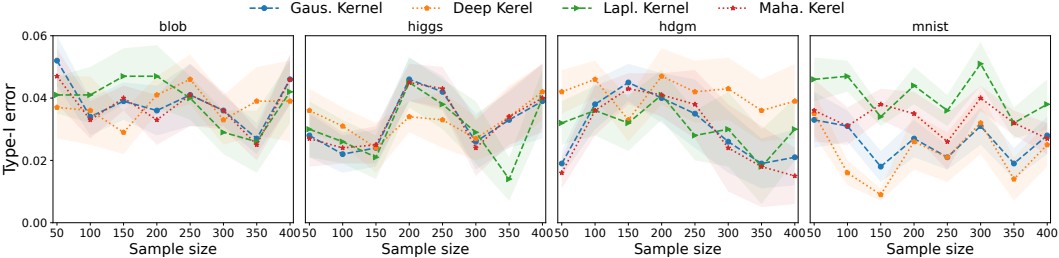

**Figure 7:** The Type-I error is limited about $\alpha = 0.05$ w.r.t different kernels for our NAMMD test with $\epsilon = 0$.

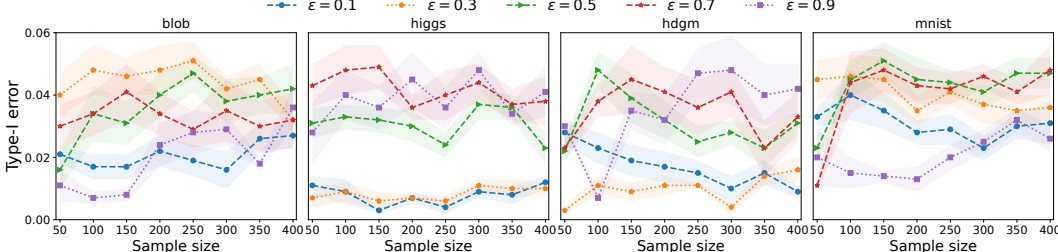

**Figure 8:** The Type-I error is limited about $\alpha = 0.05$ w.r.t different $\epsilon \in (0, 1)$ for our NAMMD test.

