# OpenReview forum: "A Kernel Distribution Closeness Testing"
_ICLR.cc/2025/Conference — Submitted to ICLR 2025_

### Official Review · Reviewer_8GF1 · 2024-11-02

**Soundness:** 3
**Presentation:** 3
**Contribution:** 3
**Rating:** 8
**Confidence:** 3

**Summary:**

This paper proposes an improved Maximum Mean Discrepancy (MMD) method, namely norm-adaptive maximum mean discrepancy (NAMMD), for testing distribution similarity. By incorporating hypothesis testing, this method introduces a practical approach for distribution closeness testing in real-world scenarios.

**Strengths:**

Rigorous theoretical analysis provides reliability for the paper.
A comprehensive background description makes it easy for readers to understand the problem the paper aims to address.

**Weaknesses:**

The author does not explain why the definition of NAMMD is formatted as Equation (1). Although the author proves the effectiveness of NAMMD from the perspectives of complexity, hypotheses testing power, and closeness testing power in sections 4.1, 4.2, and 4.3, the fundamental motivation for designing NAMMD is not clearly described.

**Questions:**

As the author stated, MMD has an inherent issue in closeness testing: "the same MMD value may reflect different levels of closeness between distributions, which makes it less informative." Why doesn't the author redesign a kernel-based closeness testing method? Such approach seems to be able to circumvent the inherent problems of MMD and further address the closeness testing of high-dimensional data.

Why the definition of NAMMD is formatted as Equation (1) and sufficient to achieve the desired goal? This point lacks explanation. After defining NAMMD, the authors do not provide a corresponding justification.

---

> ### Author Response · Authors · 2024-11-20
>
> Thank you for reading our paper and for your insightful comments. We provide more explanations of the definition of our NAMMD below.
>
> >[Q1] The author does not explain why the definition of NAMMD is formatted as Equation (1). Although ..., the fundamental motivation for designing NAMMD is not clearly described. Why the definition of NAMMD is formatted as Equation (1) and sufficient to achieve the desired goal? This point lacks explanation. After defining NAMMD, the authors do not provide a corresponding justification.
>
> [A1] Thank your for pointing out this concern about our fundamental motivation. We will add a **Remark** immediately after the definition of NAMMD in Equation (1), providing the detailed explanation outlined as follows.
>
> In our NAMMD, we aims to capture differences between two distributions based on their kernel mean embeddings (i.e. $\mu_P$ and $\mu_Q$), leveraging the property that a kernel mean embedding uniquely represents a probability distribution and capture unique characteristics to support efficient distribution comparisons [Sriperumbudur et al., 2011]. To measure the difference, a natural way is to calculate the distancen between two mean embeddings (i.e. $||\mu_P-\mu_Q||^2_H$), which is the MMD distance. However, we find that the MMD can yield same values for many pairs of distributions (i.e. $P$ and $Q$) that have different norms (i.e. $||\mu_P||^2_H$ and $||\mu_Q||^2_H$), which actually indicates different levels of closeness. For example, in Figure 1a and 1b, distribution pair ($P_1,Q_1$) shares the same MMD distance as ($P_2, Q_2$) but has different norms. Larger norms indicate more tightly concentrated distributions, leading to smaller standard deviations and $p$-values of the empirical MMD estimator as shown in Figure 1c, even with a constance MMD value $0.15$. Notably, smaller $p$-values indicate more significant difference and less closeness between distributions. Hence, the MMD fails to distinguish the closeness levels for the distribution pairs in Figures 1a and 1b. To mitigate this issue, we scale the MMD distance by incorporating the norms information $4K-||\mu_P||^2_H-||\mu_Q||^2_H$. It is evident that the NAMMD value scaled up as norms ($||\mu_P||^2_H+||\mu_Q||^2_H$) increase, which aligns with the fact that larger norms indicates larger distance between distributions (when MMD value remains constant). Figure 1c and 1d demonstrate that our NAMMD exhibits a stronger correlation with the $p$-value (i.e. the level of closeness), while the MMD value is held constant.
>
> We further demonstrate the advantages of our NAMMD over the original MMD in Theorems 9 and 11, where the improvement in test power is achieved through scaling with $4K - ||\mu_P||^2_H - ||\mu_Q||^2_H$.
>
> >[Q2] As the author stated, MMD has an inherent issue in closeness testing: "the same MMD value may reflect different levels of closeness between distributions, which makes it less informative." Why doesn't the author redesign a kernel-based closeness testing method? Such approach seems to be able to circumvent the inherent problems of MMD and further address the closeness testing of high-dimensional data.
>
> [A2] Thanks for pointing out this interesting Quetion. The key issue identified in this paper (i.e. the "less informative" of MMD) arises because MMD yields same values for many pairs of distributions that have different norms of mean embeddings (which results in different levels of closeness). By incorporating norm of mean embeddings in our NAMMD, we effectively mitigate this issue. In designing a new kernel-based closeness testing method, the mean embedding of a distribution is crucial, as it is injective for characteristic kernels, ensuring that different distributions have distinct embeddings and capturing unique characteristics to support efficient comparison for distributions [Sriperumbudur et al., 2011]. The MMD directly measures the Euclidean-like distance between mean embeddings, and its simplicity and well-established theoretical properties make it a reliable choice for various applications [Gretton et al., 2012]. Hence, our NAMMD builds on the importance of mean embeddings and strengths of MMD, enhancing the statistic's effectiveness in distribution closeness testing by incorporating additional information from the mean embeddings.
>
> We genuinely appreciate your valuable insights. While we have made efforts to address your concerns, we acknowledge that there may still be aspects requiring further clarification. We would be delighted to engage in further discussions to ensure your concerns are fully addressed.

---

> ### Author Response · Authors · 2024-11-22
>
> Thank you for your valuable comments. We have submitted our response to address your concerns and answer your questions. We would greatly appreciate it if you could let us know whether our response has resolved your concerns, and we look forward to further discussing this with you.

---

> ### Author Response · Authors · 2024-11-24
> **Reminder - Discussion Stage Closing Soon - 24 November**
>
> Dear Reviewer 8GF1,
>
> Thank you for taking the time and effort to review our manuscript.
>
> We have carefully addressed all your comments and prepared detailed responses. Could you kindly review them at your earliest convenience?
>
> We hope our responses have satisfactorily addressed your key concerns. If anything remains unclear or requires further clarification, please do not hesitate to let us know, and we will address it promptly.
>
> We look forward to your feedback.
>
> Best regards,
>
> Authors of Submission 6378

---

> ### Author Response · Authors · 2024-11-25
> **Reminder - Discussion Stage Closing Soon - 26 November**
>
> Dear Reviewer 8GF1,
>
> Thank you again for your time and effort in reviewing our paper.
>
> We have carefully addressed all your comments and prepared detailed responses. Could you kindly review them at your earliest convenience?
>
> As the rebuttal period is coming to a close, we wanted to check if there are any remaining concerns preventing you from adjusting your score. If there is any additional clarification we can provide, please do not hesitate to let us know and, we will address it promptly.
>
> Best regards,
>
> Authors of Submission 6378

---

> > ### Comment · Reviewer_8GF1 · 2024-11-26
> >
> > Apologies for the wait. Your response has effectively addressed my doubts. I appreciate the merit of this work and am willing to raise the score.

---

> > > ### Author Response · Authors · 2024-11-26
> > > **Thanks for your support and increasing your score to 8!**
> > >
> > > Dear Reviewer 8GF1,
> > >
> > > Glad to hear that your concerns are effectively addressed. Many thanks for your support and increasing your score to 8!
> > >
> > > Best regards,
> > >
> > > Authors of Submission 6378

---

### Official Review · Reviewer_u8Wh · 2024-11-03

**Soundness:** 3
**Presentation:** 3
**Contribution:** 3
**Rating:** 6
**Confidence:** 3

**Summary:**

This paper proposed the norm-adaptive MMD as a testing statistic for distribution closeness testing. Compared with the MMD statistic, it scales MMD with the norm of distributions in RKHS. Theoretical analysis and numerical study demonstrate the superior performance of this framework.

**Strengths:**

1. The paper is well-written, with explicit technical assumptions and technical proofs included.
2. The numerical study is solid in justifying the good performance of this framework.
3. Based on the theoretical analysis and numerical study, I am convinced that NAMMD is a reasonable framework.

**Weaknesses:**

1. The threshold plays a key role in hypothesis testing. Under the null hypothesis test, the authors utilize the permutation test to estimate the threshold, which is commonly used in literature. Under the case where $NAMMD(P,Q)=\epsilon$, the authors estimate it based on the variance of asymptotic distribution, which further approximates it using empirical variance estimator $\sigma_{X,Y}$. Can the authors provide more theoretical analysis regarding this estiamtor (such as bias, variance, etc?) to justify the soundness of this approximation?
2. Can the authors elaborate more on how to modify NAMMD in the fusing statistics approach?

**Questions:**

N/A

---

> ### Author Response · Authors · 2024-11-20
>
> Thank you for reading our paper and for your insightful comments. We provide further theoretical analysis of the variance estimator and details of the NAMMDFuse below.
>
> >[Q1] The threshold plays a key role in hypothesis testing. .... Under the case where the authors estimate it based on the variance of the asymptotic distribution, ... using the empirical variance estimator $\sigma_{X,Y}$. Can the authors provide more theoretical analysis regarding this estimator...?
>
> [A1] Thank you for your concern. We present that for our empirical variance estimator $\sigma_{X,Y}$ with sample size $m$, the following holds:
> $$
> |E[\sigma^2_{X,Y}] - \sigma^2_{P,Q}|=O\left(\frac{1}{\sqrt{m}}\right)
> $$
>
> **Proof Sketch:**
> For simplicity, we denote:
> $$
> \hat{A}=\sqrt{\frac{(4m-8)\zeta_1+2\zeta_2}{m-1}}, \quad A=\sqrt{4E[H_{1,2}H_{1,3}]-4(E[H_{1,2}])^2}
> $$
> and
> $$
> \hat{B}=\frac{1}{m^2-m}\sum_{i\neq j }\left(4K-\kappa(\mathbf{x}_i,\mathbf{x}_j)-\kappa(\mathbf{y}_i,\mathbf{y}_j)\right), \quad B=4K-||\mathbf{\mu}_P||_H^2-||\mathbf{\mu}_Q||_H^2.
> $$
>
> Hence, we establish the equivalence: $\sigma^2_{X,Y} =\hat{A}^2/\hat{B}^2$ and $\sigma^2_{P,Q}=A^2/B^2$. Building on this, we further investigate that:
>
> $$
> \left|E\left[\frac{\hat{A}^2}{\hat{B}^2}\right] - \frac{A^2}{B^2}\right| = \left|E\left[\frac{\hat{A}^2}{\hat{B}^2}\right]-E\left[\frac{\hat{A}^2}{B^2}\right]+E\left[\frac{\hat{A}^2}{B^2}\right] - \frac{A^2}{B^2}\right|
> $$
> Breaking it down:
> $$
> \left|E\left[\frac{\hat{A}^2}{\hat{B}^2}\right] - \frac{A^2}{B^2}\right| \leq \left|E\left[\frac{\hat{A}^2}{\hat{B}^2}\right]-E\left[\frac{\hat{A}^2}{B^2}\right]\right|+\left|E\left[\frac{\hat{A}^2}{B^2}\right]-\frac{A^2}{B^2}\right| = \left|E\left[\frac{\hat{A}^2}{\hat{B}^2}\right]-E\left[\frac{\hat{A}^2}{B^2}\right]\right|
> $$
> The second equality relies on the unbiased variance estimator of the U-statistic, i.e., $\hat{A}$. We further have the following inequality:
> $$
> E\left[\left|\frac{\hat{A}^2}{\hat{B}^2}-\frac{\hat{A}^2}{B^2}\right|\right] = E\left[\left|\frac{\hat{A}^2(B-\hat{B})(B+\hat{B})}{\hat{B}^2B^2}\right|\right]\leq C \cdot E\left[\left|B-\hat{B}\right|\right]\ ,
> $$
> where $C>0$ is a constant that ensures $\frac{\hat{A}^2(B+\hat{B})}{\hat{B}^2B^2} \leq C$, and it exists since the kernel is bounded.
>
> Based on the large deviation bound for $B$, we have:
> $$
> \Pr\left(\left|B-\hat{B}\right|\geq t\right) \leq 2\exp{(-mt^2/4K^2)}.
> $$
> Integrating this probability:
> $$
> C \cdot E\left[\left|B-\hat{B}\right|\right] = C \cdot \int_0^\infty \Pr\left(\left|B-\hat{B}\right|\geq t\right) dt\leq C \cdot \int_0^\infty 2\exp{(-mt^2/4K^2)}dt
> $$
> Using substitution $u = mt^2/4K^2$:
> $$
> C \cdot E\left[\left|B-\hat{B}\right|\right]\leq C \cdot \int_0^\infty 2\exp{(-u)}\frac{K}{\sqrt{m}\sqrt{u}}du= C \cdot \frac{2K\sqrt{\pi}}{\sqrt{m}} = O\left(\frac{1}{\sqrt{m}}\right).
> $$
>
> >[Q2] Can the authors elaborate more on how to modify NAMMD in the fusing statistics approach?
>
> [A2] Thanks for pointing our this quesiton. We will include more details about NAMMDFuse in our revision. Following the fusing statistics approach of [1], we introduce the NAMMDFuse statistic through exponentiation of NAMMD with samples $X$ and $Y$ as follows
> $$
> \widehat{\textnormal{FUSE}}(X,Y)=\frac{1}{\lambda}\log\left(E_{\kappa\sim\pi(\langle X,Y\rangle)}\left[\exp\left(\lambda\frac{\widehat{\textnormal{NAMMD}}(X,Y,\kappa)}{\sqrt{\widehat{N}(X,Y)}}\right)\right]\right)
> $$
> where $\lambda>0$ and $\widehat{N}(X,Y)=\frac{1}{m(m-1)}\sum_{i\neq j}^m \kappa(\mathbf{x}_i,\mathbf{x}_j)^2+\kappa(\mathbf{y}_i,\mathbf{y}_j)^2$ is permutation invariant. $\pi(\langle X,Y\rangle)$ is the prior distribution on the kernel space $\mathcal{K}$. In experiments, we set the prior distribution $\pi(\langle X,Y\rangle)$ and the kernel space $\mathcal{K}$ to be the same for MMDFuse.
>
> [1] Biggs, F., Schrab, A., & Gretton, A. (2024). MMD-FUSE: Learning and combining kernels for two-sample testing without data splitting. Advances in Neural Information Processing Systems, 36.
>
> We genuinely value your insights and would be delighted to engage in further discussions with you regarding your concern.

---

> ### Author Response · Authors · 2024-11-22
>
> Thank you for your thoughtful comments. We have submitted our response to address your concerns and questions. We would greatly appreciate it if you could confirm whether our response resolves your concerns. We look forward to your feedback and further discussion.

---

> ### Author Response · Authors · 2024-11-24
> **Reminder - Discussion Stage Closing Soon - 24 November**
>
> Dear Reviewer u8Wh,
>
> Thank you for taking the time and effort to review our manuscript.
>
> We have carefully addressed all your comments and prepared detailed responses. Could you kindly review them at your earliest convenience?
>
> We hope our responses have satisfactorily addressed your key concerns. If anything remains unclear or requires further clarification, please do not hesitate to let us know, and we will address it promptly.
>
> We look forward to your feedback.
>
> Best regards,
>
> Authors of Submission 6378

---

> ### Author Response · Authors · 2024-11-25
> **Reminder - Discussion Stage Closing Soon - 26 November**
>
> Dear Reviewer u8Wh,
>
> Thank you again for your time and effort in reviewing our paper.
>
> We have carefully addressed all your comments and prepared detailed responses. Could you kindly review them at your earliest convenience?
>
> As the rebuttal period is coming to a close, we wanted to check if there are any remaining concerns preventing you from adjusting your score. If there is any additional clarification we can provide, please do not hesitate to let us know and, we will address it promptly.
>
> Best regards,
>
> Authors of Submission 6378

---

> ### Author Response · Authors · 2024-11-26
> **Reminder - Discussion Stage Closing Soon - 27 November**
>
> Dear Reviewer u8Wh,
>
> Thank you again for your time and effort in reviewing our paper.
>
> We have carefully addressed all your comments and prepared detailed responses. Could you kindly review them at your earliest convenience?
>
> As the discussion period is coming to a close, we wanted to check if there are any remaining concerns preventing you from adjusting your score. If there is any additional clarification we can provide, please do not hesitate to let us know, and we will address it promptly.
>
> Best regards,
>
> Authors of Submission 6378

---

> > ### Author Response · Authors · 2024-11-29
> > **Reminder - Discussion Stage Closing Soon - 29 November**
> >
> > Dear Reviewer u8Wh,
> >
> > Thank you again for your time and effort in reviewing our paper.
> >
> > We have carefully addressed all your comments and provided detailed responses. We have also revised our paper based on your suggestions. Could you kindly review them at your earliest convenience?
> >
> > As the discussion period is coming to a close, we wanted to check if there are any remaining concerns preventing you from adjusting your score. If there is any additional clarification we can provide, please do not hesitate to let us know, and we will address it promptly.
> >
> > Best regards,
> >
> > Authors of Submission 6378

---

> > > ### Author Response · Authors · 2024-11-30
> > > **Reminder - Discussion Stage Closing Soon - 30 November**
> > >
> > > Dear Reviewer u8Wh,
> > >
> > > Thank you again for your time and effort in reviewing our paper.
> > >
> > > We have carefully addressed all your comments and provided detailed responses. We have also revised our paper based on your suggestions. Could you kindly review them at your earliest convenience?
> > >
> > > As the discussion period is coming to a close, we wanted to check if there are any remaining concerns preventing you from adjusting your score. If there is any additional clarification we can provide, please do not hesitate to let us know, and we will address it promptly.
> > >
> > > Best regards,
> > >
> > > Authors of Submission 6378

---

> > > > ### Author Response · Authors · 2024-12-01
> > > > **Reminder - Discussion Stage Closing Soon - 1 December**
> > > >
> > > > Dear Reviewer u8Wh,
> > > >
> > > > Thank you again for your time and effort in reviewing our paper.
> > > >
> > > > We have carefully addressed all your comments and provided detailed responses. We have also revised our paper based on your suggestions. Could you kindly review them at your earliest convenience?
> > > >
> > > > As the discussion period is coming to a close, we wanted to check if there are any remaining concerns preventing you from adjusting your score. If there is any additional clarification we can provide, please do not hesitate to let us know, and we will address it promptly.
> > > >
> > > > Best regards,
> > > >
> > > > Authors of Submission 6378

---

> ### Author Response · Authors · 2024-12-02
> **Reminder - Discussion Stage Closing Soon - 2 December**
>
> Dear Reviewer u8Wh,
>
> Thank you again for your time and effort in reviewing our paper.
>
> We have carefully addressed all your comments and provided detailed responses. We have also revised our paper based on your suggestions. Could you kindly review them at your earliest convenience?
>
> As the discussion period is coming to a close, we wanted to check if there are any remaining concerns preventing you from adjusting your score. If there is any additional clarification we can provide, please do not hesitate to let us know, and we will address it promptly.
>
> Best regards,
>
> Authors of Submission 6378

---

> ### Author Response · Authors · 2024-12-02
> **Reminder - Discussion Stage Closing Soon - 3 December**
>
> Dear Reviewer u8Wh,
>
> Thank you again for your time and effort in reviewing our paper.
>
> We have carefully addressed all your comments and provided detailed responses. We have also revised our paper based on your suggestions. Could you kindly review them at your earliest convenience?
>
> As the discussion period is coming to a close, we wanted to check if there are any remaining concerns preventing you from adjusting your score. If there is any additional clarification we can provide, please do not hesitate to let us know, and we will address it promptly.
>
> Best regards,
>
> Authors of Submission 6378

---

> ### Author Response · Authors · 2024-12-03
> **Discussion Period Ending in Less Than 24 Hours**
>
> Dear Reviewer u8Wh,
>
> Thank you again for your valuable time and effort in reviewing our paper.
>
> With the discussion period ending in less than 24 hours, we wanted to ensure all your concerns have been addressed and check if anything remains that might prevent you from updating your score. If there are any additional clarifications or adjustments you would like us to make, please let us know, and we will address them promptly.
>
> Thank you once again for your consideration.
>
> Best regards,
> Authors of Submission 6378

---

### Official Review · Reviewer_TqLo · 2024-11-04

**Soundness:** 4
**Presentation:** 3
**Contribution:** 3
**Rating:** 6
**Confidence:** 3

**Summary:**

The paper develops a new distribution closeness testing method, called norm adaptive MMD.

MMD can be used to measure the discrepancy between two distributions. The statistical properties of MMD-based estimators are well-known, and these can be used to create closeness testing methods.

The naive MMD-based method, however, is not very informative, because, the closeness measured by MMD varies with the norms of the distributions. This implies that pairs of distributions with different variances can have the same MMD value, even though these pairs of distributions visually can be very different. To overcome this issue, the authors propose to scale the MMD values with the norms of the distributions. This new method is called Norm-Adaptive MMD (NAMMD).

**Strengths:**

* The paper is well-written and easy to follow.
* The new method is simple and the paper provides new theoretical results for the proposed NAMMD estimator.
* The paper provides experiments on five datasets, and demonstrates that it can work better than MMD and Canonne's tests.

**Weaknesses:**

The paper claims that standard closeness measures don't work well on complex, high-dimensional datasets, e.g. images.
The provided numerical experiments show that this distribution closeness works better than other hypothesis tests on benchmark datasets, but it doesn't demonstrate how this improved test can make a difference in some important real-world applications, e.g. on high-dimensional images.

**Questions:**

Would it be possible to show how the proposed distribution closeness test can make a difference in some important real-world applications?

---

> ### Author Response · Authors · 2024-11-20
>
> Thank you for reading our paper and for your insightful comments. We will add more experiments to illustrate the practical applications of our NAMMD closeness testing.
>
> >[Q] The paper claims that standard closeness measures don't work well on complex, high-dimensional datasets, e.g. images. ... but it doesn't demonstrate how this improved test can make a difference in some important real-world applications, e.g. on high-dimensional images. Would it be possible to show how the proposed distribution closeness test can make a difference in some important real-world applications?
>
> [A] Thank you for your concern. We have added additional discussions and experiments on the application of our NAMMD in real-world scenarios. In machine learning, it is empirically proved that models trained on a large dataset (e.g., ImageNet) can have good performance on relevant/similar downstream test data that is different from training dataset. This means that, even if training and test data are from different distributions, we can still expect relatively good performance because they might be close to each other. Build on this, we present a case study to illustrate how our NAMMD distribution closeness testing can be utilized to assess whether a model performs similarly across source and target domains for domain adaptation.
>
> Given the pre-trained ResNet50, which performs well in the original ImageNet dataset, we aim to evaluate how it performs on variants of ImageNet (ImageNet-SK, ImageNet-R, ImageNet-V2, ImageNet-A). A natural measure is the **accuracy margin** (i.e., the gap between the accuracies of original ImageNet and variant ImageNet) as shown in the following table.
> |                         | ImageNet-SK | ImageNet-R | ImageNet-V2 | ImageNet-A |
> |-------------------------|-----------------|----------------|-----------------|----------------|
> | Accuracy Margin    | 0.529           | 0.564          | 0.751           | 0.827          |
>
> A smaller margin indicates similar model performance in the original Imagenet and its variant, suggesting better performance on the variant ImageNet. However, obtaining **ground truth labels** for the variant ImageNet is often challenging or costly. In this case, we can evaluate the model performance based on our NAMMD closeness testing with reference distributions.
>
> From the table above, we observe that the accuracy margin of ImageNet-V2 is smaller than that of ImageNet-A, indicating that ImageNet-V2 is closer to the original ImageNet than ImageNet-A. Now, we wish to validate that our NAMMD distance can also reflect the closeness relationship and performs effectively in distribution closeness testing to identify that the ImageNet-A is farther from original ImageNet than ImageNet-V2. Following Definition 10, we set the original ImageNet as $P_1=P_2$, and set the ImageNet-V2 as $Q_1$ for **null hypothesis**. We further set ImageNet-A as $Q_2$ and perform the distribution closeness testing to assess if the distance between $P_2$ and $Q_2$ is larger than that between $P_1$ and $Q_1$. The **test powers** of our NAMMD and MMD with different sample sizes are presented in the table below.
> | Sample Size | 30  | 60  | 90  | 120 | 150 | 180 | 210 | 240 | 270 | 300 |
> |------------------|---------|---------|---------|---------|---------|---------|---------|---------|---------|---------|
> | NAMMD       | 0.102   | 0.377   | 0.636   | 0.780   | 0.887   | 0.916   | 0.962   | 0.978   | 0.987   | 0.994   |
> | MMD         | 0.080   | 0.330   | 0.561   | 0.724   | 0.843   | 0.889   | 0.939   | 0.969   | 0.973   | 0.987   |
>
> It is evident that our NAMMD distance achieves higher test power than MMD by incorporating the norm information of distributions, and effectively reflects the closeness relationship indicated by accuracy margin. Moreover, even with a limited sample size (much smaller than that of the original ImageNet and ImageNet-A), our NAMMD distance can successfully identify that the ImageNet-A is farther from original ImageNet than ImageNet-V2. We conducted similar experiments on other ImageNet variants and obtained consistent results, which are presented in the following parts.

---

> ### Author Response · Authors · 2024-11-20
>
> Similarly, we test if the ImageNet-V2 is farther from original ImageNet than ImageNet-R. Following Definition 10, we set the original ImageNet as $P_1=P_2$, and set the ImageNet-R as $Q_1$. We further set ImageNet-V2 as $Q_2$ and test if the distance between $P_2$ and $Q_2$ is larger than that between $P_1$ and $Q_1$. The **test powers** of our NAMMD and MMD with different sample sizes are presented in the table below.
> | Sample Size | 10  | 20  | 30  | 40  | 50  | 60  | 70  | 80  | 90  | 100 |
> |------------------|---------|---------|---------|---------|---------|---------|---------|---------|---------|---------|
> | NAMMD       | 0.001   | 0.250   | 0.665   | 0.873   | 0.962   | 0.989   | 0.995   | 1.000   | 1.000   | 1.000   |
> | MMD         | 0.001   | 0.220   | 0.615   | 0.840   | 0.944   | 0.987   | 0.995   | 1.000   | 1.000   | 1.000   |
>
> Similarly, we test if the ImageNet-R is farther from original ImageNet than ImageNet-SK. Following Definition 10, we set the original ImageNet as $P_1=P_2$, and set the ImageNet-SK as $Q_1$. We further set ImageNet-R as $Q_2$ and test if the distance between $P_2$ and $Q_2$ is larger than that between $P_1$ and $Q_1$. The **test powers** of our NAMMD and MMD with different sample sizes are presented in the table below.
> | Sample Size | 200 | 400 | 600 | 800 | 1000 | 1200 | 1400 | 1600 | 1800 | 2000 |
> |------------------|---------|---------|---------|---------|----------|----------|----------|----------|----------|----------|
> | NAMMD       | 0.233   | 0.424   | 0.553   | 0.684   | 0.750    | 0.851    | 0.879    | 0.911    | 0.949    | 0.966    |
> | MMD         | 0.215   | 0.397   | 0.524   | 0.661   | 0.732    | 0.834    | 0.864    | 0.890    | 0.933    | 0.954    |
>
> Similarly, we test if the ImageNet-SK is farther from original ImageNet than a slightly noised version of ImageNet. Following Definition 10, we set the original ImageNet as $P_1=P_2$, and set the slightly noised version of ImageNet as $Q_1$. We further set ImageNet-SK as $Q_2$ and test if the distance between $P_2$ and $Q_2$ is larger than that between $P_1$ and $Q_1$. The **test powers** of our NAMMD and MMD with different sample sizes are presented in the table below.
> | Sample Size | 10  | 20  | 30  | 40  | 50  | 60  | 70  | 80  | 90  | 100 |
> |------------------|---------|---------|---------|---------|---------|---------|---------|---------|---------|---------|
> | NAMMD       | 0.001   | 0.176   | 0.550   | 0.822   | 0.928   | 0.976   | 0.995   | 0.999   | 1.000   | 1.000   |
> | MMD         | 0.001   | 0.140   | 0.470   | 0.750   | 0.877   | 0.954   | 0.986   | 0.993   | 0.995   | 0.998   |
>
> In summary, our NAMMD demonstrates improved performance over the original MMD, effectively distinguishing levels of closeness as indicated by accuracy margins between the original ImageNet and its variants.
>
> &nbsp;
>
> For datasets with limited samples, the accuracy margin may be dispersed and fail to reliably capture differences in model performance. We further introduce the **confidence margin** between source domain $S$ and target domain $T$ for a model $f$. Let $f(x)$ represent the probability assigned by the model $f$ to the true label. The confidence margin is defined as $|E_{x\in S}[1-f(x)] - E_{x\in T}[1-f(x)]|$. Now, we illustrate that our NAMMD distribution closeness testing can be used to test confidence margin with limited sample sizes. Using the pre-trainded ResNet-50, we compute the confidence margin for each class separately between **ImageNet** and **ImageNetV2**. Following Definition 10, we define the classes with an average margin **0.186** in ImageNet and ImageNetV2 as $P_1$ and $Q_1$ for the **null hypothesis**. We further set $P_2$ and $Q_2$ as the classes in ImageNet and ImageNetV2 with average margins in $\\{0.154, 0.165, 0.176, 0.186, 0.196, 0.205, 0.214, 0.224, 0.233, 0.241\\}$. We perform the distribution closeness testing and the **rejection rates** of NAMMD and MMD are presented in the Table with sample size 150.
> | Confidence Margin | 0.154 | 0.165 | 0.176 | 0.186 | 0.196 | 0.205 | 0.214 | 0.224 | 0.233 | 0.241 |
> |-------------------------|-----------|-----------|-----------|-----------|-----------|-----------|-----------|-----------|-----------|-----------|
> | NAMMD              | 0.000     | 0.003     | 0.010     | 0.050     | 0.206     | 0.501     | 0.774     | 0.879     | 0.979     | 1.000     |
> | MMD                | 0.000     | 0.003     | 0.009     | 0.044     | 0.187     | 0.471     | 0.734     | 0.853     | 0.958     | 0.998     |
>
> We observe that when the margin is less than or equal to **0.186**, the **rejection rates (type-I errors)** are limited given $\alpha=0.05$. Conversely, when margin exceeds **0.186**, our NAMMD achieves higher **rejection rates (test powers)**, indicating that our NAMMD distribution closeness testing can be effectively used to assess the confidence margin.

---

> ### Author Response · Authors · 2024-11-20
>
> In a similar manner, we present that our NAMMD can be used to assess the level of adversarial perturbation over the **CIFAR10** dataset. Using ResNet-18 as the base model, we apply the PGD attack method with perturbations set to $\\{i/255\\}_{i=1}^{[10]}$ for CIFAR10. As expected, a larger perturbation generally result in poor model performance on the perturbed CIFAR10 dataset, indicating that the perturbed CIFAR10 is farther from the original CIFAR10. Following Definition10, we set the original CIFAR10 as $P_1=P_2$ and the CIFAR10 dataset after a **4/255** perturbation as $Q_1$ for **null hypothesis**. We further set $Q_2$ as the CIFAR10 dataset after applying perturbations $i/255$ with $i\in[10]$, and perform the distribution closeness testing. The **rejection rates** of NAMMD and MMD are presented in the Table with sample size 1500.
> | Adver. Pert. | 1/255 | 2/255 | 3/255 | 4/255 | 5/255 | 6/255 | 7/255 | 8/255 | 9/255 | 10/255 |
> |-------------------|-----------|-----------|-----------|-----------|-----------|-----------|-----------|-----------|-----------|------------|
> | NAMMD         | 0.000     | 0.000     | 0.000     | 0.050     | 0.588     | 0.854     | 0.912     | 0.935     | 0.948     | 0.952      |
> | MMD           | 0.000     | 0.000     | 0.000     | 0.050     | 0.498     | 0.750     | 0.837     | 0.860     | 0.878     | 0.891      |
>
> When perturbation is less than or equal to **4/255**, the **rejection rates (type-I errors)** are limited given $\alpha=0.05$. Conversely, when perturbation exceeds **4/255**, our NAMMD achieves **higher rejection rates (test powers)**, indicating that our NAMMD distribution closeness testing can be effectively used to assess the adversarial perturbation.
>
> We sincerely value your insights and would be pleased to engage in further discussions with you regarding your concern.

---

> ### Author Response · Authors · 2024-11-22
>
> Thank you for your insightful reviews. We have submitted our response to address the concerns and questions you raised. We would greatly appreciate it if you could kindly confirm whether our response has resolved your concerns. We look forward to engaging in further discussions.

---

> ### Author Response · Authors · 2024-11-24
> **Reminder - Discussion Stage Closing Soon - 24 November**
>
> Dear Reviewer TqLo,
>
> Thank you for taking the time and effort to review our manuscript.
>
> We have carefully addressed all your comments and prepared detailed responses. Could you kindly review them at your earliest convenience?
>
> We hope our responses have satisfactorily addressed your key concerns. If anything remains unclear or requires further clarification, please do not hesitate to let us know, and we will address it promptly.
>
> We look forward to your feedback.
>
> Best regards,
>
> Authors of Submission 6378

---

> ### Author Response · Authors · 2024-11-25
> **Reminder - Discussion Stage Closing Soon - 26 November**
>
> Dear Reviewer TqLo,
>
> Thank you again for your time and effort in reviewing our paper.
>
> We have carefully addressed all your comments and prepared detailed responses. Could you kindly review them at your earliest convenience?
>
> As the rebuttal period is coming to a close, we wanted to check if there are any remaining concerns preventing you from adjusting your score. If there is any additional clarification we can provide, please do not hesitate to let us know and, we will address it promptly.
>
> Best regards,
>
> Authors of Submission 6378

---

> > ### Author Response · Authors · 2024-11-26
> > **Reminder - Discussion Stage Closing Soon - 27 November**
> >
> > Dear Reviewer TqLo,
> >
> > Thank you again for your time and effort in reviewing our paper.
> >
> > We have carefully addressed all your comments and prepared detailed responses. Could you kindly review them at your earliest convenience?
> >
> > As the discussion period is coming to a close, we wanted to check if there are any remaining concerns preventing you from adjusting your score. If there is any additional clarification we can provide, please do not hesitate to let us know and, we will address it promptly.
> >
> > Best regards,
> >
> > Authors of Submission 6378

---

> > > ### Author Response · Authors · 2024-11-29
> > > **Reminder - Discussion Stage Closing Soon - 29 November**
> > >
> > > Dear Reviewer TqLo,
> > >
> > > Thank you again for your time and effort in reviewing our paper.
> > >
> > > We have carefully addressed all your comments and provided detailed responses. We have also revised our paper based on your suggestions. Could you kindly review them at your earliest convenience?
> > >
> > > As the discussion period is coming to a close, we wanted to check if there are any remaining concerns preventing you from adjusting your score. If there is any additional clarification we can provide, please do not hesitate to let us know, and we will address it promptly.
> > >
> > > Best regards,
> > >
> > > Authors of Submission 6378

---

> > > > ### Author Response · Authors · 2024-11-30
> > > > **Reminder - Discussion Stage Closing Soon - 30 November**
> > > >
> > > > Dear Reviewer TqLo,
> > > >
> > > > Thank you again for your time and effort in reviewing our paper.
> > > >
> > > > We have carefully addressed all your comments and provided detailed responses. We have also revised our paper based on your suggestions. Could you kindly review them at your earliest convenience?
> > > >
> > > > As the discussion period is coming to a close, we wanted to check if there are any remaining concerns preventing you from adjusting your score. If there is any additional clarification we can provide, please do not hesitate to let us know, and we will address it promptly.
> > > >
> > > > Best regards,
> > > >
> > > > Authors of Submission 6378

---

> > > > > ### Author Response · Authors · 2024-12-01
> > > > > **Reminder - Discussion Stage Closing Soon -1 December**
> > > > >
> > > > > Dear Reviewer TqLo,
> > > > >
> > > > > Thank you again for your time and effort in reviewing our paper.
> > > > >
> > > > > We have carefully addressed all your comments and provided detailed responses. We have also revised our paper based on your suggestions. Could you kindly review them at your earliest convenience?
> > > > >
> > > > > As the discussion period is coming to a close, we wanted to check if there are any remaining concerns preventing you from adjusting your score. If there is any additional clarification we can provide, please do not hesitate to let us know, and we will address it promptly.
> > > > >
> > > > > Best regards,
> > > > >
> > > > > Authors of Submission 6378

---

> > > > > > ### Comment · Reviewer_TqLo · 2024-12-02
> > > > > > **Scores updated**
> > > > > >
> > > > > > Thanks for addressing my concerns about the applicability and significance of the proposed method.
> > > > > > I believe the additional numerical experiments you provided can improve the paper, so I updated my rating to 6.

---

> > > > > > > ### Author Response · Authors · 2024-12-02
> > > > > > >
> > > > > > > Dear Reviewer TqLo,
> > > > > > >
> > > > > > > Glad to hear that your concerns are effectively addressed. Many thanks for your support and increasing your score to 6!
> > > > > > >
> > > > > > > Best regards,
> > > > > > >
> > > > > > > Authors of Submission 6378

---

### Official Review · Reviewer_TXYq · 2024-11-04

**Soundness:** 1
**Presentation:** 2
**Contribution:** 3
**Rating:** 3
**Confidence:** 2

**Summary:**

The authors study distribution closeness testing (DCT) using the maximum mean discrepancy (MMD).
They find that the MMD can be the same for distributions with different norms in a reproducing kernel Hilbert space (RKHS).
To address this issue, they propose a novel kernel DCT with a norm-adaptive MMD (NAMMD), which scales the MMD along with the norms of the distributions in the RKHS.
Theoretical results for the NAMMD test are presented, which demonstrate that the proposed DCT achieves higher test power compared to the standard MMD test. Furthermore, they derive upper bounds on the sample complexity of the NAMMD test. Through empirical analysis, they demonstrate that their kernel DCT can effectively test the closeness of two distributions using both synthetic and real-world data.

#### A brief disclaime:
Please note that my review reflects a limited level of expertise in the specific area for this study. I would appreciate it if this could be taken into account when considering my assessment.

**Strengths:**

* An important issue of existing DCT using the MMD has been identified, specifically the MMD can be the same for distributions with different norms in a reproducing kernel Hilbert space.
* Rigorous theoretical results are provided.
* Sufficient experiment results are included.

**Weaknesses:**

#### Major Weaknesses:
* The issue identified by this study may reflect a different perspective on a known issue of the existing MMD DCT related to kernel selection (see quesions below).

#### Minor Weaknesses:
Lines 475-476:
Does “$\|z_0, z_1\| \le \|z_0, z_2\| \le \cdots \le \|z_0, z_9\|$” mean “$\|z_0 - z_1\| \le \|z_0 - z_2\| \le \cdots \le \|z_0 - z_9\|$”?

**Questions:**

The selection of kernels and the configuration of kernel hyperparameters may be related to the issue identified in this study.
Some research has addressed issues related to kernel selection and its hyperparameter configuration ([1], [2] and [3]).
I have questions regarding both determination of kernel hyperparameters and kernel selection.
These considerations could be beneficial for clarifying the challenges and advancing your method.

#### Questions Regarding Kernel Parameter determination.
In lines 106-107, it is possible to adjust the magnitude of the norm by modifying the length-scale hyperparameter $\gamma$ of the Gaussian kernels. Specifically, using a smaller value of $\gamma$ in Figure 1b than in Figure 1a will yield norms of the same magnitude.

1. In your numerical experiments, did you observe any significant changes in test results by manually adjusting kernel hyperparameters, such as those of the Gaussian kernels?
2. For existing MMD methods, can the selection of kernel parameters help mitigate the issues identified in this study? If so, what are the strengths and weaknesses of your approach compared to other existing MMD DCT methods that adjust kernel parameters ([1], [2] and [3])?


#### Questions Regarding Kernel Selection.
3. Does the issue observed in this study, i.e., the phenomenon where the MMD can be the same for distributions with different norms in a reproducing kernel Hilbert space (RKHS), also occur for the MMD DCT when using the following two types of kernels?

   - Unbounded kernels: For instance, kernels defined by polynomials or matrix products are unbounded but (could be) available within your theoretical framework when using observational data from bounded variables.

    - Kernels with a positive limit at infinity: kernels satisfying $\lim_{\\| \mathbf{x}-\mathbf{x}' \\|_{\infty} \rightarrow \infty} \kappa(\mathbf{x}, \mathbf{x}') = c > 0$.  An example might be a kernel defined as $\kappa (\mathbf{x}, \mathbf{x}') = \exp \left( - \frac{ \\|\mathbf{x} - \mathbf{x}'\\|^2 }{2\gamma} \right)$
    when $\\| \mathbf{x} - \mathbf{x}' \\|\_{\infty} < K$, and otherwise $\kappa(\mathbf{x}, \mathbf{x}') =c$ with positive constats $K$ and $c$.


###### Background for Questions 3.
In lines 084–102 and Figure 1, a kernel has been used such that
$\lim_{\\|\mathbf{x}-\mathbf{x}'\\|_{\infty} \rightarrow \infty} \kappa(\mathbf{x}, \mathbf{x}') = 0$.
This phenomenon could arise when a kernel fails to effectively measure similarity between distant data points. I am concerned that this issue might result from choosing a kernel that cannot capture similarity for data points beyond a certain distance.

---

[1] Biggs, F., Schrab, A., & Gretton, A. (2024). MMD-FUSE: Learning and combining kernels for two-sample testing without data splitting. Advances in Neural Information Processing Systems, 36.

[2] Schrab, A., Kim, I., Albert, M., Laurent, B., Guedj, B., & Gretton, A. (2023). MMD aggregated two-sample test. Journal of Machine Learning Research, 24(194), 1-81.

[3] Schrab, A., Kim, I., Guedj, B., & Gretton, A. (2022). Efficient Aggregated Kernel Tests using Incomplete $ U $-statistics. Advances in Neural Information Processing Systems, 35, 18793-18807.

---
### *The following comment was added to my initial review on December 17, 2024*


Dear Authors,

Thank you for your submission and the effort you have invested in this work. I appreciate the depth of your research and the importance of the problem you are addressing.

I would like to sincerely apologize for the delay in identifying and communicating the following issue. After the review period, I reported a potentially critical error in the manuscript to the Area Chair. Under the guidance of the Area Chair, I have now updated my official review to reflect this concern. I deeply regret the delay in providing this feedback and appreciate your understanding.



---

Main Comment:

The concern pertains to the proofs regarding the test power of the proposed method. Specifically, it appears that the test statistic under the null hypothesis was used to compute the test power, whereas the test statistic under the alternative hypothesis should have been employed. This discrepancy may impact the conclusions regarding the advantage of the proposed method, NAMMD, over MMD.

In particular, the following test statistic seems to have been used in the calculation of the test power:

$$
r_N = \frac{r_M }{4K - \\| (\mu_P + \mu_Q)/\sqrt{2} \\|^2\_{\mathcal{H}\_{\kappa}}}. \tag{E1}
$$

According to lines 1172–1174, this test statistic (Equation (E1)) assumes that $ \mu_P = \mu_Q = (\mu_P + \mu_Q)/2 $, as discussed in lines 848–849. Thus, it appears to be derived under the null hypothesis.

However, since the test power is defined as the probability of rejecting the null hypothesis under the alternative hypothesis, the appropriate test statistic should instead be:

$$
r_N = \frac{r_M}{4K - \\| \mu_P \\|^2\_{\mathcal{H}\_{\kappa}} - \\| \mu_Q \\|^2\_{\mathcal{H}\_{\kappa}}}. \tag{E2}
$$

The proof of Theorem 10 appears to rely on Equation (E1). Specifically, the calculations in lines 1221–1225 employ Equation (E1) rather than Equation (E2). Furthermore, when Equation (E2) is used, the test powers $ p_M $ and $ p_N $ for MMD and NAMMD, respectively, appear identical under the alternative hypothesis (lines 1211–1218).

This raises a concern that the test powers of both MMD and NAMMD may indeed be identical, which could imply that the proposed method does not offer a significant theoretical advantage over MMD.

---

Additional Comment:

While this issue may be critical to the theoretical justification of your method, I acknowledge that the numerical experiments in your paper demonstrate the effectiveness of the proposed approach. Therefore, it remains unclear why a discrepancy arises between the theoretical analysis and the numerical results.

If your method does indeed exhibit superior performance in the numerical experiments, I encourage you to investigate this phenomenon more deeply. For example, it may be beneficial to analyze whether a specific distribution, such as the t-distribution, emerges in the proposed approach. Additionally, since both the MMD and the NAMMD exhibit different convergence orders under the null and alternative hypotheses, it is essential to evaluate the tail behavior of their asymptotic distributions under the alternative hypothesis. More specifically, carefully revisiting the two equations presented in lines 1211–1218 could help clarify the observed differences.

Identifying the root cause of this discrepancy might be challenging, but I believe such an analysis will not only strengthen the validity of your results but also contribute to further advancing your proposed method.

---
**Update on Scores**:  *(Added to my additional feedback on January 6, 2025)*

Based on the identified issue, I have updated the following scores to better reflect the paper's current state:

- **Soundness**: Changed from "4: excellent" to "1: poor" due to the critical error in the proof.
- **Rating**: Changed from "8: accept, good paper" to "3: reject, not good enough" as the identified issue significantly impacts the validity of the paper's main contributions.

---

Once again, I apologize for the delay in raising this concern, and I greatly appreciate your understanding. Please feel free to clarify or address this issue in your response.

Best regards,
Reviewer TXYq

---

> ### Author Response · Authors · 2024-11-20
>
> Thank you for reading our paper and for your insightful comments. We answer your questions and concerns below.
> >[Q1] The issue identified by this study may ... related to kernel selection ... The ... configuration of kernel hyperparameters may be related to the issue identified in this study. Some research has addressed issues related to kernel selection and its hyperparameter configuration ([1], [2], and [3]).
>
> [A1] Thank you for pointing out this question. The selection of the kernel and the configuration of its parameters significantly impact the test results, which we did not discuss thoroughly in our original paper. Specifically, with selected kernels (e.g. those in [1], [2], and [3]), many distribution pairs can still exhibit the same MMD value but have different kernel norms, which actually indicates different closeness levels. In this paper, our NAMMD method leverages the norm information of distributions to mitigate the issue, which is compatible with existing kernel selection approaches (e.g., NAMMDFuse in the experiments shown in Fig.2, which is based on NAMMD and selected kernels from [1];  The kernel selection for our NAMMD provided in Appendix C.1 following Sutherland et al., 2017; Liu et al., 2020)). In Theorem 9 and 11, we prove that our NAMMD achieves better performance than MMD, when the same kernel is used for both distances. Naturally, if the optimal kernels are selected for MMD and NAMMD respectively, our NAMMD can still achieves better performance. In our current experiments, we use the same kernel for both MMD and NAMMD. Additional experiments exploring different optimal kernels for MMD and NAMMD will be conducted and included in the coming days.
>
> >[Q2] Lines 475-476: Does “$||\mathbf{z}_0, \mathbf{z}_1||^2 \geq ||\mathbf{z}_0, \mathbf{z}_2||^2 \geq \cdots \geq ||\mathbf{z}_0, \mathbf{z}_9||^2$” mean $||\mathbf{z}_0 - \mathbf{z}_1||^2 \geq ||\mathbf{z}_0 - \mathbf{z}_2||^2 \geq \cdots \geq ||\mathbf{z}_0 - \mathbf{z}_9||^2$?
>
> [A2] Sorry for the typo, it should be the $||\mathbf{z}_0 - \mathbf{z}_1||^2 \geq ||\mathbf{z}_0 - \mathbf{z}_2||^2 \geq \cdots \geq ||\mathbf{z}_0 - \mathbf{z}_9||^2$.
>
> >[Q3] In lines 106-107, ... adjust the magnitude of the norm by modifying ... $\gamma$ of the Gaussian kernels. Specifically, .....
> >- ... did you observe any significant changes in test results by manually adjusting kernel hyperparameters, such as those of the Gaussian kernels?
> >- For existing MMD methods, can the selection of kernel parameters help mitigate the issues identified in this study? If so, what are the strengths and weaknesses of your approach compared to other existing MMD DCT methods that adjust kernel parameters ([1], [2], and [3])?
>
> [A3] Thanks for your concern. It is possible to adjust the norms by modifying hyperparameter $\gamma$.
> - Selecting kernel parameters is essential for capturing differences between distributions and significantly impacts test results. In our paper, we can select the kernel by maximizing the probability of correctly recognizing different distributions. This is grounded in maximizing the test power estimator from asymptotic distribution given in Theorem 2 and has been well studied in previous approaches (Sutherland et al., 2017; Liu et al., 2020). Further details are provided in Appendix C.1. We will include additional discussion and emphasize this in our revision.
> - The selection of kernels can help mitigate the issues over specific distribution pairs. However, since the selected kernel is consistently used to evaluate the closeness of diverse and multiple distribution pairs for fair comparisons, the issue persists, particularly when testing distribution pairs not involved in the kernel selection process. By incorporating the norm information of distributions, our method effectively mitigates the issue in the general case and remains fully compatible with existing kernel selection approaches.

---

> ### Author Response · Authors · 2024-11-20
>
> >[Q4] Does the issue observed in this study, ... also occur for the MMD DCT when using the following two types of kernels?
> >- Unbounded kernels: ...
> >- Kernels with a positive limit at infinity: ...
>
> [A4] Thanks for pointing out these interesting kernels. We will expand our discussion on kernel selection and demonstrate that the issue identified in this paper is a general problem across various kernel types and hyperparameter settings. Specifically, for the two types of kernels mentioned in the question, we provide examples to illustrate that the issue (i.e., the phenomenon where distributions with different norms can yield the same MMD) also occurs with the MMD DCT.
> - Unbounded kernels: For polynomial kernels of the form
> $$
> \kappa(\mathbf{x},\mathbf{x}')=(\mathbf{x}^T\mathbf{x}'+c)^d
> $$
> We define $P_1=\\{\frac{1}{4},\frac{3}{4}\\}$ and $Q_1=\\{\frac{1}{2},\frac{1}{2}\\}$ over vector domains $\\{(\sqrt{c},...,0),(-\sqrt{c},...,0)\\}$, respectively. Furthermore, we define $P_2=\\{\frac{3}{4},\frac{1}{4}\\}$ and $Q_2=\\{1,0\\}$ over domains $\\{(\sqrt{c},...,0),(-\sqrt{c},...,0)\\}$. It is evident that
> $$
> \textnormal{MMD}(P_1,Q_1)=\textnormal{MMD}(P_2,Q_2)=\frac{1}{8}(2c)^d\ ,
> $$
> with different norms for distributions pairs $||P1||_H^2 + ||Q_1||_H^2=\frac{9}{8}(2c)^d$, and $||P_2||_H^2 + ||Q_2||_H^2=\frac{13}{8}(2c)^d$. Specifically, we have $||P_1||_H^2=\frac{5}{8}(2c)^d$, $||Q_1||_H^2=\frac{1}{2}(2c)^d$, $||P_2||_H^2=\frac{5}{8}(2c)^d$ and $||Q_2||_H^2=(2c)^d$.
>
>   In a similar manner, for matrix products kernels of the form
>   $$
>   \kappa(\mathbf{x},\mathbf{x}')=(\mathbf{x}^TM\mathbf{x}'+c)^d\ ,
>   $$
>   and denote by $M_{11}$ the element in the first row and first column of the matrix $M$. We define $P_1=\\{\frac{1}{4},\frac{3}{4}\\}$ and $Q_1=\\{\frac{1}{2},\frac{1}{2}\\}$ over vector domains $\\{(\sqrt{c/M_{11}},...,0),(-\sqrt{c/M_{11}},...,0)\\}$, respectively. Furthermore, we define $P_2=\\{\frac{3}{4},\frac{1}{4}\\}$ and $Q_2=\{1,0\}$ over domains $\\{(\sqrt{c/M_{11}},...,0),(-\sqrt{c/M_{11}},...,0)\\}$. We can obtain the same results as discussed for polynomial kernels.
> - Kernels with a positive limit at infinity: Using the given example kernel as $\kappa(\mathbf{x},\mathbf{x}')=\exp(-\frac{||\mathbf{x}-\mathbf{x}'||^2}{2\gamma})$ when $||\mathbf{x}-\mathbf{x}'||_{\infty}<K$, and otherwise $\kappa(\mathbf{x},\mathbf{x}')$ with positive constants $K$ and $c$. We define $P_1=\\{\frac{1}{4},\frac{3}{4}\\}$ and $Q_1=\\{\frac{3}{4},\frac{1}{4}\\}$ over vector domains $\\{(K,...,0),(4K,...,0)\\}$, respectively. Furthermore, we define $P_2=\\{\frac{1}{2},\frac{1}{2}\\}$ and $Q_2=\\{1,0\\}$ over domains $\\{(K,...,0),(4K,...,0)\\}$. It is evident that
>   $$
>   \textnormal{MMD}(P_1,Q_1)=\textnormal{MMD}(P_2,Q_2)=\frac{1}{2}(1-c)\ ,
>   $$
>   with different norms for distributions pairs $||P_1||_H^2 + ||Q_1||_H^2=\frac{5+3c}{4}$, and $||P_2||_H^2 + ||Q_2||_H^2=\frac{3+c}{2}$. Specifically, we have $||P_1||_H^2=\frac{5+3c}{8}$, $||Q_1||_H^2=\frac{5+3c}{8}$, $||P_2||_H^2=\frac{1+c}{2}$ and $||Q_2||_H^2=1$.
>
> >[Q5] Background for Questions 3 (i.e. [Q4] Questions Regarding Kernel Selection).
>
> [A5] We will add the background for kernel selection. A brief overview is as follows:
>
> Characteristic kernels are essential in measuring distribution differences because their kernel mean embedding uniquely represents a probability distribution, capturing all its relevant information for comparison [Sriperumbudur et al., 2011]. Kernel selection is crucial in two-sample testing, helping to choose the best characteristic kernels for detecting differences in a specific distribution pair. Some methods select kernels supervisedly using held-out data [Sutherland et al., 2017], while others use unsupervised approaches like the median heuristic [Gretton et al., 2012] or adaptively select multiple kernels [Schrab et al., 2022; Biggs et al., 2023]. Selected kernels can effectively capture differences between distributions using MMD but lack norm information, leading to challenges in comparing closeness across diverse distribution pairs, regardless of kernel type or hyperparameter settings.

---

> > ### Comment · Reviewer_TXYq · 2024-11-24
> >
> > Thank you for your detailed responses to the comments on your manuscript.
> >
> > > [A4] Thanks for pointing out these interesting kernels. We will expand our ...
> >
> > The example your raised seems to be exceedingly specific.
> > However, I understand that theoretical considerations of [Q4] are challenging.
> > It might suffice to verify whether the same results as Figure 1 can be empirically demonstrated using sysentic datasets from a normal distribution.

---

> ### Author Response · Authors · 2024-11-20
>
> [Q6] In lines 084–102 and Figure 1, a kernel has been used such that $\lim_{||\mathbf{x}-\mathbf{x}'||_{\infty}\rightarrow\infty}=0$. This phenomenon could arise when a kernel fails to effectively measure similarity between distant data points. I am concerned that this issue might result from choosing a kernel that cannot capture similarity for data points beyond a certain distance.
>
> [A6] Thanks for pointing out this concern. We use the commonly used Gaussian kernel as an example for illustration in Fig.1; however, the issue also arises when using more complex kernels, such as deep kernels [Liu et al., 2020](kernels based on neural network representations). The kernels used in our paper are all characteristic kernels, which ensure an injective mapping between mean embeddings and distributions, i.e., $P\rightarrow\mu_P$. This guarantees that different distributions have distinct embeddings, capturing unique characteristics to support efficient distribution comparison. Yet, the phenomenon still arises due to the lack of norm information of distributions in the original MMD, which we address by incorporating norm information in our NAMMD. We will add more discussion and highlight this in our revision.
>
> We deeply appreciate your insights and would welcome the opportunity to discuss any further concerns you may have.

---

> ### Author Response · Authors · 2024-11-22
>
> [A1_exp] Dear reviewer, we add the experiments for [A1] as follows:
>
> We conduct experiments to compare our NAMMD test and original MMD test in distribution closeness testing with optimal kernels. Following Definition 10, we perform the distribution closeness testing on two pairs of distributions:  $P_1$ and $Q_1$, and $P_2$ and $Q_2$, where $\textnormal{NAMMD}(P_1,Q_1)=\epsilon$ and $\textnormal{NAMMD}(Q_2,P_2)=\epsilon+0.01$. For comparison, we set $\epsilon\in\\{0.1,0.3,0.5,0.7\\}$ and conduct experiments to assess if the distance between $P_2$ and $Q_2$ is larger than that between $P_1$ and $Q_1$. To construct distributions, we draw two sets of 500 elements from dataset, denoted as $Z$ and $Z'$. Let $P_1$ and $Q_1$ be uniform distributions over $Z$ and $Z'$. We then optimize $Z$ and $Z'$ by gradient method to ensure that $\textnormal{NAMMD}(P_1,Q_1)=\epsilon$ (Appendix D.2). It is straightforward to calculate $\textnormal{MMD}(P_1,Q_1)$ and to construct $P_2$ and $Q_2$ in a similar manner. For blob dataset, the test powers with optimal kernels are given as follows
> | $\epsilon$ | 0.1  | 0.3  | 0.5  | 0.7 |
> |------------------|---------|---------|---------|---------|
> | NAMMD       | 0.983   | 0.991   | 0.999   | 1.000   |
> | MMD         | 0.939   | 0.968   | 0.952   | 0.934   |
>
> The Table shows that our NAMMD achieves higher test power than MMD in distribution closeness testing with optimal kernels. We provide similar results for other four benchmark datasets in the following parts.
>
>
> For higgs dataset, the test powers with optimal kernels are given as follows
> | $\epsilon$ | 0.1  | 0.3  | 0.5  | 0.7 |
> |------------------|---------|---------|---------|---------|
> | NAMMD       | 0.972   | 0.976   | 0.994   | 1.000   |
> | MMD         | 0.914   | 0.934   | 0.967   | 0.949   |
>
> For hdgm dataset, the test powers with optimal kernels are given as follows
> | $\epsilon$ | 0.1  | 0.3  | 0.5  | 0.7 |
> |------------------|---------|---------|---------|---------|
> | NAMMD       | 0.976   | 0.978   | 0.984   | 1.000   |
> | MMD         | 0.925   | 0.915   | 0.913   | 0.938   |
>
> For mnist dataset, the test powers with optimal kernels are given as follows
> | $\epsilon$ | 0.1  | 0.3  | 0.5  | 0.7 |
> |------------------|---------|---------|---------|---------|
> | NAMMD       | 0.962   | 0.961   | 0.967   | 0.992   |
> | MMD         | 0.951   | 0.955   | 0.935   | 0.977   |
>
> For cifar10 dataset, the test powers with optimal kernels are given as follows
> | $\epsilon$ | 0.1  | 0.3  | 0.5  | 0.7 |
> |------------------|---------|---------|---------|---------|
> | NAMMD       | 0.987   | 0.988   | 1.000   | 1.000   |
> | MMD         | 0.976   | 0.971   | 0.991   | 1.000   |

---

> ### Author Response · Authors · 2024-11-22
>
> Thank you for your thoughtful reviews. We have carefully addressed your concerns and questions in our response. We would appreciate it if you could let us know whether our response has adequately resolved your concerns. We look forward to continuing the discussion.

---

> ### Author Response · Authors · 2024-11-24
> **Reminder - Discussion Stage Closing Soon - 24 November**
>
> Dear Reviewer TXYq,
>
> Thank you for taking the time and effort to review our manuscript.
>
> We have carefully addressed all your comments and prepared detailed responses. Could you kindly review them at your earliest convenience?
>
> We hope our responses have satisfactorily addressed your key concerns. If anything remains unclear or requires further clarification, please do not hesitate to let us know, and we will address it promptly.
>
> We look forward to your feedback.
>
> Best regards,
>
> Authors of Submission 6378

---

> > ### Comment · Reviewer_TXYq · 2024-11-24
> >
> > I am terribly sorry for keeping you waiting for my response. I am currently summarizing my thoughts on the above answer. I apologize, but please wait just a little longer.

---

> ### Author Response · Authors · 2024-11-24
>
> Dear Reviewer TXYq,
>
> Thank you for your response. We completely understand that you need time to gather your thoughts and provide feedback. Please feel free to take the time you need—we truly value your insights and look forward to receiving your response whenever you're ready.
>
> If there is anything we can assist with or clarify further in the meantime, please do not hesitate to let us know.
>
> Best regards,
> Authors of Submission 6378

---

> ### Comment · Reviewer_TXYq · 2024-11-24
>
> Thank you very much for your detailed and thoughtful responses to my previous comments. I deeply appreciate the effort you have put into addressing the points I raised.
>
> I would also like to sincerely apologize for the delay in providing my reply. I appreciate your patience in awaiting my feedback.
>
> Below, I have summarized my thoughts and observations based on your responses. I would be grateful if you could share your opinions or any additional insights on these points. Please note that I plan to provide a specific feedback on each of your responses later. For now, I would like to focus on presenting a comprehensive summary of my concerns and suggestions.
>
> ---
> The issue addressed in this study ("MMD value is less informative when measuring the closeness between two distributions, i.e., MMD value can be the same for many pairs of distributions that have different norms in the RKHS") mainly arises due to the selection of the kernel parameter used in the norm. I believe that the following four revisions focusing on this are necessary.
>
> However, I am concerned that these changes are so significant that they may require a fundamental re-examination of the study. Could you please let me know your opinions on these points?
>
> ### 1. The need to accurately rewrite the description of the issue addressed in this study:
>
>   - The statement "MMD value is less informative when measuring the closeness between two distributions, i.e., MMD value can be the same for many pairs of distributions that have different norms in the RKHS" refers to the case when the kernel parameter is fixed. I believe that this issue does not occur when an appropriate kernel parameter is optimally selected with a characteristic kernel.
>
>   - Additionally, the assertion "MMD value can be the same for many pairs of distributions that have different norms in the RKHS" may not hold for kernels other than the Gaussian kernel (see Supplement Note 1 below).
>
> ### 2. Necessity to rewrite the main body of the paper to reflect point 1:
>
>   - Based on point 1, the discussion regarding kernel parameter selection should not be treated as merely supplementary. I believe that a comprehensive revision of the manuscript, including the introduction, is necessary.
>
>
> ### 3. Conducting numerical experiments considering point 1:
>
>   - Each numerical experiment should involve the selection of kernel parameters.
>
>   - Furthermore, I anticipate that the results of Figure 1 (c) and (d) would differ significantly if kernel parameter optimization is performed (see Supplement Note 2 below).
>
> ### 4. Addition or modification of theoretical results in response to point 1:
>
>   - The main result of this study might be more appropriately characterized as a "Proposal of a robust estimation method with respect to kernel parameter selection". The following theoretical aspects need to be described more rigorously to clarify the properties of the proposed method.
>
>       * What effects can be expected from the proposed method?
>         (e.g., the variance of the estimator remains below a certain level)
>
>       * For which kernels can these effects be expected? (e.g., for characteristic kernels satisfying $\lim\_{\|\mathbf{x}-\mathbf{x}'\|\_{\infty} \rightarrow \infty} \kappa(\mathbf{x}, \mathbf{x}') = 0$)
>
> ---
> ### Supplement Note 1:
>
> This study proposes a method considering a kernel such as where $ \operatorname{Var}(P, \kappa) = 1 - \\| \mu_P \\|^2\_{\mathcal{H}\_\kappa} $ and  $\operatorname{Var}(P, \kappa) = 1 - \\| \mu_P \\|^2\_{\mathcal{H}_\kappa}$ (in line 90).
> I believe that the proposed method can achieve robustness in the MMD estimation (the variance of the estimator remains below a certain level), when there exists such a relationship between the increase in data variance and the decrease in embedding norm. On the other hand, for kernels not like this one (e.g., polynomial kernels on a compact support, where the embedding norm increases as the data variance increases), it is anticipated that the proposed method may not work well.
>
> ### Supplement Note 2:
>
> Instead of fixing the kernel parameters for these graphs, the results should be those where the kernel parameters are tuned for the variance of the data in each case (i.e., "Norms of Distributions"). I anticipate that the situation would change significantly in both graphs.

---

> ### Comment · Reviewer_TXYq · 2024-11-24
>
> > [A1] Thank you for pointing out this question. The selection of ...
>
> > [A3] Thanks for your concern. It is possible to adjust the norms by modifying hyperparameter ...
> ---
> Thank you for your detailed responses to the comments on your manuscript. I would also like to apologize for the delay in providing my feedback.
>
> I agree that there are cases where kernel selection cannot be performed. However, the problem setting in your research appears to be ambiguous on this point. It might be better to clarify whether you are considering a scenario where kernel parameter selection is not performed, or if you allow situations where kernel parameter tuning is permitted.
>
> Additionally, in the former case, it may be more appropriate to demonstrate the effectiveness of the proposed method through comparisons, such as minimax evaluations with respect to a kernel parameter set, rather than comparing estimation accuracies for specific parameters (Theorems 9 and 11).

---

> ### Comment · Reviewer_TXYq · 2024-11-24
>
> > [Q6] Thanks for pointing out this concern. We use the commonly used Gaussian kernel ...
>
> Thank you for your detailed responses to the comments on your manuscript.
>
> I disagree with the following point:
>
>     Yet, the phenomenon still arises due to the lack of norm information of distributions in the original MMD, which we address by incorporating norm information in our NAMMD. We will add more discussion and highlight this in our revision.
>
> This point is difficult to agree with for the following two reasons:
>
> 1. The phenomenon of “lack of norm information in distributions” does not occur for characteristic kernels with optimal kernel parameters.
> 2. Even when the kernel parameters are not optimized, the results of your numerical experiments (Figure 1 (d)) may vary significantly depending on the choice of kernel. In particular,  based on the definition of the MMD (line 139), I anticipate the MMD may increase in accordance with increases in the data variance (“Norms of Distributions”)  in the case of "Unbounded kernels: For polynomial kernels of the form".

---

> ### Author Response · Authors · 2024-11-25
>
> Dear Reviewer TXYq,
>
> Many thanks for taking your time to review our responses and providing many insightful comments, which will help us refine and strengthen our work. We acknowledge that our current explanation of the motivation and contributions may not have been as clear as intended, and we sincerely apologize for any confusion/overclaims this may have caused. We would like to provide a clearer and more concise explanation as follows.
>
> To avoid potential overclaims, we will adopt a more cautious tone in the paper. We will emphasize that this paper's analysis applies to kernels with the form $\kappa(x,x')=\Psi(x-x')\leq K$ for a positive-definite $\Psi(\cdot)$ and $\Psi(0)=K$, which includes the Gaussian kernel, Laplace kernel, Mahalanobis kernel, and Deep kernel (some frequently used kernels in kernel-based hypothesis testing). Additionally, we will include a **limitation statement** regarding the applicability of our findings to other kernels.
>
> We would like first to **highlight** the following points, which should be very useful to mitigate your concerns. It would be great if you can read them first ^^.
>
> - Existing kernel selection methods are primarily designed for **two-sample testing (TST)**, focusing on selecting the optimal kernel that maximizes the **test power estimator of TST** to distinguish two **fixed** distributions $P$ and $Q$ [Sutherland et al., 2017; Liu et al., 2020]. For TST, NAMMD and MMD actually share the **same test power estimator** because, asymptotically, after we fixed two distributions $P$ and $Q$, NAMMD can be viewed as MMD scaled by a constant $4K-||\mu_P||-||\mu_Q||$. Hence, the NAMMD and MMD has the **same optimal kernel for TST**. For the same kernel, when we use permutation test to do perform two-sample tests, NAMMD achieves higher test power than MMD due to its scaling as stated in Theorem 9.
>
> - In **kernel-based distribution closeness testing (DCT)**, the distance between a distribution pair ($P_2$ and $Q_2$) is compared to a predefined threshold $\epsilon$, which is determined as the distance between a reference pair of distributions ($P_1$ and $Q_1$). In this case, we use a kernel to calculate the $\epsilon$, and we also need to use the **same fixed kernel** to calculate the distance for distribution pair ($P_2$ and $Q_2$) to ensure the **same distance measurement**. However, how to obtain the **test power estimator of DCT with multiple distribution pairs** and select a **global optimal kernel for DCT** based on the estimator remains **an open question** and an important future research.
>
> **In this work**, **we select the kernel following the TST method with the fixed reference distribution pair ($P_1$ and $Q_1$)** (on lines 175-179 and Appendix C.1). However, the **TST-optimal kernel** may exhibit the issue we identified in **DCT** with MMD, as the kernel selection does not account for distribution pair ($P_2$ and $Q_2$). We propose NAMMD to mitigate this issue by incorporating norm information, $4K-||\mu_P||-||\mu_Q||$, for kernels with the form $\kappa(x,x')=\Psi(x-x')\leq K$ for a positive-definite $\Psi(\cdot)$ and $\Psi(0)=K$, which includes the Gaussian kernel, Laplace kernel, Mahalanobis kernel, and Deep kernel. Since the **global optimal kernel** cannot be obtained analytically, we prove that, as long as MMD and NAMMD share the same kernel, our NAMMD performs better than MMD as shown in **Theorem 11**.
>
> **One conjunction.** Now, based on Theorem 11, we might have an interesting conjunction. We can assume a scenario where we can obtain the best kernel $k^{\rm MD}$ for MMD DCT (instead of MMD TST) and the best kernel $k^{\rm ND}$ for NAMMD DCT. Based on Theorem 11, if we use the kernel $k^{\rm MD}$ (MMD's best kernel) for NAMMD, then NAMMD DCT will perform better than MMD DCT already. Because $k^{\rm ND}$ is the kernel to make NAMMD DCT have the highest test power (in DCT, instead of TST), NAMMD DCT with $k^{\rm ND}$ should have a higher or equal test power compared to NAMMD DCT with $k^{\rm MD}$. Thus, NAMMD DCT with $k^{\rm ND}$ has a higher test power than NAMMD DCT with $k^{\rm MD}$ (because NAMMD DCT with $k^{\rm MD}$ has a higher power than MMD DCT with $k^{\rm MD}$). We are happy to discuss more with you regading the above content.
>
> [1] D. J. Sutherland, H.-Y. Tung, H. Strathmann, S. De, A. Ramdas, A.J. Smola, and A. Gretton. Generative models and model criticism via optimized maximum mean discrepancy. 2017.
>
> [2] F. Liu, W.-K. Xu, J. Lu, G.-Q. Zhang, A. Gretton, and D. J. Sutherland. Learning deep kernels for non-parametric two-sample tests. 2020.

---

> > ### Author Response · Authors · 2024-11-25
> >
> > We then address your points in detail as follows.
> >
> > >[Q1] The need to accurately rewrite the description of the issue addressed in this study:
> > >- The statement ".... for many pairs of distributions that have different norms in the RKHS" refers to the case when the kernel parameter is fixed. I believe that this issue does not occur when an appropriate kernel parameter is optimally selected with a characteristic kernel.
> > >- Additionally, the assertion "MMD value can be the same for many pairs of distributions that have different norms in the RKHS" may not hold for kernels other than the Gaussian kernel (see Supplement Note 1 below).
> >
> > [A1] Thank you for your concern.
> > - As mentioned in above **highlighted part**, using the same fixed kernel for DCT is essential when comparing multiple distirbution pairs, and the selection of **a global optimal kernel** for DCT remains an open question and an important future work. We will add a paragraph in the introduction to include the current kernel selection approaches for TST and disccuss on the gloabal optimal kernel for DCT.
> > - We will lower our tone to avoid potential overclaims and emphasize that this paper's analysis applies to kernels with the form $\kappa(x,x')=\Psi(x-x')\leq K$ for a positive-definite $\Psi(\cdot)$ and $\Psi(0)=K$, which includes the Gaussian kernel, Laplace kernel, Mahalanobis kernel, and Deep kernel (frequently used in kernel-based hypothesis testing).
> >
> > >[Q2] Necessity to rewrite the main body of the paper to reflect point 1:
> > >- Based on point 1, the discussion regarding kernel parameter selection should not be treated as merely supplementary. I believe that a comprehensive revision of the manuscript, including the introduction, is necessary.
> >
> > [A2] Thanks for pointing out this question. As mentioned in [A1], we will include a paragraph about kernel selection in introduction, as current kernel selections are designed for TST and identifying the global optimal kernel for DCT is an open question and an important future work. To the best of our knowledge, this is **the first study on Kernel-DCT**, and our main contribution is to mitigate the current issue by proposing a new distance, which is highly compatible with existing and future kernel selection methods. Given this, the main body of the paper is centered on the main contribution, and we plan to disccuss the kernel selection in the introduction. We will also emphasize that this paper's analysis applies to kernels with the specific form in the introduction to avoid overclaiming. Do you think this revision is acceptable? We would greatly appreciate your feedback or any additional insights on any unclear points that may require further revision.
> >
> > >[Q3] Conducting numerical experiments considering point 1:
> > >- Each numerical experiment should involve the selection of kernel parameters.
> > >- Furthermore, I anticipate that the results of Figure 1 (c) and (d) would differ significantly if kernel parameter optimization is performed (see Supplement Note 2 below).
> >
> > [A3] Thank you for the comments.
> > - We apologize for the lack of clarity for the experimental details. We will clarify at the beginning of the Experiment Section that the kernel selection in all numerical experiments follows the approach in TST [Liu et al., 2020].
> > - As we mentioned in the above **highlighted part**, we select and use the **same fixed TST-optimal kernel** in DCT for different distribution pairs, and the selction of **a global optimal kernel** for DCT remains an open question and an important future work. Given this, the Figure 1 remains valid and effective, and we will update its caption to indicate that it represents an empirical example with Gaussian kernel, but extendable to other kernels with the specified form.
> >
> > >[Q4] Addition or modification of theoretical results in response to point 1:
> > >- The main result of this study might be more appropriately characterized as a "Proposal of a robust estimation method with respect to kernel parameter selection". The following theoretical aspects need to be described more rigorously to clarify the properties of the proposed method.
> > -- What effects can be expected from the proposed method? (e.g., the variance of the estimator remains below a certain level)
> > -- For which kernels can these effects be expected? (e.g., for characteristic kernels satisfying)
> >
> > [A4] Thank you for raising this question.
> > - All of our theoretical results hold regardless of the level of estimator variance. Specifically, for Theorem 9 and 11, the level of estimator variance will affect the sample size $C'$ and $C''$, as detailed in the proofs on lines 1184–1194 and 1320–1330. Smaller estimator variance corresponds to larger required sample sizes.
> > - Above effects are expected for bounded characteristic kernels, where boundedness is a fundamental condition used in kernel-based hypothesis testing [Hoeffding, 1994; Serfling, 2009; Gretton et al., 2012].

---

> > > ### Author Response · Authors · 2024-11-25
> > >
> > > >[Q5] I agree that there are cases where kernel selection cannot be performed. However, the problem setting in your research appears to be ambiguous on this point. It might be better to clarify whether you are considering a scenario where kernel parameter selection is not performed, or if you allow situations where kernel parameter tuning is permitted. Additionally, in the former case, it may be more appropriate to demonstrate the effectiveness of the proposed method through comparisons, such as minimax evaluations with respect to a kernel parameter set, rather than comparing estimation accuracies for specific parameters (Theorems 9 and 11).
> > >
> > > [A5] Thanks for your question. As mentioned in above **highlighted part**, we select the kernel following the TST method with the fixed reference distribution pair ($P_1$ and $Q_1$) and we use this **fixed TST-optimal kernel** in DCT for mutiple distribution pairs **(where the global optimal kernel for DCT is unknown)**.
> > >
> > > Thank you for pointing out this interesting theoretical analysis regarding minimax evaluations w.r.t. a kernel parameter set. We attempted to address this issue but found that identifying a global optimal kernel for DCT within the kernel parameter set remains a significant challenge. We acknowledge this as an **open question** and leave it for future work.
> > >
> > > >[Q6] The example your raised seems to be exceedingly specific. However, I understand that theoretical considerations of [Q4] are challenging. It might suffice to verify whether the same results as Figure 1 can be empirically demonstrated using sysentic datasets from a normal distribution.
> > >
> > > [A6] Thanks for your concern. We will include additional figures with different kernels for better illustration.
> > >
> > > >[Q7] The phenomenon of “lack of norm information in distributions” does not occur for characteristic kernels with optimal kernel parameters.
> > >
> > > [A7] Thanks for your question. We will change all relevant statements in the whole work as follows: Since the **global optimal kernel for DCT** cannot be determined analytically, we use an empirically selected and fixed kernel for DCT with multiple distribution pairs. In this case, the observed phenomenon may arise due to the absence of norm information for distributions in the original MMD with kernels of the form: $\kappa(x,x')=\Psi(x-x')\leq K$ for a positive-definite $\Psi(\cdot)$ and $\Psi(0)=K$, which includes the Gaussian kernel, Laplace kernel, Mahalanobis kernel, and Deep kernel (some frequently used kernels in kernel-based hypothesis testing).
> > >
> > > >[Q8] Even when the kernel parameters are not optimized, the results of your numerical experiments (Figure 1 (d)) may vary significantly depending on the choice of kernel. In particular, based on the definition of the MMD (line 139), I anticipate the MMD may increase in accordance with increases in the data variance (“Norms of Distributions”) in the case of "Unbounded kernels: For polynomial kernels of the form".
> > >
> > > [A8] Thanks for your concern. We will emphasize that this paper's analysis applies to kernels with the form $\kappa(x,x')=\Psi(x-x')\leq K$ for a positive-definite $\Psi(\cdot)$ and $\Psi(0)=K$, which includes the Gaussian kernel, Laplace kernel, Mahalanobis kernel, and Deep kernel (frequently used in kernel-based hypothesis testing). we will update Figure 1 caption to indicate that it represents an empirical example with Gaussian kernel, but extendable to other kernels with the specified form. Additionally, we will include a **limitation statement** regarding the applicability of our findings to other kernels, where the norms of distributions may increase in accordance with increases in the data variance.
> > >
> > > To avoid potential overclaims, we will adopt a more cautious tone in the paper. We will emphasize that this paper's analysis applies to kernels with specific form. Thank you very much for your valuable comments. If anything remains unclear or requires further clarification, please do not hesitate to let us know, and we will address it promptly.
> > >
> > > Best regards,
> > >
> > > Authors of Submission 6378

---

> ### Comment · Reviewer_TXYq · 2024-11-25
>
> Thank you for your prompt and thoughtful response to my comments. I appreciate the effort you have made to address the concerns I raised in such a short time.
>
> I believe that the approach you have considered completely addresses the concerns I raised.
>
> In particular, I found that the following points are not only effective in clarifying the contributions of your research but also indicate directions for future research by identifying the two distinct issues, TST and DCT, in distribution closeness testing:
>
> 1. The separation of discussions on test power under optimal kernel parameters (TST) and test power when the kernel parameters are not optimal (DCT).
> 2. The demonstration of the effectiveness of the proposed method in the DCT setting.
>
> Additionally, regarding points 1 and 2, I believe that the following conjunction is highly persuasive:
>
> > One conjunction. Now, based on Theorem 11, we might have an interesting conjunction. We can assume a scenario where we can obtain the best kernel for MMD DCT (instead of MMD TST) and the best kernel for NAMMD DCT. Based on Theorem 11, if we use the kernel (MMD's best kernel) for NAMMD, then NAMMD DCT will perform better than MMD DCT already. Because is the kernel to make NAMMD DCT have the highest test power (in DCT, instead of TST), NAMMD DCT with should have a higher or equal test power compared to NAMMD DCT with. Thus, NAMMD DCT with  has a higher test power than NAMMD DCT with (because NAMMD DCT with has a higher power than MMD DCT with ). We are happy to discuss more with you regading the above content.
>
> Thank you for considering such effective revisions to the research outcomes in such a short period, despite the delay in my response. Thank you again for the highly engaging discussions regarding your research.

---

> > ### Author Response · Authors · 2024-11-25
> > **Thanks for your support!!**
> >
> > Dear Reviewer TXYq,
> >
> > It is our pleasure to discuss our paper with you. It is always important to have this kind of insightful discussion (the meaning of OpenReview in our opinion), which can help us a lot! Given our discussion, we get the position of our paper and how it will influence future research much more clearly.
> >
> > Really thank you for your reply and insightful critical thinking!
> >
> > We will revise our paper based on our discussion. Thanks for your firm support.
> >
> > Best regards,
> >
> > Authors of Submission 6378

---

> ### Public Comment · ~Feng_Liu2 · 2025-02-14
> **A factual error in your additional comments (Part I: Testing threshold in two-sample testing)**
>
> Dear Reviewer TXYq,
>
> Thank you for reading our paper and for your further reviews. Unfortunately, **the issue was raised 20 days after the rebuttal period**, and we cannot reply through the OpenReview system after the author rebuttal deadline. Although we have reached out to ICLR program chairs via email, we were still unable to resolve this matter.
>
> We address your concern below and hope that future readers will find our clarification helpful. Although the paper has already been rejected, we appreciate the opportunity to provide further insights and contribute to the ongoing discussion with you.
>
> >[Q1] In particular, the following test statistic seems to have been used in the calculation of the test power:
> $$
> r_N = \frac{r_M}{4K-||(\mu_P+\mu_Q)/\sqrt{2}||^2}\ .
> $$
> According to lines 1172–1174, this test statistic (Equation (E1)) assumes that $\mu_P=\mu_Q=(\mu_P+\mu_Q)/2$, as discussed in lines 848–849. Thus, it appears to be derived under the null hypothesis. However, since the test power is defined as the probability of rejecting the null hypothesis under the alternative hypothesis, the appropriate test statistic should instead be:
> $$
> r_N = \frac{r_M}{4K-||\mu_P||^2+||\mu_Q||^2}\ .
> $$
>
> [A1] We believe there is a **key misunderstanding about the testing thresholds $r_M$ and $r_N$.** These thresholds should be determined based on **the null distribution** [1-4] (the distribution of the test statistics under the null hypothesis, i.e., $H_0: P = Q$ for Theorem 10 ). **In contrast,**, in your additional comments, we found that you might think the **testing thresholds** should be from **the alternative distribution** (the distribution of test statistics under the alternative hypothesis, i.e., $H_1:P\neq Q$), **which is incorrect**.
>
> To clarify this point, we first introduce the background of hypothesis testing. Hypothesis testing evaluates whether there is enough evidence to reject a null hypothesis $H_0$ in favor of an alternative hypothesis $H_1$. The process involves defining a **test statistic $T$** and comparing it to a **testing threshold $r$** to determine whether to reject the null hypothesis, where
> 1. Test Statistic $T$: The test statistic $T$ is a numerical value calculated from the sample data that reflects how much the data deviates from what is expected under the null hypothesis.
> 2. Testing Threshold $r$:  The testing threshold is determined based on the significance level $\alpha$, which represents the probability of rejecting $H_0$ **when it is true**. Hence, this threshold is often derived from the $(1-\alpha$)-quantile of the (null) distribution of the test statistic **under null hypothesis $H_0$**.
>
> Hence, the test power (i.e., **the probability of correctly rejecting the null hypothesis when the alternative hypothesis is true**) is given by
>
> $$
> \Pr [T\geq r]\ ,
> $$
>
> which represents the probability that the **test statistic $T$ (calculated under alternative hypothesis where two samples are from different distributions)** exceeds the testing threshold $r$ (calculated under the **null hypothesis**).

---

> ### Public Comment · ~Feng_Liu2 · 2025-02-14
> **A factual error in your additional comments (Part II: Why our proof is correct)**
>
> Then, we will show **why our $r_N$ is correct** and why NAMMD has higher test power than MMD. In our paper, for Theorem 10, which addresses the two-sample testing with hypotheses
> $$
> H_0: P=Q  \quad \quad and \quad \quad H_1: P\neq Q\ .
> $$
>
> For MMD test, we have the test statistic $T=m\widehat{\text{MMD}}$ and the asymptotic testing threshold $r_M$, which is the $(1-\alpha)$-quantile of **the null distirbution** of the test statistic, which is calculated **under the null hypothesis $H_0: P=Q$**. The corresponding test power is given by
> $$
> p_M = \Pr[m\widehat{\text{MMD}}\geq r_M]\ .
> $$
>
> In a similar manner, for NAMMD test, we have the test statistic $T=m\widehat{\text{NAMMD}}$ and the asymptotic testing threshold $r_N$, which is the $(1-\alpha)$-quantile of **the null distirbution** of the test statistic, calculated **under the null hypothesis $H_0: P=Q$**. Hence, we have that
> $$
> r_N = \frac{r_M}{4K-||(\mu_P+\mu_Q)/\sqrt{2}||^2}\ ,
> $$
> meaning that our calculated $r_N$ is correct, and **the $r_N$ in your comment is incorrect**. Furthermore, the test power of NAMMD test is given by
> $$
> p_N = \Pr[m\widehat{\text{NAMMD}}\geq r_N]=\Pr[m\widehat{\text{NAMMD}}\geq \frac{r_M}{4K-||(\mu_P+\mu_Q)/\sqrt{2}||^2}]\ .
> $$
>
> Consequently, the difference between the test powers of MMD and NAMMD is
> $$
> \varsigma=p_N-p_M=\Pr[m\widehat{\text{NAMMD}}\geq \frac{r_M}{4K-||(\mu_P+\mu_Q)/\sqrt{2}||^2}] - \Pr[m\widehat{\text{MMD}}\geq r_M]\ .
> $$
> **This aligns the critical result of our proof in line 1236 and confirms the correctness of Theorem 10.**
>
> Building on this, we prove in Theorem 10 that under the alternative hypothesis, if MMD test **rejects null hypothesis correctly** (i.e., $m\widehat{MMD}$>$r_M$, where $m\widehat{MMD}$ is calculated under alternative hypothesis and $r_M$ is calculated under null hypothesis), our NAMMD test also do so with high probability (i.e., $m\widehat{NAMMD}$>$r_N$, where $m\widehat{NAMMD}$ is calculated under alternative hypothesis and $r_N$ is calculated under null hypothesis). Moreover, NAMMD can reject null hypothesis in cases where MMD fails, **resulting in higher test power. This is consistent with our experimental results, where NAMMD demonstrates better performance.**
>
> All the below **references** mentioned that the testing threshold is calculated **under the null hypothesis**.
>
> [1] W. Jitkrittum, Z. Szabó, K. P. Chwialkowski, and A. Gretton. Interpretable distribution features with maximum testing power. In Advances in Neural Information Processing Systems 29 pages 181–189. Curran Associates, Dutchess, NY, 2016.
> **"The threshold is given by the $(1 − \alpha)$-quantile of the (asymptotic) distribution of $\lambda^n$ under $H_0$, the null distribution."**
>
> [2] K. Chwialkowski, A. Ramdas, D. Sejdinovic, and A. Gretton. Fast two-sample testing with analytic representations of probability measures. In Advances in Neural Information Processing Systems 28, pages 1981–1989. Curran Associates, Dutchess, NY, 2015.
> **"the test threshold $T_\alpha$ is given by the $(1 − \alpha)$-quantile of the asymptotic null distribution $\chi^2(J)$"**
>
> [3] Gretton, K. M. Borgwardt, M. J. Rasch, B. Schölkopf, and A. Smola. A kernel two-sample test. Journal of Machine Learning Research, 13(1):723–773, 2012.
> **"define a test threshold under the null hypothesis $P = Q$"**
>
> [4] F. Biggs, A. Schrab, and A. Gretton. MMD-Fuse: Learning and combining kernels for two-sample testing without data splitting. In Advances in Neural Information Processing Systems36. Curran Associates, Dutchess, NY, 2023.
> **"We can use permutations to construct an approximate cumulative distribution function (CDF) of our test statistic under the null, and choose an appropriate quantile of this CDF as our test threshold"**
>
> Looking forward to your further reply!
>
> Best regards,
>
> Feng

---

> ### Comment · Reviewer_TXYq · 2025-02-16
> **Re: Follow-up on Testing Threshold in Two-Sample Testing**
>
> Dear Dr. Liu,
>
> Thank you for reaching out and for providing a detailed clarification regarding my concerns.
> I sincerely regret that my identification of this issue came after the rebuttal period, ultimately leading to a decision where you were unable to fully respond.
> I understand how frustrating this must be, and I truly appreciate the effort you have put into explaining your perspective.
>
> Because I highly value the work you have put into this research and the effort you have made to clarify your position, I want to ensure that I carefully review your explanation, as well as the references you have cited, to fully understand the theoretical framework you have presented.
>
> While the review process has concluded and the decision was made through discussions among multiple reviewers and the meta-reviewer, I would like to personally provide a well-considered response that not only addresses the specific issue at hand but also offers feedback that may be useful for the future development of your work.
>
> However, due to my current time constraints, I kindly ask for your patience as I take the necessary time to thoroughly re-examine your arguments.
>
> I apologize for keeping you waiting for my response, and I sincerely appreciate your kind understanding.
>
>
> Best regards,
>
> Reviewer TXYq

---

> > ### Public Comment · ~Feng_Liu2 · 2025-02-16
> >
> > Dear Reviewer TXYq,
> >
> > Many thanks for the reply.
> >
> > We can also feel that you really want to carefully review our paper, which is very much appreciated!
> >
> > Although the decision is not ideal, we still want to discuss and clarify this issue. As it is a public issue against our paper, it would be great if we could further address this and deliver a clearer message to potential readers.
> >
> > We fully understand your time constraint and hope to get your responses when you are available.
> >
> > Best,
> >
> > Feng

---

> ### Comment · Reviewer_TXYq · 2025-02-23
> **Re: Follow-up on Testing Threshold in Two-Sample Testing  (Part I: Main Arguments)**
>
> Dear Dr. Liu,
>
> I sincerely apologize for the delay in my response. I have carefully reviewed your explanation and the referenced materials you kindly provided, and I have reassessed my original concern regarding Theorem 10 in your paper.
>
>
> In summary, I remain unconvinced by your explanation. I still believe that the proof of Theorem 10 does not adequately demonstrate the effectiveness of the proposed method. Below, I clarify my position, outline my main arguments, and provide supplementary explanations and suggestions regarding your proof.
>
> ---
>
> ## 1. Clarifying the Points of Agreement
>
> Let me restate my position to avoid any confusion. I fully agree with the following two points you made:
>
> 1. **Test Statistic $T$:**
>    The test statistic $T$ is a numerical value calculated from the sample data that reflects how much the data deviate from what is expected under the null hypothesis $H_0$.
>
> 2. **Testing Threshold $r$:**
>    The testing threshold is determined based on the significance level $\alpha$, which represents the probability of rejecting $H_0$ when it is in fact true. This threshold is typically derived from the $(1-\alpha)$-quantile of the null distribution of the test statistic.
>
> Given these two points, my concern focuses on the following aspect:
>
> 3. **Power of the Test:**
>    The power of the test is the probability of correctly rejecting the null hypothesis, evaluated under the alternative hypothesis $H_1$. In other words, it is the probability that the test statistic $T$ exceeds the threshold $r$ (calculated under $H_0$) when the true distribution is given by $H_1$.
>
> To make my notation clear, let
> - $r_M$ be the testing threshold for the MMD-based test (determined under $H_0$),
> - $r_N^1$ be
>   $$
>     r_N^1 = \frac{r_M}{4K - \| (\mu_P + \mu_Q)/\sqrt{2} \|^2_{\mathcal{H}\_{\kappa}}},
>     \tag{E1}
>   $$
> - $r_N^2$ be
>   $$
>     r_N^2 = \frac{r_M}{4K - \|\mu_P\|^2_{\mathcal{H}\_{\kappa}} - \|\mu_Q\|^2_{\mathcal{H}\_{\kappa}}},
>     \tag{E2}
>   $$
> where $r_N^1$ and $r_N^2$ are two candidate expressions related to NNMMD.
>
> ---
>
> ### 2. Main Arguments
>
> The fundamental issue with the proof of Theorem 10, as I see it, can be understood via two observations (labeled as R1 and R2 below). These observations suggest that the proposed argument in your paper does not sufficiently establish the advantage of the new method:
>
> * **R1. Under the null hypothesis, $r_N^1$ and $r_N^2$ coincide.**
> * **R2. In general, changing only the scale of a test statistic does not alter its power.**
>
> I view $r_N^1$ and $r_N^2$ not as the definition of the NNMMD test statistic itself, but rather as (asymptotic) relationships capturing possible scalings between the MMD statistic and the NNMMD statistic. In lines 221–237 of your paper, one can find a more direct definition of the NNMMD test statistic, which I believe should serve as the starting point.
>
> (Addendum on February 24, 2025) Based on point 3 in '1. Clarifying the Points of Agreement', note that Equation (E1) holds only under the null hypothesis and thus cannot be employed when assessing test power under the alternative hypothesis. A relation like Equation (E2), valid under the alternative hypothesis, must be used.
>
> ---
>
> ### R1. Explanation
>
> Under the null hypothesis $H_0$, $P = Q$. Therefore, the following equality holds:
>
> $$\|\mu_P\|\_{\mathcal{H}\_{\kappa}}^2 = \|\mu_Q\|\_{\mathcal{H}\_{\kappa}}^2,
> $$
>
> which implies
>
> $$ r_N^1 = \frac{r_M}{4K -
> \left\|\frac{\mu_P + \mu_Q}{\sqrt{2}}\right\|\_{\mathcal{H}\_{\kappa}}^2} =
> \frac{r_M}{4K - \|\mu_P\|\_{\mathcal{H}\_{\kappa}}^2 - \|\mu_Q\|\_{\mathcal{H}\_{\kappa}}^2} =
> r_N^2.
> $$
>
> Hence, if $r_N^1$ is defined based on the null hypothesis distribution (as you suggest), then $r_N^2$ is equally definable under the same assumption. They coincide under $H_0$.
>
> ---
>
> ### R2. Explanation
>
> Let me restate the second key idea using an abstract setting. Suppose we have a test statistic $R$ with null distribution $P_{\mathrm{null}}(R)$ and alternative distribution $P_{\mathrm{alt}}(R)$. The original test rejects $H_0$ when
>
> $$
> P_{\mathrm{null}}(R > r) = \alpha,
> $$
> where $\alpha$ is the significance level.
> Now consider a scaled version of the same statistic, $R / \sigma$. To preserve the same Type I error $\alpha$, the threshold $r'$ must satisfy
>
> $$
> P_{\mathrm{null}}\Bigl(\frac{R}{\sigma} > r'\Bigr) = \alpha.
> $$
>
> Because $\frac{R}{\sigma} > r'$ is equivalent to $R > \sigma \cdot r'$, if we set $r' = r/\sigma$, both tests have the same $\alpha$. Under $H_1$, we then get
>
> $$
> P_{\mathrm{alt}}(R > r) = P_{\mathrm{alt}}\Bigl(\frac{R}{\sigma} > r'\Bigr).
> $$
>
> Hence, simply scaling the test statistic does not change the power of the test.

---

> > ### Comment · Reviewer_TXYq · 2025-02-23
> > **Re: Follow-up on Testing Threshold in Two-Sample Testing  (Part II: Additional Remarks and Future Directions on the Proof)**
> >
> > ## 3. Additional Remarks and Future Directions on the Proof
> > Note that the asymptotic analysis shows that the NNMMD test statistic is essentially a scaled version of the MMD test statistic. However, the reasoning in R2 suggests that this scaling does not reveal any advantage of the proposed method in terms of test power.
> >
> >
> > A more fruitful approach might therefore involve analyzing exact (non-asymptotic) distributions or exploring alternative properties of the statistic to uncover the potential strengths of NNMMD.
> > This situation is analogous to the relationship between the $t$-statistic and the sample mean: although the $t$-test is uniformly most powerful unbiased, the detection powers of both tests become identical as the sample size increases (assuming finite variance).
> >
> > ---
> >
> > I regret that I was unable to communicate these concerns during the review period, and I apologize for the delay in providing this detailed response. If you have any further questions or require additional clarification, please do not hesitate to contact me.
> >
> > Sincerely,
> >
> > Reviewer TXYq

---

> > > ### Public Comment · ~Feng_Liu2 · 2025-03-03
> > >
> > > Dear Reviewer TXYq,
> > >
> > > Thank you for your valuable feedback. We are pleased that we have been able to address part of your concerns and would be happy to provide further responses regarding the remaining issues. We find that there are still **major misunderstandings** in your latest concern. We will address these concerns and show why our theoretical result (thm 10) and proof are not wrong below.
> > >
> > > >[Q1] These observations suggest that the proposed argument in your paper does not sufficiently establish the advantage of the new method:
> > > >- R1. Under the null hypothesis $P=Q$, $r_N^1$ and $r_N^2$ coincide.
> > > >- R2. In general, changing only the scale of a test statistic does not alter its power.
> > >
> > > [A1] Thank you for highlighting this issue.
> > >
> > > - We acknowledge that R1 is correct **under null hypothesis $H_0:P=Q$** and, **for notational simplicity**, we adopt the form $r_N^1$ in our paper. In practice, because **the condition $P=Q$ is unknown**, we employ the **permutation test** to estimate the null distribution, where **samples under the null hypothesis** are drawn by mixing the original samples $X\sim P$ and $Y\sim Q$, resulting in the mixed distribution$(P+Q)/2$. Accordingly, we use the notation
> > > $$
> > > r_N^1=\frac{r_M}{4K-||(\mu_P+\mu_Q)/\sqrt{2}||^2}\ .
> > > $$
> > >
> > > - The observation **R2 is incorrect**.  Following the abstract framework outlined in the review, we provide a detailed explanation below.
> > >
> > >   Let $R$ be a statistic with null distribution $P_{null}(R)$ and alternative distribution $P_{alt}(R)$. For clarity, we denote its realizations under $H_0$ as $R^{null}$ and under $H_1$ as $R^{alt}$. The Type I error $\alpha$ of the original test, i.e., the probability of rejecting $H_0$ with test statistic $R^{null}$, is defined as
> > >   $$
> > >   P_{null}(R^{null}>r)=\alpha.
> > >   $$
> > >   Now consider a scaled version of the test statistic $R^{null}/\sigma_{null}$. To maintain the same Type I error rate, we define the threshold as $r'=r/\sigma_{null}$, which satisfies
> > >   $$
> > >   P_{null}\left(\frac{R^{null}}{\sigma_{null}}>r'\right)=\alpha.
> > >   $$
> > >   Then, under the alternative hypothesis $H_1$, the power of the test using the scaled statistic becomes
> > >   $$
> > >   P_{alt}\left(\frac{R^{alt}}{\sigma_{alt}}>r'\right)=P_{alt}\left(\frac{R^{alt}}{\sigma_{alt}}>\frac{r}{\sigma_{null}}\right) .
> > >   $$
> > >   Notably, **the test power is calculated under alternative hypothesis $H_1:P\neq Q$**, where we have $\sigma_{alt}=4K-||\mu_P||^2-||\mu_Q||^2$ and $\sigma_{null}=4K-||(\mu_P+\mu_Q)/\sqrt{2}||^2$ based on permutation test. Since we have proven that $\sigma_{alt}\leq\sigma_{null}$, it follows that
> > >   $$
> > >   P_{alt}\left(\frac{R^{alt}}{\sigma_{alt}}>\frac{r}{\sigma_{null}}\right)\geq P_{alt}(R^{alt}>r).
> > >   $$
> > >   This inequality confirms that scaling the test statistic increases the power of the test, **thereby resolving the reviewer’s concern and substantiating the correctness of Theorem 10**.
> > >
> > > For enhanced clarity, we note that $R^{null}$ and $R^{null}/\sigma_{null}$ correspond to MMD and NAMMD under null hypothesis $H_0$; Similarly, $R^{alt}$ and $R^{alt}/\sigma_{alt}$ represent the MMD and NAMMD under alternative hypothesis $H_1$.
> > >
> > > >[Q2] A more fruitful approach might therefore involve analyzing exact (non-asymptotic) distributions or exploring alternative properties of the statistic to uncover the potential strengths of NNMMD. This situation is analogous to the relationship between the statistic and the sample mean: although the test is uniformly most powerful unbiased, the detection powers of both tests become identical as the sample size increases (assuming finite variance).
> > >
> > > [A2] Thanks for pointing out this issue. Analysis based on the exact distribution is an interesting topic. However, deriving an exact distribution for the test statistic **requires additional assumptions on the distribution of samples, e.g., assuming the samples are drawn from a standard normal distribution in t-test**. In our paper, we focus on a non-parametric hypothesis testing approach, which does not impose assumptions on the underlying data distribution. This is particularly relevant in machine learning, where data distributions are often unknown or complex.
> > >
> > > For non-parametric hypothesis testing, the asymptotic distribution plays a critical role in determining the validity and reliability of the test, along with the corresponding analysis of test power and Type I error, **providing insights into its consistency and robustness across different data distributions** [1-11].

---

> ### Public Comment · ~Feng_Liu2 · 2025-03-03
>
> **References**
>
> [1] Gretton, A., Borgwardt, K. M., Rasch, M. J., Schölkopf, B., & Smola, A. (2012). A kernel two-sample test. The Journal of Machine Learning Research, 13(1), 723-773.
>
> [2] Liu, F., Xu, W., Lu, J., Zhang, G., Gretton, A., & Sutherland, D. J. (2020, November). Learning deep kernels for non-parametric two-sample tests. In International conference on machine learning (pp. 6316-6326). PMLR.
>
> [3] Korolyuk, V. S., & Borovskich, Y. V. (2013). Theory of U-statistics (Vol. 273). Springer Science & Business Media.
>
> [4] Lee, A. J. (2019). U-statistics: Theory and Practice. Routledge.
>
> [5] Van der Vaart, A. W. (2000). Asymptotic statistics (Vol. 3). Cambridge university press.
>
> [6] Serfling, R. J. (2009). Approximation theorems of mathematical statistics. John Wiley & Sons.
>
> [7] Hoeffding, W. (1992). A class of statistics with asymptotically normal distribution. Breakthroughs in statistics: Foundations and basic theory, 308-334.
>
> [8] Lehmann, E. L. (1951). Consistency and unbiasedness of certain nonparametric tests. The annals of mathematical statistics, 165-179.
>
> [9] Shekhar, S., & Ramdas, A. (2023). Nonparametric two-sample testing by betting. IEEE Transactions on Information Theory, 70(2), 1178-1203.
>
> [10] Sutherland, D. J., Tung, H. Y., Strathmann, H., De, S., Ramdas, A., Smola, A., & Gretton, A. Generative Models and Model Criticism via Optimized Maximum Mean Discrepancy. In International Conference on Learning Representations.
>
> [11] Gao, R., Liu, F., Zhang, J., Han, B., Liu, T., Niu, G., & Sugiyama, M. (2021, July). Maximum mean discrepancy test is aware of adversarial attacks. In International Conference on Machine Learning (pp. 3564-3575). PMLR.
>
> Best regards,
>
> Feng

---

> > ### Comment · Reviewer_TXYq · 2025-03-03
> >
> > Dear Dr. Liu,
> >
> > Thank you once again for your thoughtful and detailed responses throughout this extended discussion. I truly appreciate your dedication and the significant effort you have put into clarifying your position regarding the theoretical aspects of your paper.
> >
> >
> >
> > I sincerely regret that my identification of the potential issue with Theorem 10 came after the rebuttal period, thus leaving you without an appropriate opportunity to respond during the formal review process.
> > Additionally, I acknowledge that my explanation of the issue may not have been sufficiently clear or complete.
> >
> >
> >
> > In light of these circumstances, I deeply apologize for any unnecessary stress or difficulty my delayed and insufficiently detailed comments may have caused you. Moving forward, I will make every effort to ensure that such a situation does not arise again, being more careful and thorough in my reviews and timely in communicating any concerns.
> >
> > Given that the formal review process has concluded, I believe it would be constructive for both sides to respectfully acknowledge the differences in our interpretations of the theoretical results, without pressing this matter further.
> >
> > Your empirical results clearly demonstrate the potential of your proposed approach, and I genuinely look forward to seeing further development of your method in future work.
> >
> > Thank you again for your patience and understanding.
> >
> > Best regards,
> >
> > Reviewer TXYq

---

> > > ### Public Comment · ~Feng_Liu2 · 2025-03-25
> > > **Conclusion on the disagreement**
> > >
> > > Dear Reviewer TXYq,
> > >
> > > Thank you for your kind and considerate message. We sincerely appreciate the time and effort you have devoted to reviewing our work and engaging in this extended discussion. Your insights and critical feedback have been invaluable, and we recognize the challenges that come with providing a rigorous and timely review. **As we conclude this discussion, we would like to summarize the key points addressed and remain open to further discussions on the OpenReview platform with future readers.**
> > >
> > > Best,
> > >
> > > Feng
> > >
> > > ---
> > >
> > > ### **Summary of Key Points**
> > >
> > > The reviewer initially identified a potential technical flaw in the proof of Theorem 10, **which stemmed from a key misunderstanding about the testing threshold $r_N$**. Specifically, the reviewer assumed that the threshold can be given by
> > > $$
> > > r_N^2=\frac{r_M}{4K-||\mu_P||^2-||\mu_Q||^2}\ .
> > > $$
> > >
> > > However, **this formulation is not valid under the alternative hypothesis $H_1:P\neq Q$**, which is the setting in which the test power is evaluated and also **the scenario considered in Theorem 10**. As we know, **the testing threshold $r_N$ should be calculated under the null hypothesis $H_0:P=Q$**. Therefore, in practice, when **only samples $X\sim P$ and $Y\sim Q$ are available under the alternative hypothesis**, it is necessary to **construct the null samples** based on these observed samples.
> > >
> > > Building on this, we employ the **permutation test** to estimate $r_N$ (reviewer might misunderstood this with a high probability), where null samples are constructed by randomly shuffling and reassigning the original samples $X\sim P$ and $Y\sim Q$ into two new samples $X'\sim P'=(P+Q)/2$ and $Y'\sim Q'=(P+Q)/2$. With $X'$ and $Y'$, we have **the testing threshold under null hypothesis $H_0:P'=Q'$** as follows
> > > $$
> > > r_N^2=\frac{r_M}{4K-||(\mu_P+\mu_Q)/2||^2-||(\mu_P+\mu_Q)/2||^2}\ ,
> > > $$
> > > **which is equivalent to the formulation used in our paper:**
> > > $$
> > > r_N^1=\frac{r_M}{4K-||(\mu_P+\mu_Q)/\sqrt{2}||^2}\ .
> > > $$
> > > **This addresses the reviewer's concern and provides further validation of the correctness of Theorem 10.**
> > >
> > > We sincerely welcome further discussions on the OpenReview platform and look forward to engaging in meaningful exchanges with future readers to enhance the clarity and impact of our work.

---

> > > > ### Comment · Reviewer_TXYq · 2025-03-25
> > > > **Resolution of My Concern Regarding Your Proof**
> > > >
> > > > Dear Dr. Liu,
> > > >
> > > > Thank you very much for your latest message and the clear summary of our discussion. I have carefully reviewed your explanation once again, and I would like to sincerely let you know that all of my previous concerns have now been fully resolved.
> > > >
> > > > In particular, I now recognize that the central misunderstanding on my part lay in how I reasoned about the role of the test statistic in the evaluation of test power. Specifically, I had previously assumed the following to be true:
> > > >
> > > > > 1. **Test Statistic $T$:**
> > > > > The test statistic $T$ is a numerical value calculated from the sample data that reflects how much the data deviate from what is expected under the null hypothesis $H_0$.
> > > >
> > > > Through your detailed explanation, I have come to understand that this perspective was not appropriate when discussing test power. More precisely, **the test statistic $T$ should be considered with respect to the *test scenario under the alternative hypothesis*, not the null hypothesis**, which completely resolves the concerns I previously raised regarding the proof of Theorem 10.
> > > >
> > > > I would like to offer my sincere apologies to you on two fronts. First, I regret that my concern was raised only after the rebuttal phase, thereby preventing you from addressing it during the formal review process. Second, and more significantly, I apologize that my inaccurate critique has unfairly impacted the evaluation of your work.
> > > >
> > > > I deeply appreciate your patience, thoughtfulness, and commitment to this discussion.
> > > > Your thorough responses not only helped me refine my own understanding of hypothesis testing in this context but also improved my approach to reviewing and engaging in rebuttal processes.
> > > >
> > > > I truly look forward to seeing further developments of your research and future contributions to the field.
> > > >
> > > > Warmest regards,
> > > > Reviewer TXYq

---

> ### Public Comment · ~Feng_Liu2 · 2025-03-25
> **Happy to see your concerns are finally addressed!**
>
> Dear Reviewer TXYq,
>
> We are pleased to hear that your concerns have finally been clarified! ^^
>
> To avoid misunderstanding among future readers, may we have your responses to the first public comment (top of comments and just below the abstract) to say your concerns are addressed after the post-decision discussion?
>
> Then, we can safely arxiv this paper without flaw conflicts in public venues.
>
> Best regards,
>
> Feng

---

> > ### Comment · Reviewer_TXYq · 2025-03-26
> > **Re: Happy to see your concerns are finally addressed!**
> >
> > Dear Dr. Feng,
> >
> > Thank you very much for your message and suggestion.
> > I fully agree with your proposal and would be happy to explicitly confirm in the public comment section that all my previous concerns have been completely resolved through our post-decision discussion.
> >
> > I have posted the following statement in your suggested location:
> > > #### Statement from Reviewer TXYq (Post-Decision Clarification on March 26, 2025):
> > >
> > > After careful reconsideration and thorough discussions with the authors following the decision, I now confirm that all of my previously raised concerns regarding the proof of Theorem 10 have been fully addressed. My earlier comment reflected a misunderstanding on my part about the role of the test statistic under the alternative hypothesis scenario. This issue has been clarified entirely by the authors, and I no longer have any reservations concerning this work.
> >
> >
> > Warm regards,
> >
> > Reviewer TXYq

---

### Author Response · Authors · 2024-11-28
**Revision Details According to Reviews**

We sincerely thank the reviewers for their insightful comments, which have greatly contributed to improving our work. We have carefully revised our paper based on the discussions and feedback provided.

>Based on the discussions with **Reviewer TXYq**, we have adopted a more cautious tone in the paper to avoid potential overclaims, with revisions as follows.
- We emphasize in the **abstract, introduction, and other relevant parts** that this paper's analysis applies to **kernels with form** $\kappa(x,x')=\Psi(x-x')\leq K$ for a positive-definite $\Psi(\cdot)$ and $\Psi(0)=K$, which includes the Gaussian, Laplace, Mahalanobis, and Deep kernels (some frequently used kernels in kernel-based hypothesis testing).
- We emphasize in the **abstract, introduction, and other relevant parts** that the issue identified in the paper applies to assessing the closeness levels for multiple distribution pairs using **the same kernel**.
- We update **Figure 1 caption** to indicate that it represents **an empirical example** with Gaussian kernel, but extendable to other kernels with above form.
- We add **a paragraph in the introduction** to include the current **kernel selection methods** for TST and discuss the global kernel selection for DCT, where identifying the optimal global kernel remains an future work. This is due to the fact that deriving a test power estimator for DCT with multiple distribution pairs and selecting a optimal global  kernel based on this estimator remains an open question and poses a significant challenge.
- We also add **a paragraph in relevant work** about **background of kernel selection methods** for TST with MMD and further discuss kernel selection for NAMMD in DCT.
- We add content immediately after **Theorem 10(original Theorem 9)** to discuss the improved test power of NAMMD compared to MMD under **the identical optimal kernel** for **two-sample testing**.
- We add content immediately after **Theorem 12(original Theorem 11)** to discuss the improved test power of NAMMD compared to MMD under **respective optimal kernels** for **distribution closeness testing**.
- We add further details regarding the above two points in **Appendix B.10**.
- We add experiments comparing NAMMD and MMD with their respectively and empirically selected kernels in distribution closeness testing. The experiments results is given in Table 5 in **Appendix D.6**.
- We add additional details on **kernel selection** in experiments.
- We present the kernel selection for distribution closeness testing **as a future work in conclusion**.
- We add a **Limitation Statement** in **Appendix C.4** regarding the applicability of our findings to other kernels, where the norms of distributions may increase in accordance with increases in the data variance, including the discussions on unbounded kernels and kernels with a positive limit at infinity.

>Based on the discussions with **Reviewer TqLo**, we present three case studies (with details on pages 9 and 10) demonstrating the application of our NAMMD test on complex, high-dimensional datasets as follows.
- Testing whether the pre-trained ResNet50 performs similarly for **original ImageNet and its variants (ImageNet-SK, ImageNet-R, Imagenet-V2, ImageNet-A)**. The ground-truth similarities are measured by the accuracy margin (the difference in model accuracy between the ImageNet and its variant) with ground-truth labels. We demonstrate that our NAMMD distance (do not need ground-truth labels) effectively reflects the closeness relationships indicated by the accuracy margin, and the NAMMD test successfully identifies these relationship, as shown in **Figure 3**.
- Testing whether the pre-trained ResNet50 performs similarly for each class of **the original ImageNet and ImageNetV2**. Similar to the first case, but considering the limited sample size, the sample sizes for each class of ImageNet and ImageNetV2 are 50 and 10, respectively. The ground-truth similarities are measured by the confidence margin (as the accuracy margin may be dispersed and fail to reliably capture differences in model performance) with ground-truth labels. We also demonstrate that our NAMMD distance effectively reflects the closeness relationships indicated by the confidence margin, and the NAMMD test successfully identifies these relationships, as shown in **Figure 4**.
- Testing whether the trained ResNet18 performs similarly for **original CIFAR10 and CIFAR10 after adversarial perturbation**. As expected, a larger perturbation generally result in poor model performance on the perturbed CIFAR10 dataset. We demonstrate that our NAMMD distance effectively reflects the level of adversarial perturbation, and the NAMMD test successfully assesses different adversarial perturbation levels, as shown in **Figure 5**.
- We present detailed formalizations of accuracy and confidence margins in **Appendix D.5**.
- We revise the **last paragraph of introduction** to include the above-mentioned three case studies.

---

> ### Author Response · Authors · 2024-11-28
>
> >Based on the discussions with **Reviewer u8Wh**, we have included additional theoretical analysis and provided details about NAMMDFuse as follows.
> - We present the **asymptotic behavior of the variance estimator $\sigma_{X,Y}$** in **Lemma 4** (renumber subsequent Definitions, Lemmas, and Theorems accordingly), along with the corresponding detailed proofs in **Appendix B.3**. Specifically, we prove that $|E[\sigma^2_{X,Y}] - \sigma^2_{P,Q}|=O\left(1/\sqrt{m}\right)$ with sample size $m$.
> - We present the details of the **NAMMDFuse** method in **Appendix C.3**. Specifically, following the fusing statistics approach of [Biggs et al., 2023], we introduce the NAMMDFuse statistic through exponentiation of NAMMD, with kerenl space and prior kernel distribution to be the same for MMDFuse.
>
> >Based on the discussions with **Reviewer 8GF1**, we have provided more explanations of the definition of our NAMMD as follows.
> - We present a **Remark** immediately after **the definition of NAMMD in Equation (1)**, providing a detailed explanation of why the definition of NAMMD is formulated as Equation (1), why it is sufficient to achieve the desired goal, and why we do not redesign a kernel-based closeness testing method without considering MMD. Essentially, we capture differences between two distributions using their characteristic kernel mean embeddings (i.e. $\mu_P$ and $\mu_Q$). A natural way is to calculate the distancen between two mean embeddings (i.e. $||\mu_P-\mu_Q||^2_H$), which is the MMD distance. However, we find that the MMD is less informative in distribution closeness with a fixed kernel. We improve it by incorporating the norms information $4K-||\mu_P||^2_H-||\mu_Q||^2_H$, as shown in the NAMMD formulation in Equation (1).
>
> >Other relevant revisions.
> - Delete the limitation parts in preliminaries to save space.
> - Move kernel details for experiments of Table 1 to Appendix D.4 to save space
> - Move details of six state-of-the-art two-sample testing methods for experiments of Figure 2 to Appendix D.3 to save space
> - Move original Figure 3 and Table 3, along with corresponding experiment details, to Appendix D.6 (now Figure 6 and Table 4) to save space.
> - Add one reference: [40] B. K. Sriperumbudur, K. Fukumizu, and G. Lanckriet. Universality, characteristic kernels and RKHS embedding of measures. Journal of Machine Learning Research, 12:2389–2410, 2011.

---

### Public Comment · ~Feng_Liu2 · 2025-02-14
**Post-decision statement: Factual error in Reviewer TXYq's after-dicussion comments (Part I: Testing threshold in two-sample testing)**

Dear Reviewer TXYq,

Thank you for reading our paper and for your further reviews. Unfortunately, **the issue was raised 20 days after the rebuttal period**, and we cannot reply through the OpenReview system after the author rebuttal deadline. Although we have reached out to ICLR program chairs via email, we were still unable to resolve this matter.

We address your concern below and hope that future readers will find our clarification helpful. Although the paper has already been rejected (based on meta-review, the decision is made **mainly based on your new concern**, and **the score changes from 8866 to 8863**), we appreciate the opportunity to provide further insights and contribute to the ongoing discussion with you.

>[Q1] In particular, the following test statistic seems to have been used in the calculation of the test power:
$$
r_N = \frac{r_M}{4K-||(\mu_P+\mu_Q)/\sqrt{2}||^2}\ .
$$
According to lines 1172–1174, this test statistic (Equation (E1)) assumes that $\mu_P=\mu_Q=(\mu_P+\mu_Q)/2$, as discussed in lines 848–849. Thus, it appears to be derived under the null hypothesis. However, since the test power is defined as the probability of rejecting the null hypothesis under the alternative hypothesis, the appropriate test statistic should instead be:
$$
r_N = \frac{r_M}{4K-||\mu_P||^2+||\mu_Q||^2}\ .
$$

[A1] We believe there is a **key misunderstanding about the testing thresholds $r_M$ and $r_N$.** These thresholds should be determined based on **the null distribution** [1-4] (the distribution of the test statistics under the null hypothesis, i.e., $H_0: P = Q$ for Theorem 10 ). **In contrast,**, in your additional comments, we found that you might think the **testing thresholds** should be from **the alternative distribution** (the distribution of test statistics under the alternative hypothesis, i.e., $H_1:P\neq Q$), **which is incorrect**.

To clarify this point, we first introduce the background of hypothesis testing. Hypothesis testing evaluates whether there is enough evidence to reject a null hypothesis $H_0$ in favor of an alternative hypothesis $H_1$. The process involves defining a **test statistic $T$** and comparing it to a **testing threshold $r$** to determine whether to reject the null hypothesis, where
1. Test Statistic $T$: The test statistic $T$ is a numerical value calculated from the sample data that reflects how much the data deviates from what is expected under the null hypothesis.
2. Testing Threshold $r$:  The testing threshold is determined based on the significance level $\alpha$, which represents the probability of rejecting $H_0$ **when it is true**. Hence, this threshold is often derived from the $(1-\alpha$)-quantile of the (null) distribution of the test statistic **under null hypothesis $H_0$**.

Hence, the test power (i.e., **the probability of correctly rejecting the null hypothesis when the alternative hypothesis is true**) is given by

$$
\Pr [T\geq r]\ ,
$$

which represents the probability that the **test statistic $T$ (calculated under alternative hypothesis where two samples are from different distributions)** exceeds the testing threshold $r$ (calculated under the **null hypothesis**).

---

> ### Public Comment · ~Feng_Liu2 · 2025-02-14
> **Post-decision statement: Factual error in Reviewer TXYq's after-dicussion comments (Part II: Why our proof is correct)**
>
> Then, we will show **why our $r_N$ is correct** and why NAMMD has higher test power than MMD. In our paper, for Theorem 10, which addresses the two-sample testing with hypotheses
> $$
> H_0: P=Q  \quad \quad and \quad \quad H_1: P\neq Q\ .
> $$
>
> For MMD test, we have the test statistic $T=m\widehat{\text{MMD}}$ and the asymptotic testing threshold $r_M$, which is the $(1-\alpha)$-quantile of **the null distirbution** of the test statistic, which is calculated **under the null hypothesis $H_0: P=Q$**. The corresponding test power is given by
> $$
> p_M = \Pr[m\widehat{\text{MMD}}\geq r_M]\ .
> $$
>
> In a similar manner, for NAMMD test, we have the test statistic $T=m\widehat{\text{NAMMD}}$ and the asymptotic testing threshold $r_N$, which is the $(1-\alpha)$-quantile of **the null distirbution** of the test statistic, calculated **under the null hypothesis $H_0: P=Q$**. Hence, we have that
> $$
> r_N = \frac{r_M}{4K-||(\mu_P+\mu_Q)/\sqrt{2}||^2}\ ,
> $$
> meaning that our calculated $r_N$ is correct, and **the $r_N$ in your comment is incorrect**. Furthermore, the test power of NAMMD test is given by
> $$
> p_N = \Pr[m\widehat{\text{NAMMD}}\geq r_N]=\Pr[m\widehat{\text{NAMMD}}\geq \frac{r_M}{4K-||(\mu_P+\mu_Q)/\sqrt{2}||^2}]\ .
> $$
>
> Consequently, the difference between the test powers of MMD and NAMMD is
> $$
> \varsigma=p_N-p_M=\Pr[m\widehat{\text{NAMMD}}\geq \frac{r_M}{4K-||(\mu_P+\mu_Q)/\sqrt{2}||^2}] - \Pr[m\widehat{\text{MMD}}\geq r_M]\ .
> $$
> **This aligns the critical result of our proof in line 1236 and confirms the correctness of Theorem 10.**
>
> Building on this, we prove in Theorem 10 that under the alternative hypothesis, if MMD test **rejects null hypothesis correctly** (i.e., $m\widehat{MMD}$>$r_M$, where $m\widehat{MMD}$ is calculated under alternative hypothesis and $r_M$ is calculated under null hypothesis), our NAMMD test also do so with high probability (i.e., $m\widehat{NAMMD}$>$r_N$, where $m\widehat{NAMMD}$ is calculated under alternative hypothesis and $r_N$ is calculated under null hypothesis). Moreover, NAMMD can reject null hypothesis in cases where MMD fails, **resulting in higher test power. This is consistent with our experimental results, where NAMMD demonstrates better performance.**
>
> All the below **references** mentioned that the testing threshold is calculated **under the null hypothesis**.
>
> [1] W. Jitkrittum, Z. Szabó, K. P. Chwialkowski, and A. Gretton. Interpretable distribution features with maximum testing power. In Advances in Neural Information Processing Systems 29 pages 181–189. Curran Associates, Dutchess, NY, 2016.
> **"The threshold is given by the $(1 − \alpha)$-quantile of the (asymptotic) distribution of $\lambda^n$ under $H_0$, the null distribution."**
>
> [2] K. Chwialkowski, A. Ramdas, D. Sejdinovic, and A. Gretton. Fast two-sample testing with analytic representations of probability measures. In Advances in Neural Information Processing Systems 28, pages 1981–1989. Curran Associates, Dutchess, NY, 2015.
> **"the test threshold $T_\alpha$ is given by the $(1 − \alpha)$-quantile of the asymptotic null distribution $\chi^2(J)$"**
>
> [3] Gretton, K. M. Borgwardt, M. J. Rasch, B. Schölkopf, and A. Smola. A kernel two-sample test. Journal of Machine Learning Research, 13(1):723–773, 2012.
> **"define a test threshold under the null hypothesis $P = Q$"**
>
> [4] F. Biggs, A. Schrab, and A. Gretton. MMD-Fuse: Learning and combining kernels for two-sample testing without data splitting. In Advances in Neural Information Processing Systems36. Curran Associates, Dutchess, NY, 2023.
> **"We can use permutations to construct an approximate cumulative distribution function (CDF) of our test statistic under the null, and choose an appropriate quantile of this CDF as our test threshold"**
>
> Looking forward to your further reply!
>
> Best regards,
>
> Feng

---

### Comment · Reviewer_TXYq · 2025-03-26
**Statement from Reviewer TXYq (Post-Decision Clarification on March 26, 2025)**

After careful reconsideration and thorough discussions with the authors following the decision, I now confirm that all of my previously raised concerns regarding the proof of Theorem 10 have been fully addressed. My earlier comment reflected a misunderstanding on my part about the role of the test statistic under the alternative hypothesis scenario. This issue has been clarified entirely by the authors, and I no longer have any reservations concerning this work.

---

> ### Public Comment · ~Feng_Liu2 · 2025-03-27
> **Public reply from the authors**
>
> Many thanks for this clarification! Glad to see no concerns now ^^
>
> Best,
>
> Feng

---

### Meta-Review · Area_Chair_EXzZ · 2024-12-20

**Metareview:**

This paper investigates the problem of distribution closeness testing (DCT) and proposes a novel kernel-based DCT method called norm-adaptive MMD (NAMMD), which extends the well-known MMD by accounting for distributions with relatively different norms.
Both theoretical and empirical results are provided to demonstrate the effectiveness of NAMMD.

While the authors did an excellent job during the rebuttal phase and successfully convinced the reviewers of the significance of their contribution, a **critical issue** was raised during the (further) discussion phase among reviewers.
Reviewer TXYq identified a **technical flaw in the proof of Theorem 10**, specifically pointing out that the authors appear to focus on the null distribution rather than  $H_1$  in their proof of test power (please refer to Reviewer TXYq’s updated review for details).
This flaw is significant, casting doubt on the theoretical validity of the proposed NAMMD.
From an empirical perspective, its advantages remain evident, as demonstrated by the authors' numerical results.

However, given the importance of the identified flaw, I recommend rejecting the paper.

**Additional Comments On Reviewer Discussion:**

The reviewers raised the following points:
- Clarification of many results: The authors did an excellent job during the rebuttal phase, successfully convincing the reviewers of the significance of their contribution.
- Additional experimental results: In reply to the reviewers' comments, the authors provided further empirical evidence to support their method.
- **Technical flaw in the proof of Theorem 10**, raised by Reviewer TXYq: Unfortunately, this issue was identified **after** the rebuttal phase, leaving the authors **unable** to address it.

I have carefully considered all of the above points in making my final decision.

---

> ### Public Comment · ~Feng_Liu2 · 2025-03-25
> **Request a formal clarification from Area Chair**
>
> Dear Area Chair,
>
> After the post-decision discussions, Reviewer TXYq and we finally clarified the correctness of the proof of Theorem 10, namely there is **no technical flaw** in the proof of Theorem 10 (which is the **only reason** why our paper is finally rejected based on meta-review).
>
> To avoid misunderstanding among future readers, **may we request a formal clarification** from you (e.g., reply to this comment) to demonstrate that the **critical issue in the meta-review is inaccurate** and there is actually no technical flaw in the proof of Theorem 10 after the post-decision discussion between Reviewer TXYq and the authors.
>
> Since technical flaws in proofs are serious issues, we do hope it can be formally clarified after the final agreement between us and the Reviewer TXYq has been reached.
>
> Best regards,
>
> Feng

---

> > ### Comment · Area_Chair_EXzZ · 2025-03-26
> >
> > Dear Feng,
> >
> > Thank you for your effort in providing clarification, which effectively resolves the issue raised by Reviewer TXYq.
> > (This was also a mistake on my part, as I should have recognized this earlier.)
> >
> > I sincerely regret having wrongly rejected this excellent submission from ICLR 2025.
> > I truly hope it will be accepted at another top-tier venue in the near future.
> >
> > Best regards,
> > AC of ICLR 2025 Submission 6378

---

> > > ### Public Comment · ~Feng_Liu2 · 2025-03-26
> > > **Thanks for the formal confirmation**
> > >
> > > Dear Area Chair,
> > >
> > > Many thanks for the prompt reply and confirmation! It is good that this critical issue is finally clarified.
> > >
> > > Best regards,
> > >
> > > Feng

---

### Decision · Program_Chairs · 2025-01-22

Reject